# Bringing robotics taxonomies to continuous domains via GPLVM on hyperbolic manifolds

## Abstract

Robotic taxonomies serve as high-level hierarchical abstractions that classify how humans move and interact with their environment. They have proven useful to analyse grasps, manipulation skills, and whole-body support poses. Despite substantial efforts devoted to design their hierarchy and underlying categories, their use in application fields remains limited. This may be attributed to the lack of computational models that fill the gap between the discrete hierarchical structure of the taxonomy and the high-dimensional heterogeneous data associated to its categories. To overcome this problem, we propose to model taxonomy data via hyperbolic embeddings that capture the associated hierarchical structure. We achieve this by formulating a novel Gaussian process hyperbolic latent variable model that incorporates the taxonomy structure through graph-based priors on the latent space and distance-preserving back constraints. We validate our model on three different robotics taxonomies to learn hyperbolic embeddings that faithfully preserve the original graph structure. We show that our model properly encodes unseen poses from existing or new taxonomy categories, can be used to generate trajectories between the embeddings, and outperforms its Euclidean counterparts.

## 1 Introduction

Roboticists are often inspired by biological insights to create robotic systems that exhibit human- or animal-like capabilities (Siciliano & Khatib, 2016). In particular, it is first necessary to understand how humans move and interact with their environment to then generate biologically-inspired motions and behaviors of robotics hands, arms or humanoids. In this endeavor, researchers proposed to structure and categorize human hand grasps and body poses into hierarchical classifications known as *taxonomies*. Their structure depends on the sensory variables considered to categorize human motions and their interactions with the environment, as well as on associated qualitative measures.

Different taxonomies have been proposed in the area of human and robot grasping (Cutkosky, 1989; Feix et al., 2016; Abbasi et al., 2016; Stival et al., 2019; Krebs & Asfour, 2022). Feix et al. (2016) introduced a taxonomy of hand grasps whose structure was mainly defined by the hand pose and the type of contact with the object. Later, Stival et al. (2019) claimed that the taxonomy designed in (Feix et al., 2016) heavily depended on subjective qualitative measures, and proposed a quantitative tree-like taxonomy of hand grasps based on muscular and kinematic patterns. A similar data-driven approach was used to design a grasp taxonomy based on contact forces in (Abbasi et al., 2016). Robotic manipulation also gave rise to various taxonomies. Bullock et al. (2013) introduced a hand-centric manipulation taxonomy that classifies manipulation skills according to the type of contact with the objects and the object motion imparted by the hand. A different strategy was developed in (Paulius et al., 2019), where a manipulation taxonomy was designed based on a categorization of contacts and motion trajectories. Humanoid robotics also made significant efforts to analyze human motions, thus proposing taxonomies as high-level abstractions of human motion configurations. Borràs et al. (2017) analyzed the contacts of the human limbs with the environment and designed a taxonomy of whole-body support poses.

Besides their analytical purpose in robotics or biomechanics, some of the aforementioned taxonomies were employed for modeling grasp actions (Romero et al., 2010; Lin & Sun, 2015), for planning contact-aware whole-body pose sequences (Mandery et al., 2016a), and for learning manipulation skills embeddings (Paulius et al., 2020). However, despite most of these taxonomies carry

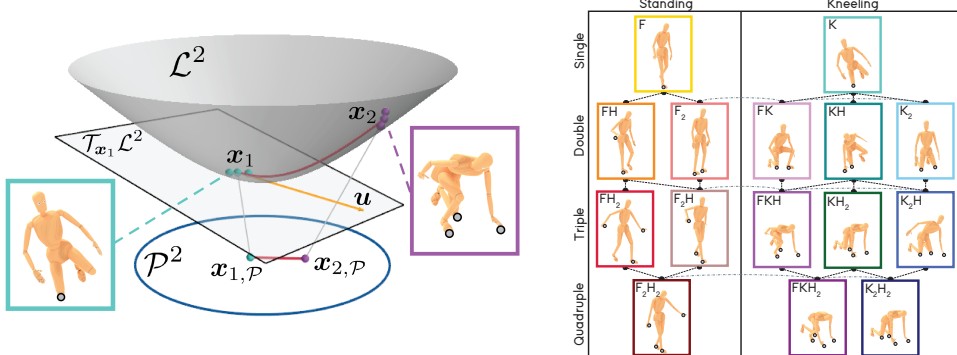

Figure 1: *Left:* Illustration of the Lorentz $\mathcal{L}^2$ and Poincaré $\mathcal{P}^2$ models of the hyperbolic manifold. The former is depicted as the gray hyperboloid, while the latter is represented by the blue circle. Both models show a geodesic (——) between two points $x_1$ (•) and $x_2$ (•). The vector $u$ (——) lies on the tangent space of $x_1$ such that $u = \mathrm{Log}_{x_1}(x_2)$. *Right:* Subset of the whole-body support pose taxonomy (Borràs et al., 2017) used in one of our experiments. Each node is a support pose defined by the type of contacts (foot F, hand H, knee K). The lines represent graph transitions between the taxonomy nodes. Contacts are depicted by grey dots.

a well-defined hierarchical structure, it was often overlooked. First, these taxonomies were usually exploited for classification tasks whose target classes were mainly the tree leaves, disregarding the full taxonomy structure (Feix et al., 2016; Abbasi et al., 2016). Second, the discrete representation of the taxonomy categories hindered their use for motion generation (Romero et al., 2010).

Arguably the main difficulty of leveraging robotic taxonomies is the lack of computational models that exploit *(i)* the domain knowledge encoded in the hierarchy, and *(ii)* the information of the high-dimensional data associated to the taxonomy categories. We tackle this problem from a representation learning perspective by modeling taxonomy data as embeddings that capture the associated hierarchical structure. Inspired by recent advances on word embeddings (Nickel & Kiela, 2017; 2018; Mathieu et al., 2019), we propose to leverage the *hyperbolic manifold* (Ratcliffe, 2019) to learn such embeddings. An important property of the hyperbolic manifold is that distances grow exponentially when moving away from the origin, and shortest paths between distant points tend to pass through it, resembling a *continuous hierarchical structure*. Therefore, we hypothesize that the geometry of the hyperbolic manifold allows us to learn embeddings that comply with the original hierarchical structure of robotic taxonomies.

**In this paper** we propose a Gaussian process hyperbolic latent variable model (GPHLVM) to learn embeddings of taxonomy data on the hyperbolic manifold. To do so, we impose a hyperbolic geometry to the latent space of the well-known GPLVM (Lawrence, 2003; Titsias & Lawrence, 2010). This demands to reformulate the Gaussian distribution, the kernel, and the optimization process of the vanilla GPLVM to account for the geometry of the hyperbolic latent space. To do so, we leverage the hyperbolic wrapped Gaussian distribution (Nagano et al., 2019), and provide a positive-definite-guaranteed approximation of the hyperbolic kernel proposed by McKean (1970). Moreover, we resort to Riemannian optimization (Absil et al., 2007; Boumal, 2022) to optimize the GPHLVM embeddings. We enforce the taxonomy graph structure in the learned embeddings through graph-based priors on the latent space and via graph-distance-preserving back constraints (Lawrence & Quiñonero Candela, 2006; Urtasun et al., 2008). Our GPHLVM is conceptually similar to the GPLVM for Lie groups introduced in (Jensen et al., 2020), which also imposes geometric properties to the GPLVM latent space. However, our formulation is specifically designed for the hyperbolic manifold and fully built on tools from Riemannian geometry. Moreover, unlike (Tosi et al., 2014) and (Jørgensen & Hauberg, 2021), where the latent space was endowed with a pullback Riemannian metric learned via the GPLVM mapping, we impose the hyperbolic geometry to the GPHLVM latent space as an inductive bias adapted to our targeted applications.

We validate our approach on three distinct robotic taxonomies: a hand grasps taxonomy (Stival et al., 2019), a taxonomy of whole-body support poses (Borràs et al., 2017), and a bimanual manipulation taxonomy (Krebs & Asfour, 2022). The proposed GPHLVM successfully learns hyperbolic embeddings that comply with the original graph structure of all the considered taxonomies, and it properly encodes unseen poses from existing or new taxonomy nodes. Moreover, we show how we can exploit the continuous geometry of the hyperbolic manifold to generate trajectories between different embeddings pairs via geodesic paths in the latent space. To the best of our knowledge, this paper is the first to leverage the hyperbolic manifold for robotic domains.

## 2 BACKGROUND

**Gaussian Process Latent Variable Models:** A GPLVM defines a generative mapping from latent variables $\{\boldsymbol{x}_n\}_{n=1}^N, \boldsymbol{x}_n \in \mathbb{R}^Q$ to observations $\{\boldsymbol{y}_n\}_{n=1}^N, \boldsymbol{y}_n \in \mathbb{R}^D$ by modeling the corresponding non-linear transformation with Gaussian processes (GPs) (Lawrence, 2003). The GPLVM is described as,

$$y_{n,d} \sim \mathcal{N}(y_{n,d}; f_{n,d}, \sigma_d^2) \quad \text{with} \quad f_{n,d} \sim \text{GP}(m_d(\boldsymbol{x}_n), k_d(\boldsymbol{x}_n, \boldsymbol{x}_n)) \quad \text{and} \quad \boldsymbol{x}_n \sim \mathcal{N}(\mathbf{0}, \boldsymbol{I}), \quad (1)$$

where $y_{n,d}$ denotes the $d$-th dimension of the observation $\boldsymbol{y}_n$, $m_d(\cdot) : \mathbb{R}^Q \mapsto \mathbb{R}$ and $k_d(\cdot, \cdot) : \mathbb{R}^Q \times \mathbb{R}^Q \to \mathbb{R}$ are the GP mean and kernel function, respectively, and $\sigma_d^2$ is a hyperparameter. Classically, the hyperparameters and latent variables of the GPLVM were optimized using *maximum likelihood* or *maximum a posteriori* (MAP) estimates. As this does not scale gracefully to large datasets, contemporary methods use inducing points and variational approximations of the evidence (Titsias & Lawrence, 2010). In contrast to neural-network-based generative models, GPLVMs are data efficient and provide automatic uncertainty quantification.

**Riemannian geometry:** To understand the hyperbolic manifold, it is necessary to first define some basic Riemannian geometry concepts (Lee, 2018). To begin with, consider a Riemannian manifold $\mathcal{M}$, which is a locally Euclidean topological space with a globally-defined differential structure. For each point $\boldsymbol{x} \in \mathcal{M}$, there exists a tangent space $\mathcal{T}_{\boldsymbol{x}}\mathcal{M}$ that is a vector space consisting of the tangent vectors of all the possible smooth curves passing through $\boldsymbol{x}$. A Riemannian manifold is equipped with a Riemannian metric, which permits to define curve lengths in $\mathcal{M}$. Shortest-path curves, called geodesics, can be seen as the generalization of straight lines on the Euclidean space to Riemannian manifolds, as they are minimum-length curves between two points in $\mathcal{M}$. To operate with Riemannian manifolds, it is common practice to exploit the Euclidean tangent spaces. To do so, we resort to mappings back and forth between $\mathcal{T}_{\boldsymbol{x}}\mathcal{M}$ and $\mathcal{M}$, which are the exponential and logarithmic maps. The exponential map $\text{Exp}_{\boldsymbol{x}}(\boldsymbol{u}) : \mathcal{T}_{\boldsymbol{x}}\mathcal{M} \to \mathcal{M}$ maps a point $\boldsymbol{u}$ in the tangent space of $\boldsymbol{x}$ to a point $\boldsymbol{y}$ on the manifold, so that it lies on the geodesic starting at $\boldsymbol{x}$ in the direction $\boldsymbol{u}$, and such that the geodesic distance $d_{\mathcal{M}}$ between $\boldsymbol{x}$ and $\boldsymbol{y}$ equals the distance between $\boldsymbol{x}$ and $\boldsymbol{u}$. The inverse operation is the logarithmic map $\text{Log}_{\boldsymbol{x}}(\boldsymbol{u}) : \mathcal{M} \to \mathcal{T}_{\boldsymbol{x}}\mathcal{M}$. Finally, the parallel transport $\text{P}_{\boldsymbol{x} \to \boldsymbol{y}}(\boldsymbol{u}) : \mathcal{T}_{\boldsymbol{x}}\mathcal{M} \to \mathcal{T}_{\boldsymbol{y}}\mathcal{M}$ operates with manifold elements lying on different tangent spaces.

**Hyperbolic manifold:** The hyperbolic space $\mathbb{H}^d$ is the unique simply-connected complete $d$-dimensional Riemannian manifold with a constant negative sectional curvature (Ratcliffe, 2019). There are several isometric models for the hyperbolic space, in particular, the Poincaré ball $\mathcal{P}^d$ and the Lorentz (hyperboloid) model $\mathcal{L}^d$ (see Fig. 1-*left*). The latter representation is chosen here as it is numerically more stable than the former, and thus better suited for Riemannian optimization. However, the Poincaré model provides a more intuitive representation and is here used for visualization. This is easily achieved by leveraging the isometric mapping between both models (see App. A for details). An important property of the hyperbolic manifold is the exponential rate of the volume growth of a ball with respect to its radius. In other words, distances in $\mathbb{H}^d$ grow exponentially when moving away from the origin, and shortest paths between distant points on the manifold tend to pass through the origin, resembling a continuous hierarchical structure. Because of this, the hyperbolic manifold is often exploited to embed hierarchical data such as trees or graphs (Nickel & Kiela, 2017; Chami et al., 2020). Although its potential to embed discrete data structures into a continuous space is well known in the machine learning community, its application in robotics is presently scarce.

**Hyperbolic wrapped Gaussian distribution:** Probabilistic models on Riemannian manifolds demand to work with probability distributions that consider the manifold geometry. We use the hyperbolic wrapped distribution (Nagano et al., 2019), which builds on a Gaussian distribution on the tangent space at the origin $\boldsymbol{\mu}_0 = (1, 0, \ldots, 0)^\mathsf{T}$ of $\mathbb{H}^d$, that is then projected onto the hyperbolic space after transporting the tangent space to the desired location. Intuitively, the construction of this wrapped distribution is as follows: *(1)* sample a point $\tilde{\boldsymbol{v}} \in \mathbb{R}^d$ from the Euclidean normal distribution $\mathcal{N}(\mathbf{0}, \boldsymbol{\Sigma})$, *(2)* transform $\tilde{\boldsymbol{v}}$ to an element of $\mathcal{T}_{\boldsymbol{\mu}_0}\mathbb{H}^d \subset \mathbb{R}^{d+1}$ by setting $\boldsymbol{v} = (0, \tilde{\boldsymbol{v}})^\mathsf{T}$, *(3)* apply the parallel transport $\boldsymbol{u} = \text{P}_{\boldsymbol{\mu}_0 \to \boldsymbol{\mu}}(\boldsymbol{v})$, and *(4)* project $\boldsymbol{u}$ to $\mathbb{H}^d$ via $\text{Exp}_{\boldsymbol{\mu}}(\boldsymbol{u})$. The resulting probability density function is,

$$\log \mathcal{N}_{\mathbb{H}^d}(\boldsymbol{x}; \boldsymbol{\mu}, \boldsymbol{\Sigma}) = \log \mathcal{N}(\boldsymbol{v}; \mathbf{0}, \boldsymbol{\Sigma}) - (d-1) \log\left(\sinh(\|\boldsymbol{u}\|_{\mathcal{L}}) / \|\boldsymbol{u}\|_{\mathcal{L}}\right), \quad (2)$$

where $\boldsymbol{v} = \text{P}_{\boldsymbol{\mu} \to \boldsymbol{\mu}_0}(\boldsymbol{u})$, $\boldsymbol{u} = \text{Log}_{\boldsymbol{\mu}}(\boldsymbol{x})$, and $\|\boldsymbol{u}\|_{\mathcal{L}} = \sqrt{\langle \boldsymbol{u}, \boldsymbol{u} \rangle_{\boldsymbol{\mu}}}$. The hyperbolic wrapped distribution (Nagano et al., 2019) has a more general expression given in (Skopek et al., 2020).

## 3  GAUSSIAN PROCESS HYPERBOLIC LATENT VARIABLE MODEL

We present the proposed GPHLVM, which extends GPLVM to hyperbolic latent spaces. A GPHLVM defines a generative mapping from the hyperbolic latent space $\mathbb{H}^Q$ to the observation space, e.g. the data associated to the taxonomy, based on GPs. By considering independent GPs across the observation dimensions, the GPHLVM is formally described as,

$$y_{n,d} \sim \mathcal{N}(y_{n,d}; f_{n,d}, \sigma_d^2) \quad \text{with} \quad f_{n,d} \sim \text{GP}(m_d(\boldsymbol{x}_n), k_d^{\mathbb{H}^Q}(\boldsymbol{x}_n, \boldsymbol{x}_n)) \quad \text{and} \quad \boldsymbol{x}_n \sim \mathcal{N}_{\mathbb{H}^Q}(\boldsymbol{\mu}_0, \alpha \boldsymbol{I}), \tag{3}$$

where $y_{n,d}$ denotes the $d$-th dimension of the observation $\boldsymbol{y}_n \in \mathbb{R}^D$ and $\boldsymbol{x}_n \in \mathbb{H}^Q$ is the corresponding latent variable. Our GPHLVM is built on hyperbolic GPs, characterized by a mean function $m_d(\cdot) : \mathbb{H}^Q \to \mathbb{R}$ (usually set to 0), and a kernel $k_d^{\mathbb{H}^Q}(\cdot, \cdot) : \mathbb{H}^Q \times \mathbb{H}^Q \to \mathbb{R}$. These kernels encode similarity information in the latent hyperbolic manifold and should reflect its geometry to perform effectively, as detailed in §. 3.1. Also, the latent variable $\boldsymbol{x} \in \mathbb{H}^Q$ is assigned a hyperbolic wrapped Gaussian prior $\mathcal{N}_{\mathbb{H}^Q}(\boldsymbol{\mu}_0, \alpha \boldsymbol{I})$ based on equation 2, where $\boldsymbol{\mu}_0$ is the origin of $\mathbb{H}^Q$, and the parameter $\alpha$ controls the spread of the latent variables in $\mathbb{H}^Q$. As Euclidean GPLVMs, our GPHLVM can be trained by finding a MAP estimate or via variational inference. However, special care must be taken to guarantee that the latent variables belong to the hyperbolic manifold, as explained in §. 3.2.

### 3.1  HYPERBOLIC KERNELS

For GPs in Euclidean spaces, the squared exponential (SE) and Matérn kernels are standard choices (Rasmussen & Williams, 2006). In the modern machine learning literature these were generalized to non-Euclidean spaces such as manifolds (Borovitskiy et al., 2020; Jaquier et al., 2021) or graphs (Borovitskiy et al., 2021). The generalized SE kernels may be connected to the much studied *heat kernels*. These are given (cf. Grigoryan & Noguchi (1998)) by,

$$k^{\mathbb{H}^2}(\boldsymbol{x}, \boldsymbol{x}') = \frac{\sigma^2}{C_\infty} \int_\rho^\infty \frac{s e^{-s^2/(2\kappa^2)}}{(\cosh(s) - \cosh(\rho))^{1/2}} \mathrm{d}s, \quad k^{\mathbb{H}^3}(\boldsymbol{x}, \boldsymbol{x}') = \frac{\sigma^2}{C_\infty} \frac{\rho}{\sinh \rho} e^{-\rho^2/(2\kappa^2)}, \tag{4}$$

where $\rho = \text{dist}_{\mathbb{H}^d}(\boldsymbol{x}, \boldsymbol{x}')$ denotes the geodesic distance between $\boldsymbol{x}, \boldsymbol{x}' \in \mathbb{H}^d$, $\kappa$ and $\sigma^2$ are the kernel lengthscale and variance, and $C_\infty$ is a normalizing constant. To the best of our knowledge, no closed form expression for $\mathbb{H}^2$ is known. To approximate the kernel in this case, a discretization of the integral is performed. One appealing option is the Monte Carlo approximation based on the truncated Gaussian density. Unfortunately, such approximations easily fail to be positive semidefinite if the number of samples is not very large. We address this via an alternative Monte Carlo approximation,

$$k^{\mathbb{H}^2}(\boldsymbol{x}, \boldsymbol{x}') \approx \frac{\sigma^2}{C_\infty'} \frac{1}{L} \sum_{l=1}^L s_l \tanh(\pi s_l) e^{(2s_l i + 1)\langle \boldsymbol{x}_{\mathcal{P}}, \boldsymbol{b}_l \rangle} \overline{e^{(2s_l i + 1)\langle \boldsymbol{x}'_{\mathcal{P}}, \boldsymbol{b}_l \rangle}}, \tag{5}$$

where $\langle \boldsymbol{x}_{\mathcal{P}}, \boldsymbol{b} \rangle = \frac{1}{2} \log \frac{1 - |\boldsymbol{x}_{\mathcal{P}}|^2}{|\boldsymbol{x}_{\mathcal{P}} - \boldsymbol{b}|^2}$ is the hyperbolic outer product with $\boldsymbol{x}_{\mathcal{P}}$ being the representation of $\boldsymbol{x}$ as a point on the Poincaré disk $\mathcal{P}^2 = \mathbb{D}$, $i$, $\overline{z}$ denote the imaginary unit and complex conjugation, respectively, $\boldsymbol{b}_l \overset{\text{i.i.d.}}{\sim} U(\mathbb{T})$ with $\mathbb{T}$ the unit circle, and $s_l \overset{\text{i.i.d.}}{\sim} e^{-s^2\kappa^2/2} \mathbb{1}_{[0,\infty)}(s)$. The distributions of $\boldsymbol{b}_l$ and $s_l$ are easy to sample from: The former is sampled by applying $x \to e^{2\pi i x}$ to $x \sim U([0,1])$ and the latter is (proportional to) a truncated normal distribution. Importantly, the right-hand side of equation 5 is easily recognized to be an inner product in the space $\mathbb{C}^L$, which immediately implies its positive semidefiniteness (see App. B for the development of equation 5). Note that hyperbolic kernels for $\mathbb{H}^Q$ with $Q > 3$ are generally defined as integrals of the kernels equation 4 (Grigoryan & Noguchi, 1998). Analogs of Matérn kernels for $\mathbb{H}^Q$ are obtained as integral of the SE kernel of the same dimension (Jaquier et al., 2021).

### 3.2  MODEL TRAINING

As in the Euclidean case, training the GPHLVM is equivalent to finding optimal latent variables $\boldsymbol{\mathcal{X}} = \{\boldsymbol{x}_n\}_{n=1}^N$ and hyperparameters $\boldsymbol{\Theta} = \{\theta_d\}_{d=1}^D$ by solving $\arg\max_{\boldsymbol{\mathcal{X}}, \boldsymbol{\Theta}} \mathcal{L}$, with $\boldsymbol{x}_n \in \mathbb{H}^Q$, $\theta_d$ being the hyperparameters of the $d$-th GP, and $\mathcal{L}$ as a loss function. For small datasets, the GPHLVM can be trained by maximizing the log posterior of the model, i.e., $\mathcal{L}_{\text{MAP}} = \log\left(p(\boldsymbol{Y}|\boldsymbol{X})p(\boldsymbol{X})\right)$ with $\boldsymbol{Y} = (\boldsymbol{y}_1 \ldots \boldsymbol{y}_N)^\mathsf{T}$ and $\boldsymbol{X} = (\boldsymbol{x}_1 \ldots \boldsymbol{x}_N)^\mathsf{T}$. For large datasets, the GPHLVM can be trained, similarly to the so-called Bayesian GPLVM (Titsias & Lawrence, 2010), by maximizing the marginal likelihood of the data, i.e., $\mathcal{L}_{\text{MaL}} = \log p(\boldsymbol{Y}) = \log \int p(\boldsymbol{Y}|\boldsymbol{X})p(\boldsymbol{X})d\boldsymbol{X}$. As this quantity is intractable, it is approximated via variational inference by adapting the methodology introduced in (Titsias & Lawrence, 2010) to hyperbolic latent spaces, as explained next.

**Variational inference:** We approximate the posterior $p(\boldsymbol{X}|\boldsymbol{Y})$ by a variational distribution $q(\boldsymbol{X})$ defined as a hyperbolic wrapped normal distribution over the latent variables, i.e.,

$$q_\phi(\boldsymbol{X}) = \prod_{n=1}^{N} \mathcal{N}_{\mathbb{H}^Q}(\boldsymbol{x}_n; \boldsymbol{\mu}_n, \boldsymbol{\Sigma}_n), \tag{6}$$

with variational parameters $\phi = \{\boldsymbol{\mu}_n, \boldsymbol{\Sigma}_n\}_{n=1}^{N}$, with $\boldsymbol{\mu}_n \in \mathbb{H}^Q$ and $\boldsymbol{\Sigma}_n \in \mathcal{T}_{\boldsymbol{\mu}_n}\mathbb{H}^Q$. Similarly to the Euclidean case (Titsias & Lawrence, 2010), this variational distribution allows the formulation of a lower bound,

$$\log p(\boldsymbol{Y}) \geq \mathbb{E}_{q_\phi(\boldsymbol{X})}[\log p(\boldsymbol{Y}|\boldsymbol{X})] - \mathrm{KL}\big(q_\phi(\boldsymbol{X})||p(\boldsymbol{X})\big). \tag{7}$$

The KL divergence $\mathrm{KL}\big(q_\phi(\boldsymbol{X})||p(\boldsymbol{X})\big)$ between two hyperbolic wrapped normal distributions can easily be evaluated via Monte-Carlo sampling (see App. C.1 for details). Moreover, the expectation $\mathbb{E}_{q_\phi(\boldsymbol{X})}[\log p(\boldsymbol{Y}|\boldsymbol{X})]$ can be decomposed into individual terms for each observation dimension as $\sum_{d=1}^{D} \mathbb{E}_{q_\phi(\boldsymbol{X})}[\log p(\boldsymbol{y}_d|\boldsymbol{X})]$, where $\boldsymbol{y}_d$ is the $d$-th column of $\boldsymbol{Y}$. For large datasets, each term can be evaluated via a variational sparse GP approximation (Titsias, 2009; Hensman et al., 2015). To do so, we introduce $M$ inducing inputs $\{\boldsymbol{z}_{d,m}\}_{m=1}^{M}$ with $\boldsymbol{z}_{d,m} \in \mathbb{H}^Q$ for each observation dimension $d$, whose corresponding inducing variables $\{u_{d,m}\}_{m=1}^{M}$ are defined as noiseless observations of the GP in equation 3, i.e, $u_d \sim \mathrm{GP}(m_d(\boldsymbol{z}_d), k_d^{\mathbb{H}^Q}(\boldsymbol{z}_d, \boldsymbol{z}_d))$. Similar to (Hensman et al., 2015), we can write,

$$\log p(\boldsymbol{y}_d|\boldsymbol{X}) \geq \mathbb{E}_{q_\lambda(\boldsymbol{f}_d)}\left[\log \mathcal{N}(\boldsymbol{y}_d; \boldsymbol{f}_d(\boldsymbol{X}), \sigma_d^2)\right] - \mathrm{KL}\big(q_\lambda(\boldsymbol{u}_d)||p(\boldsymbol{u}_d|\boldsymbol{Z}_d)\big), \tag{8}$$

where we defined $q_\lambda(\boldsymbol{f}_d) = \int p(\boldsymbol{f}_d|\boldsymbol{u}_d)q_\lambda(\boldsymbol{u}_d)d\boldsymbol{u}_d$ with the variational distribution $q_\lambda(\boldsymbol{u}_d) = \mathcal{N}(\boldsymbol{u}_d; \tilde{\boldsymbol{\mu}}_d, \tilde{\boldsymbol{\Sigma}}_d)$, and variational parameters $\lambda = \{\tilde{\boldsymbol{\mu}}_d, \tilde{\boldsymbol{\Sigma}}_d\}_{d=1}^{D}$. Remember that the inducing variables $u_{d,m}$ are Euclidean, i.e., the variational distribution $q_\lambda(\boldsymbol{u}_d)$ is a Euclidean Gaussian and the KL divergence in equation 8 has a closed-form solution. In this case, the training parameters of the GPHLVM are the set of inducing inputs $\{\boldsymbol{z}_{d,m}\}_{m=1}^{M}$, the variational parameters $\phi$ and $\lambda$, and the hyperparameters $\boldsymbol{\Theta}$ (see App. C.2 for the derivation of the GPHLVM variational inference process).

**Optimization:** As several training parameters of the GPHLVM belong to $\mathbb{H}^Q$, i.e., the latent variables $\boldsymbol{x}_n$ for the MAP estimation, or the inducing inputs $\boldsymbol{z}_{d,m}$ and means $\boldsymbol{\mu}_n$ for variational inference. To account for the hyperbolic geometry of these parameters, we leverage Riemannian optimization methods (Absil et al., 2007; Boumal, 2022) to train the GPHLVM. Each step of first order (stochastic) Riemannian optimization methods is generally of the form,

$$\boldsymbol{\eta}_t \leftarrow h\big(\mathrm{grad}\,\mathcal{L}(\boldsymbol{x}_t), \boldsymbol{\tau}_{t-1}\big), \qquad \boldsymbol{x}_{t+1} \leftarrow \mathrm{Exp}_{\boldsymbol{x}_t}(-\alpha_t \boldsymbol{\eta}_t), \qquad \boldsymbol{\tau}_t \leftarrow \mathrm{P}_{\boldsymbol{x}_t \rightarrow \boldsymbol{x}_{t+1}}\big(\boldsymbol{\eta}_t\big). \tag{9}$$

The update $\boldsymbol{\eta}_t \in \mathcal{T}_{\boldsymbol{x}_t}\mathcal{M}$ is first computed as a function $h$ of the Riemannian gradient $\mathrm{grad}$ of the loss $\mathcal{L}(\boldsymbol{x}_t)$ and of $\boldsymbol{\tau}_{t-1}$, the previous update that is parallel-transported to the tangent space of the new estimate $\boldsymbol{x}_t$. The estimate $\boldsymbol{x}_t$ is then updated by projecting the update $\boldsymbol{\eta}_t$ scaled by a learning rate $\alpha_t$ onto the manifold using the exponential map. The function $h$ is equivalent to computing the update of the Euclidean algorithm, e.g., $\boldsymbol{\eta}_t \leftarrow \mathrm{grad}\,\mathcal{L}(\boldsymbol{x}_t)$ for a simple gradient descent. Notice that equation 9 is applied on a product of manifolds when optimizing several parameters. In this paper, we used the Riemannian Adam (Bécigneul & Ganea, 2019) implemented in Geoopt (Kochurov et al., 2020) to optimize the GPHLVM parameters.

## 4 INCORPORATING TAXONOMY KNOWLEDGE INTO GPHLVM

While we are now able to learn hyperbolic embeddings of the data associated to a taxonomy using our GPHLVM, they do not necessarily follow the graph structure of the taxonomy. In other words, the manifold distances between pairs of embeddings do not necessarily match the graph distances. To overcome this, we introduce graph-distance information as inductive bias to learn the embeddings. To do so, we leverage two well-known techniques in the GPLVM literature: priors on the embeddings and back constraints (Lawrence & Quiñonero Candela, 2006; Urtasun et al., 2008). Both are reformulated to preserve the taxonomy graph structure in the hyperbolic latent space as a function of the node-to-node shortest paths.

**Graph-distance priors:** As shown by Urtasun et al. (2008), the structure of the latent space can be modified by adding priors of the form $p(\boldsymbol{X}) \propto e^{-\phi(\boldsymbol{X})/\sigma_\phi^2}$ to the GPLVM, where $\phi(\boldsymbol{X})$ is a function that we aim at minimizing. Incorporating such a prior may also be alternatively understood

as augmenting the GPLVM loss $\mathcal{L}$ with a regularization term $-\phi(\boldsymbol{X})$. Therefore, we propose to augment the loss of the GPHLVM with a distance-preserving graph-based regularizer. Several such losses have been proposed in the literature, see (Cruceru et al., 2021) for a review. Specifically, we define $\phi(\boldsymbol{X})$ as the stress loss,

$$\mathcal{L}_{\text{stress}}(\boldsymbol{X}) = \sum_{i<j} \left( \text{dist}_{\mathbb{G}}(c_i, c_j) - \text{dist}_{\mathbb{H}^Q}(\boldsymbol{x}_i, \boldsymbol{x}_j) \right)^2, \tag{10}$$

where $c_i$ denotes the taxonomy node to which the observation $\boldsymbol{y}_i$ belongs, and $\text{dist}_{\mathbb{G}}$, $\text{dist}_{\mathbb{H}^Q}$ are the taxonomy graph distance and the geodesic distance on $\mathbb{H}^Q$, respectively. The loss equation 10 encourages the preservation of all distances of the taxonomy graph in $\mathbb{H}^Q$. It therefore acts *globally*, thus allowing the complete taxonomy structure to be reflected by the GPHLVM. Notice that Cruceru et al. (2021) also survey a distortion loss that encourages the distance of the embeddings to match the graph distance by considering their ratio. We notice, however, that this distortion loss is only properly defined when the embeddings $\boldsymbol{x}_i$ and $\boldsymbol{x}_j$ correspond to different classes $c_i \neq c_j$. Interestingly, our empirical results using this loss were lackluster and numerically unstable (see App. E).

**Back-constraints:** The back-constrained GPLVM (Lawrence & Quiñonero Candela, 2006) defines the latent variables as a function of the observations, i.e., $x_{n,q} = g_q(\boldsymbol{y}_1 \ldots, \boldsymbol{y}_n; \boldsymbol{w}_q)$ with parameters $\{\boldsymbol{w}_q\}_{q=1}^Q$. This allows us to incorporate new observations in the latent space after training, while preserving local similarities between observations in the embeddings. To incorporate graph-distance information into the GPHLVM and ensure that latent variables lie on the hyperbolic manifold, we propose the back-constraints mapping,

$$\boldsymbol{x}_n = \text{Exp}_{\boldsymbol{\mu}_0}(\tilde{\boldsymbol{x}}_n) \quad \text{with} \quad \tilde{x}_{n,q} = \sum_{m=1}^N w_{q,m} k^{\mathbb{R}^D}(\boldsymbol{y}_n, \boldsymbol{y}_m) k^{\mathbb{G}}(c_n, c_m). \tag{11}$$

The mapping equation 11 not only expresses the similarities between data in the observation space via the kernel $k^{\mathbb{R}^J}$, but encodes the relationships between data belonging to nearby taxonomy nodes via $k^{\mathbb{G}}$. In other words, similar observations associated to the same (or near) taxonomy nodes will be close to each other in the resulting latent space. The kernel $k^{\mathbb{G}}$ is a Matérn kernel on the taxonomy graph following the formulation introduced in (Borovitskiy et al., 2021), which accounts for the graph geometry (see also App. D). We also use a Euclidean SE kernel for $k^{\mathbb{R}^D}$. Notice that the back constraints only incorporate *local* information into the latent embedding. Therefore, to preserve the *global* graph structure, we pair the proposed back-constrained GPHLVM with the stress prior equation 10. Note that both kernels are required in equation 11: By defining the mapping as a function of the graph kernel only, the observations of each taxonomy node would be encoded by a single latent point. When using the observation kernel only, dissimilar observations of the same taxonomy node would be distant in the latent space, despite the additional stress prior, as $k^{\mathbb{R}^D}(\boldsymbol{y}_n, \boldsymbol{y}_m) \approx 0$.

## 5 EXPERIMENTS

We test the proposed GPHLVM on three distinct robotics taxonomies. First, we model data from the whole-body support pose taxonomy (Borràs et al., 2017). Each node of this taxonomy graph (see Fig. 1-*right*) is a support pose defined by its contacts, so that the distance between nodes can be viewed as the number of contact changes required to go from a support pose to another. We use standing and kneeling poses of the datasets in (Mandery et al., 2016a) and (Langenstein, 2020). The former were extracted from recordings of a human walking without hand support, or using supports from a handrail or from a table on one side or on both sides. The latter were obtained from a human standing up from a kneeling position. Each pose is identified with a node of the graph of Fig. 1-*right*. We test our approach on an unbalanced dataset of 100 poses composed of 72 standing and 28 kneeling poses, where each pose is represented by a vector of joint angles $\boldsymbol{y}_n \in \mathbb{R}^{44}$. Note that we augment the taxonomy to explicitly distinguish between left and right contacts. Second, we consider a hand grasp taxonomy Stival et al. (2019) that organizes common grasp types in a tree structure based on their muscular and kinematic properties. We use 94 grasps of 19 types obtained from recordings of humans grasping different objects. Each grasp is encoded by a vector of wrist and finger joint angles $\boldsymbol{y}_n \in \mathbb{R}^{24}$. Third, we model the data from the bimanual manipulation taxonomy (Krebs & Asfour, 2022), where each node of the taxonomy tree represents a coordination pattern of human bimanual manipulation skills. We use a balanced dataset of 60 whole-body poses extracted from recordings of bimanual household activities, as in (Krebs & Asfour, 2022). Each pose

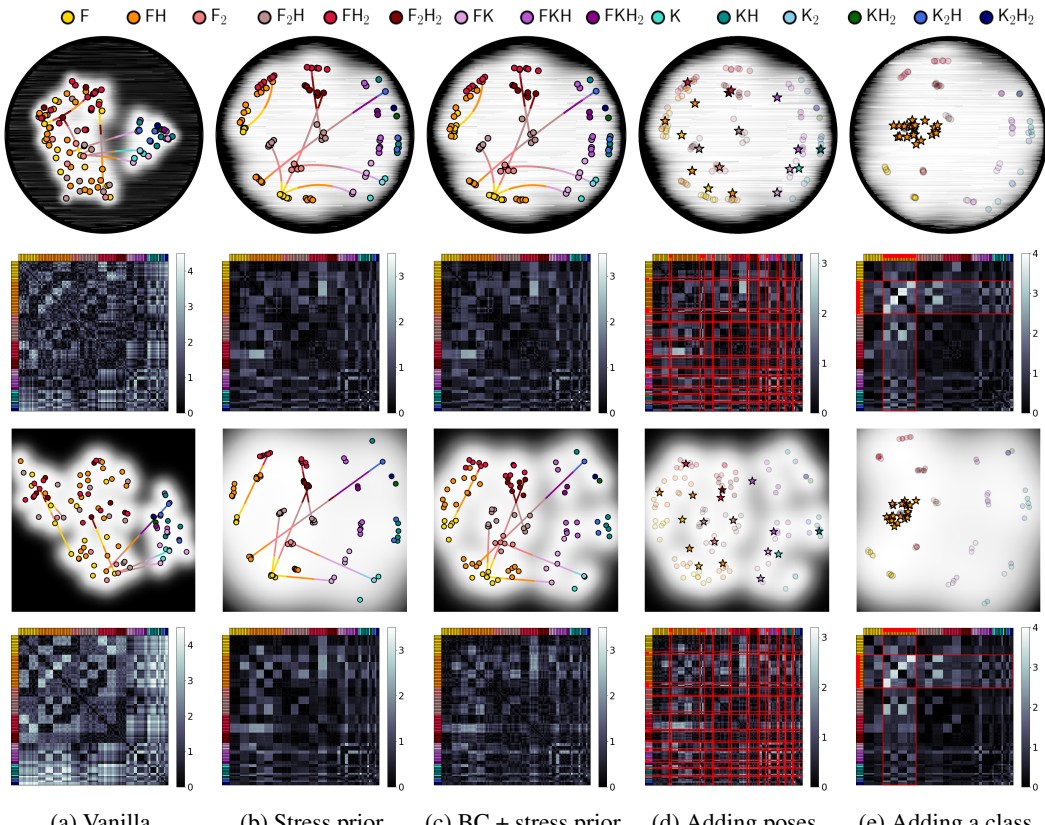

Figure 2: Support poses: The first and last two rows show the latent embeddings and examples of interpolating geodesics in $\mathcal{P}^2$ and $\mathbb{R}^2$, followed by pairwise error matrices between geodesic and taxonomy graph distances. Embeddings colors match those of Fig. 1-*right*, and background colors indicate the GPLVM uncertainty. Added poses *(d)* and classes $\mathsf{FH} = \mathsf{F}^{\{l,r\}}\mathsf{H}^{\{l,r\}}$ *(e)* are marked with stars and highlighted with red in the error matrices.

Table 1: Average stress per geometry and regularization.

| Taxonomy | Model | No regularizer | Stress | BC + Stress | Unseen poses | Unseen class |
|---|---|---|---|---|---|---|
| Support poses | GPLVM, $\mathbb{R}^2$ | $3.93 \pm 3.97$ | $0.58 \pm 0.94$ | $0.63 \pm 0.94$ | $0.66 \pm 0.99$ | $2.37 \pm 3.03$ |
| | GPHLVM, $\mathbb{H}^2$ | $\mathbf{2.05 \pm 2.50}$ | $\mathbf{0.51 \pm 0.82}$ | $\mathbf{0.53 \pm 0.83}$ | $\mathbf{0.56 \pm 0.86}$ | $\mathbf{2.27 \pm 2.88}$ |
| | GPLVM, $\mathbb{R}^3$ | $\mathbf{3.76 \pm 3.74}$ | $\mathbf{0.24 \pm 0.40}$ | $\mathbf{0.29 \pm 0.39}$ | $\mathbf{0.30 \pm 0.43}$ | $2.06 \pm 2.70$ |
| | GPHLVM, $\mathbb{H}^3$ | $3.78 \pm 3.71$ | $0.30 \pm 0.38$ | $0.35 \pm 0.45$ | $0.37 \pm 0.50$ | $\mathbf{2.02 \pm 2.63}$ |
| Grasps | GPLVM, $\mathbb{R}^2$ | $7.25 \pm 5.40$ | $0.39 \pm 0.41$ | $0.40 \pm 0.44$ | $0.53 \pm 0.77$ | $2.30 \pm 3.13$ |
| | GPHLVM, $\mathbb{H}^2$ | $\mathbf{5.47 \pm 4.07}$ | $\mathbf{0.14 \pm 0.16}$ | $\mathbf{0.18 \pm 0.29}$ | $\mathbf{0.35 \pm 0.78}$ | $\mathbf{2.11 \pm 3.01}$ |
| | GPLVM, $\mathbb{R}^3$ | $\mathbf{8.15 \pm 5.85}$ | $0.14 \pm 0.18$ | $0.15 \pm 0.19$ | $0.29 \pm 0.64$ | $2.02 \pm 2.91$ |
| | GPHLVM, $\mathbb{H}^3$ | $8.37 \pm 5.71$ | $\mathbf{0.04 \pm 0.08}$ | $\mathbf{0.07 \pm 0.18}$ | $\mathbf{0.23 \pm 0.68}$ | $\mathbf{1.92 \pm 2.86}$ |
| Bimanual manipulation categories | GPLVM, $\mathbb{R}^2$ | $2.03 \pm 2.15$ | $0.13 \pm 0.33$ | $0.15 \pm 0.31$ | $0.15 \pm 0.29$ | $1.65 \pm 2.17$ |
| | GPHLVM, $\mathbb{H}^2$ | $\mathbf{0.98 \pm 1.26}$ | $\mathbf{0.11 \pm 0.33}$ | $\mathbf{0.09 \pm 0.12}$ | $\mathbf{0.09 \pm 0.11}$ | $\mathbf{1.54 \pm 2.07}$ |
| | GPLVM, $\mathbb{R}^3$ | $2.39 \pm 2.36$ | $\mathbf{0.01 \pm 0.01}$ | $0.20 \pm 0.38$ | $0.20 \pm 0.38$ | $1.62 \pm 2.13$ |
| | GPHLVM, $\mathbb{H}^3$ | $\mathbf{1.18 \pm 1.35}$ | $\mathbf{0.01 \pm 0.03}$ | $\mathbf{0.04 \pm 0.08}$ | $\mathbf{0.03 \pm 0.07}$ | $\mathbf{1.58 \pm 2.10}$ |

is a vector of joint angles $\boldsymbol{y}_n \in \mathbb{R}^{86}$. The main results are analyzed in the sequel, while additional experimental details, results, computational analyses, and comparisons are given in App. F and G.

**Hyperbolic embeddings of taxonomy data:** We embed the taxonomy data of the aforementioned taxonomies into 2-dimensional hyperbolic and Euclidean spaces using GPHLVM and GPLVM. For each, we test the model without regularization, with stress prior, and with back-constraints coupled with stress prior (see App. F.2 for the training parameters). Concerning the support pose taxonomy, Figs. 2a-2c show the learned embeddings alongside error matrices depicting the difference between geodesic and taxonomy graph distances. As shown in Fig. 2a, the models without regularization do not encode any meaningful distance structure in latent space. In contrast, the models with stress prior result in embeddings that comply with the taxonomy graph structure: The embeddings are grouped and organized according to the taxonomy nodes, the geodesic distances match the graph ones, and arguably more so in the hyperbolic case (see error matrices in Figs. 2b-2c). Similar insights can

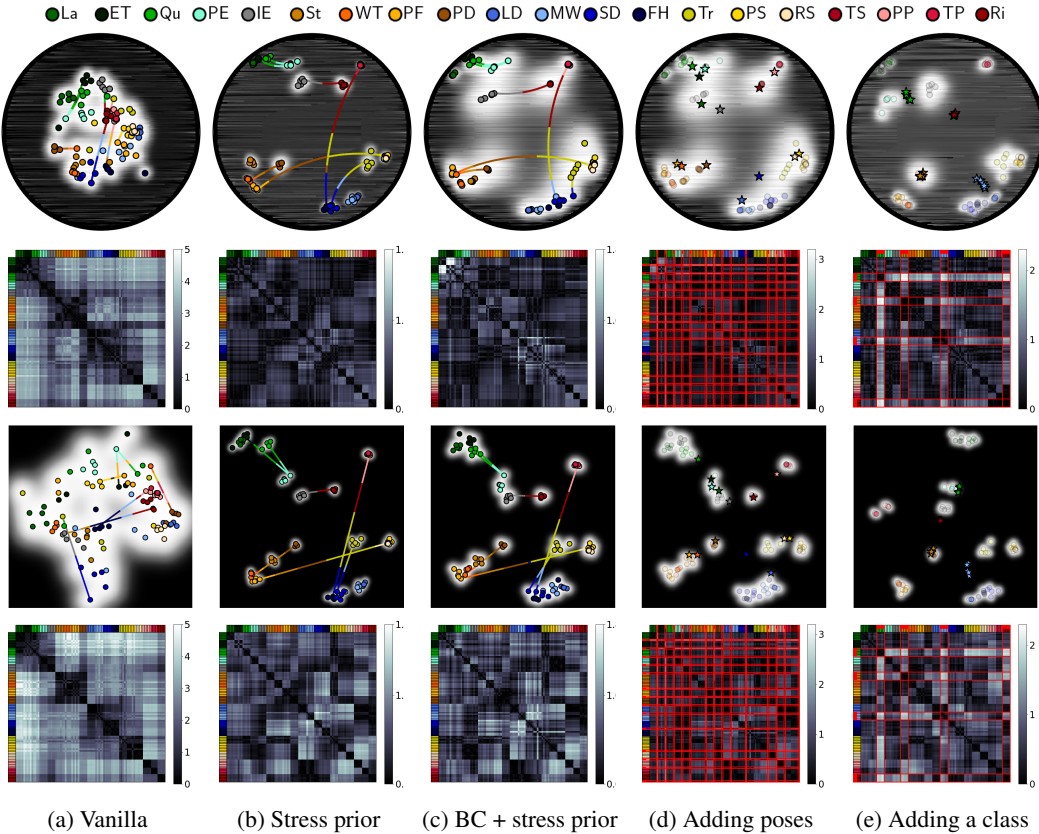

(a) Vanilla      (b) Stress prior      (c) BC + stress prior      (d) Adding poses      (e) Adding a class

Figure 3: Grasps: The first and last two rows show the latent embeddings and examples of interpolating geodesics in $\mathcal{P}^2$ and $\mathbb{R}^2$, followed by pairwise error matrices between geodesic and graph distances. Embeddings colors match those of Fig. 4, and background colors indicate the GPLVM uncertainty. Added poses *(d)* and classes Qu, St, MW, and Ri *(e)* are marked with stars and highlighted with red in the error matrices.

be drawn from the embeddings learned for the hand grasps taxonomy (Figs. 3a- 3c), and for the bimanual manipulation taxonomy, (Figs. 7a- 7c). Note that augmenting the support pose taxonomy leads to several groups of the same support pose in Figs. 2b-2c, e.g., F splits into $F^l$ and $F^r$.

A quantitative comparison of the stress values of the latent embeddings with respect to the graph distances confirms that a hyperbolic geometry captures better the data structure (see Table 1). Interestingly, the hyperbolic models of the hand grasps and bimanual manipulation taxonomies also outperform Euclidean models with 3-dimensional latent spaces (see models and discussion in App. F.4). This is due to the fact that the geometry of the hyperbolic manifold leads to exponentially-increasing distances w.r.t the origin, which provides an increased volume to match the graph structure when compared to Euclidean spaces, thus resulting in better low-dimensional representations of taxonomy data. Importantly, a comparative study reported in App. G.1 showed that GPHLVM also outperformed vanilla and hyperbolic versions of a VAE to encode meaningful taxonomy information in the latent space. In general, the tested VAEs only captured a global structure of the taxonomies data. Also, the average stress of the VAEs' latent embeddings and their reconstruction error is higher compared to the GPHLVM's (see Table 10). Notice that the GPHLVM back constraints further organize the embeddings inside a class according to the similarity between their observations (see Figs. 2c, 3c, 7c). Finally, we also tested a GPLVM for learning a Riemannian manifold (Tosi et al., 2014) of the taxonomy data, reported in App. G.2, which is unable to capture the local and global data structure as this model was not designed for hierarchical discrete data.

**Taxonomy expansion and unseen poses encoding:** An advantage of back-constrained GPLVMs is their affordance to "embed" new observations into the latent space. We test the GPHLVM ability to place unseen class instances or unobserved taxonomy classes into the latent space, hypothesizing that their respective embeddings should be placed at meaningful distances w.r.t. the rest of the latent points. First, we consider back-constrained GPHLVMs with stress prior previously trained on a subset of the taxonomies data (i.e., the models in Figs. 2c, 3c, 7c) and embedded unseen class instances. Figs. 2d, 3d and 7d show how the new data land close to their respective class cluster. Second, we train new GPHLVMs for the three taxonomies while withholding all data instance from one or several classes (see App. F.2). We then encode these data and find that they are located

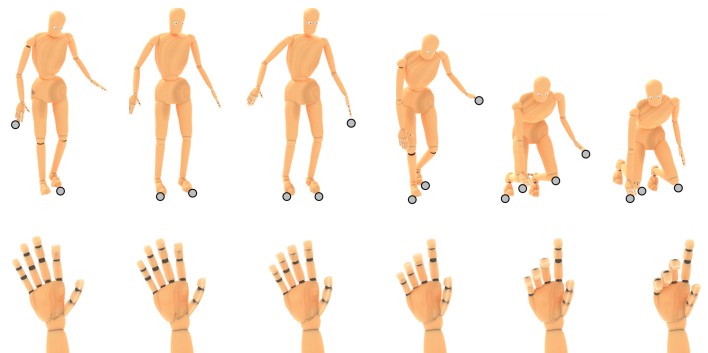

Figure 4: Motions obtained via geodesic interpolation in the back-constrained GPHLVM latent space. *Top*: Support pose taxonomy from $F^lH^r$ to $K_2H^r$. Contacts are denoted by gray circles. *Bottom*: Grasp taxonomy from ring (Ri) to index finger extension (IE).

at sensible distances when compared to the model trained on the full dataset. Although this is accomplished by both models, our GPHLVMs display lower stress values (see Table 1).

**Trajectory generation via geodesics:** The geometry of the GPHLVM latent space can also be exploited to generate trajectories in the latent space by following the geodesic, i.e., the shortest path, between two embeddings. In other words, our GPHLVM intrinsically provides a mechanism to plan motions via geodesics in the low-dimensional latent space. Examples of geodesics between two embeddings for the three taxonomies are shown in Figs. 2b-2c, 3b- 3c, and 7b- 7c, with the colors along the trajectory matching the class corresponding to the closest hyperbolic latent point. Importantly, the geodesics in GPHLVMs latent space follow the transitions between classes defined in the taxonomy. In other words, the shortest paths in the hyperbolic embedding correspond to the shortest paths in the taxonomy graph. For instance, in the case of the support pose taxonomy, the geodesic from $F^l$ to $F_2H^r$ follows $F^l \rightarrow F_2 \rightarrow F_2H^r$. Straight lines in the Euclidean embeddings are more likely to deviate from the graph shortest path, resulting in transitions that do not exist in the taxonomy, e.g., $F^rH^r \rightarrow F_2$, or $F^rH^r \rightarrow F^rH^l$ in the Euclidean latent space of Figs. 2b-2c. Fig. 4 and App. F.6 show motions resulting from geodesic interpolation in the GPHLVM latent space. The obtained motions are more realistic than those obtained via linear interpolation in the GPLVM latent space and as realistic as those obtained via VPoser (Pavlakos et al., 2019) (see Figs. 11- 15).

## 6 CONCLUSIONS

Inspired by the recent developments of taxonomies in different robotics fields, we proposed a computational model GPHLVM that leveraged two types of domain knowledge: the structure of a human-designed taxonomy and a hyperbolic geometry on the latent space which complies with the intrinsic taxonomy's hierarchical structure. Our GPHLVM allows us to learn hyperbolic embeddings of the features of the taxonomy nodes while capturing the associated hierarchical structure. To achieve this, our model exploited the curvature of the hyperbolic manifold and the graph-distance information, as inductive bias. We showed that these two forms of inductive bias are essential to: learn taxonomy-aware embeddings, encode unseen data, and potentially expand the learned taxonomy. Moreover, we reported that vanilla Euclidean approaches underperformed on all the foregoing cases. Finally, we introduced a mechanism to generate taxonomy-aware motions in the hyperbolic latent space.

Incorporating the taxonomy structure enables the use of the learned hyperbolic embeddings in various downstream robotics tasks, such as generation of hard-to-collect data (e.g., transitions between grasps types), taxonomy-aware whole-body motion generation, human motion prediction, or human poses classification. Unlike other LVMs such as VPoser (Pavlakos et al., 2019), GAN-S (Davydov et al., 2022), and TEACH (Athanasiou et al., 2022), which are trained on full human motion trajectories and thousands of datapoints, our model focuses on leveraging the taxonomies as inductive bias to better structure the learned embeddings, and it uses geodesics as a simple and effective motion generator between single poses. However, as other models, our geodesic motion generation does not use explicit knowledge on how physically feasible the generated trajectories are. We plan to investigate how to include physics constraints or explicit contact data into the GPHLVM to obtain physically-feasible motions that can be executed on real robots. We will also work on alleviating the computational cost of the hyperbolic kernel in $\mathbb{H}^d$. This could be tackled by using a different sampling strategy: Instead of sampling from a Gaussian distribution for the approximation of equation 5, we could sample from the Rayleigh distribution. This is because complex numbers, whose real and imaginary components are i.i.d. Gaussian, have absolute value that is Rayleigh-distributed. Finally, we plan to investigate other types of manifold geometries that may accommodate more complex structures coming from highly-heterogeneous graphs (Giovanni et al., 2022).

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

## A  HYPERBOLIC MANIFOLD

### A.1  EQUIVALENCE OF POINCARÉ AND LORENTZ MODELS

As pointed out in the main text (§ 2), it is possible to map points from the Lorentz model to the Poincaré ball via an isometric mapping. Formally, such an isometry is defined as the mapping function $f : \mathcal{L}^d \to \mathcal{P}^d$ such that

$$f(\boldsymbol{x}) = \frac{(x_1, \ldots, x_d)^{\mathsf{T}}}{x_0 + 1}, \tag{12}$$

where $\boldsymbol{x} \in \mathcal{L}^d$ with components $x_0, x_1, \ldots, x_d$. The inverse mapping $f^{-1} : \mathcal{P}^d \to \mathcal{L}^d$ is defined as follows

$$f^{-1}(\boldsymbol{y}) = \frac{\left(1 + \|\boldsymbol{y}\|^2, 2y_1, \ldots, 2y_d\right)^{\mathsf{T}}}{1 - \|\boldsymbol{y}^2\|}, \tag{13}$$

with $\boldsymbol{y} \in \mathcal{P}^d$ with components $y_1, \ldots, y_d$. Notice that we used the mapping equation 12 to represent the hyperbolic embeddings in the Poincaré disk throughout the paper, as well as in the computation of the kernel $k^{\mathbb{H}^2}$ equation 4.

### A.2  MANIFOLD OPERATIONS

As mentioned in the main text (§ 2), we resort to the exponential and logarithmic maps to operate with Riemannian manifold data. The exponential map $\mathrm{Exp}_{\boldsymbol{x}}(\boldsymbol{u}) : \mathcal{T}_{\boldsymbol{x}}\mathcal{M} \to \mathcal{M}$ maps a point $\boldsymbol{u}$ in the tangent space of $\boldsymbol{x}$ to a point $\boldsymbol{y}$ on the manifold, while the logarithmic map $\mathrm{Log}_{\boldsymbol{x}}(\boldsymbol{u}) : \mathcal{M} \to \mathcal{T}_{\boldsymbol{x}}\mathcal{M}$ performs the corresponding inverse operation. In some settings, it is necessary to work with data lying on different tangent spaces of the manifold. In this case, one needs to operate with all data on a single tangent space, which can be achieved by leveraging the parallel transport $\mathrm{P}_{\boldsymbol{x} \to \boldsymbol{y}}(\boldsymbol{u}) : \mathcal{T}_{\boldsymbol{x}}\mathcal{M} \to \mathcal{T}_{\boldsymbol{y}}\mathcal{M}$. All the aforementioned operators are defined in Table 2 for the Lorentz model $\mathcal{L}^d$. Moreover, we introduce the inner product $\langle \boldsymbol{u}, \boldsymbol{v} \rangle_{\boldsymbol{x}}$ between two points on $\mathcal{L}^d$, which is used to compute the geodesic distance $d_{\mathcal{M}}(\boldsymbol{u}, \boldsymbol{v})$ and all the foregoing operations in the Lorentz model, as shown in Table 2.

| Operation | Formula |
|---|---|
| $\langle \boldsymbol{u}, \boldsymbol{v} \rangle_{\boldsymbol{x}}$ | $-u_0 v_0 + \sum_{i=1}^d u_i v_i$ |
| $d_{\mathcal{M}}(\boldsymbol{u}, \boldsymbol{v})$ | $\mathrm{arcosh}(-\langle \boldsymbol{u}, \boldsymbol{v} \rangle_{\boldsymbol{x}})$ |
| $\mathrm{Exp}_{\boldsymbol{x}}(\boldsymbol{u})$ | $\cosh(\|\boldsymbol{u}\|_{\mathcal{L}})\boldsymbol{x} + \sinh(\|\boldsymbol{u}\|_{\mathcal{L}})\frac{\boldsymbol{u}}{\|\boldsymbol{u}\|_{\mathcal{L}}}$ with $\|\boldsymbol{u}\|_{\mathcal{L}} = \sqrt{\langle \boldsymbol{u}, \boldsymbol{u} \rangle_{\boldsymbol{x}}}$ |
| $\mathrm{Log}_{\boldsymbol{x}}(\boldsymbol{y})$ | $\frac{d_{\mathcal{M}}(\boldsymbol{x}, \boldsymbol{y})}{\sqrt{\alpha^2 - 1}}(\boldsymbol{y} + \alpha \boldsymbol{x})$ with $\alpha = \langle \boldsymbol{x}, \boldsymbol{y} \rangle_{\boldsymbol{x}}$ |
| $\mathrm{P}_{\boldsymbol{x} \to \boldsymbol{y}}(\boldsymbol{v})$ | $\boldsymbol{v} + \frac{\langle \boldsymbol{y}, \boldsymbol{v} \rangle_{\boldsymbol{x}}}{1 - \langle \boldsymbol{x}, \boldsymbol{y} \rangle_{\boldsymbol{x}}}(\boldsymbol{x} + \boldsymbol{y})$ |

Table 2: Principal operations on $\mathbb{H}^d$ for the Lorentz model. For more details, see (Bose et al., 2020) and (Peng et al., 2021).

## B  HYPERBOLIC KERNELS

As mentioned in the main text (§ 3.1), following the developments on kernels on manifolds like Borovitskiy et al. (2020); Jaquier et al. (2021), we may identify the generalized squared exponential kernel with the *heat kernel* — an important object studied on its own in the mathematical literature. Due to this, we can obtain the expressions equation 4. The expression for the case of $\mathbb{H}^2$ requires discretizing the integral, which may lead to an approximation that is not positive semidefinite. We address this by suggesting another approximation guaranteed to be positive semidefinite.

Reversing the derivation in (Chavel, 1984, p. 246), we obtain

$$k^{\mathbb{H}^2}_{\infty, \kappa, \sigma^2}(\boldsymbol{x}, \boldsymbol{x}') = \frac{\sigma^2}{C'_{\infty}} \int_0^{\infty} \exp(-s^2/(2\kappa^2)) P_{-1/2 + is}(\cosh(\rho)) s \tanh(\pi s) \mathrm{d}s, \tag{14}$$

where $\rho = \mathrm{dist}_{\mathbb{H}^d}(\boldsymbol{x}, \boldsymbol{x}')$ denotes the geodesic distance between $\boldsymbol{x}, \boldsymbol{x}' \in \mathbb{H}^2$, $\kappa$ and $\sigma^2$ are the kernel lengthscale and variance, $C'_\infty$ is a normalizing constant and $P_\alpha$ are Legendre functions Abramowitz & Stegun (1964). Now we prove that these Legendre functions are connected to the *spherical functions* — special functions closely tied to the geometry of the hyperbolic space and possessing a very important property.

**Proposition.** *Assume the disk model of $\mathbb{H}^2$ (i.e. the Poincaré disk). Denote the disk by $\mathbb{D}$ and its boundary, the circle, by $\mathbb{T}$. Define the hyperbolic outer product by $\langle \boldsymbol{z}, \boldsymbol{b} \rangle = \frac{1}{2} \log \frac{1 - |\boldsymbol{z}|^2}{|\boldsymbol{z} - \boldsymbol{b}|^2}$ for $\boldsymbol{z} \in \mathbb{D}, \boldsymbol{b} \in \mathbb{T}$. Then*

$$P_{-1/2+is}(\cosh(\rho)) = \underbrace{\int_{\mathbb{T}} e^{(2si+1)\langle \boldsymbol{z}, \boldsymbol{b} \rangle} \mathrm{d}b}_{\text{spherical function } \phi_{2s}(\boldsymbol{z})} = \int_{\mathbb{T}} e^{(2si+1)\langle \boldsymbol{z}_1, \boldsymbol{b} \rangle} \overline{e^{(2si+1)\langle \boldsymbol{z}_2, \boldsymbol{b} \rangle}} \mathrm{d}\boldsymbol{b}, \qquad (15)$$

*where $\boldsymbol{z} \in \mathbb{D}$ is such that $\rho = \mathrm{dist}_{\mathbb{H}^2}(\boldsymbol{z}, \boldsymbol{0})$ and $\boldsymbol{z}_1, \boldsymbol{z}_2 \in \mathbb{D}$ are such that $\rho = \mathrm{dist}_{\mathbb{H}^2}(\boldsymbol{z}_1, \boldsymbol{z}_2)$. Here $i$ denotes the imaginary unit and $\overline{z}$ is the complex conjugation.*

*Proof.* Let $\theta$ denote the angle between $\boldsymbol{z}$ and $\boldsymbol{b}$, and note the following simple identities

$$|\boldsymbol{z} - \boldsymbol{b}|^2 = |\boldsymbol{z}|^2 + 1 - 2|\boldsymbol{z}| \cos(\theta) = \tanh(\rho)^2 + 1 - 2\tanh(\rho)\cos(\theta), \qquad (16)$$

$$1 - |\boldsymbol{z}|^2 = 1 - \tanh(\rho)^2 = \cosh(\rho)^{-2}. \qquad (17)$$

Then, we write

$$e^{(2si+1)\langle \boldsymbol{z}, \boldsymbol{b} \rangle} = \left( \frac{|\boldsymbol{z} - \boldsymbol{b}|^2}{1 - |\boldsymbol{z}|^2} \right)^{-si-1/2} = \left( \cosh(\rho)^2 (\tanh(\rho)^2 + 1 - 2\tanh(\rho)\cos(\theta)) \right)^{-si-1/2}, \qquad (18)$$

$$= \left( \sinh(\rho)^2 + \cosh(\rho)^2 - 2\sinh(\rho)\cosh(\rho)\cos(\theta) \right)^{-si-1/2}, \qquad (19)$$

$$= \left( \cosh(2\rho) + \sinh(2\rho)\cos(\theta) \right)^{-si-1/2}. \qquad (20)$$

On the other hand, by (Lebedev et al., 1965, Eq. 7.4.3), we have $P_a(\cosh(x)) = \frac{1}{\pi} \int_0^\pi (\cosh(x) + \sinh(x)\cos(\theta))^a \mathrm{d}\theta$, hence

$$P_{-1/2+is}(\cosh(2\rho)) = \frac{1}{\pi} \int_0^\pi (\cosh(2\rho) + \sinh(2\rho)\cos(\theta))^{-1/2+is} \mathrm{d}\theta, \qquad (21)$$

$$= \frac{1}{2\pi} \int_{-\pi}^\pi (\cosh(2\rho) + \sinh(2\rho)\cos(\theta))^{-1/2+is} \mathrm{d}\theta, \qquad (22)$$

$$= \int_{\mathbb{T}} e^{(-2si+1)\langle \boldsymbol{z}, \boldsymbol{b} \rangle} \mathrm{d}\boldsymbol{b} = \phi_{-2s}(\boldsymbol{z}). \qquad (23)$$

This computation roughly follows Cohen & Lifshits (2012, Section 4.3.4). Now, by Cohen & Lifshits (2012, Section 3.5), we have $\phi_{-2s}(\boldsymbol{z}) = \phi_{2s}(\boldsymbol{z})$ which proves the first identity. Finally, Lemma 3.5 from Cohen & Lifshits (2012) proves the second identity. $\square$

By combining expressions equation 14 and equation 15, we get the following Monte Carlo approximation

$$k^{\mathbb{H}^2}_{\infty, \kappa, \sigma^2}(\boldsymbol{x}, \boldsymbol{x}') \approx \frac{\sigma^2}{C'_\infty} \frac{1}{L} \sum_{l=1}^{L} s_l \tanh(\pi s_l) e^{(2s_l i + 1)\langle \boldsymbol{x}_{\mathcal{P}}, \boldsymbol{b}_l \rangle} \overline{e^{(2s_l i + 1)\langle \boldsymbol{x}'_{\mathcal{P}}, \boldsymbol{b}_l \rangle}}, \qquad (24)$$

where $\boldsymbol{b}_l \stackrel{\text{i.i.d.}}{\sim} U(\mathbb{T})$ and $s_l \stackrel{\text{i.i.d.}}{\sim} e^{-s^2 \kappa^2/2} \mathbb{1}_{[0,\infty)}(s)$. This gives the approximation used in the main text (see § 3.1).

Having established a way to evaluate or approximate the heat kernel, analogs of Matérn kernels can be defined by

$$k_{\nu, \kappa, \sigma^2}(\boldsymbol{x}, \boldsymbol{x}') = \frac{\sigma^2}{C_\nu} \int_0^\infty u^{\nu-1} e^{-\frac{2\nu}{\kappa^2} u} \tilde{k}_{\infty, \sqrt{2u}, \sigma^2}(\boldsymbol{x}, \boldsymbol{x}') \mathrm{d}u, \qquad (25)$$

where $\tilde{k}_{\infty, \sqrt{2u}, \sigma^2}$ is the same as $k_{\infty, \sqrt{2u}, \sigma^2}$ but with the normalizing constant $\sigma^2/C_\infty$ dropped for simplicity. Here $C_\nu$ is the normalizing constant ensuring that $k_{\nu, \kappa, \sigma^2}(\boldsymbol{x}, \boldsymbol{x}) = \sigma^2$ for all $\boldsymbol{x}$.

## C   GPHLVM VARIATIONAL INFERENCE

As mentioned in § 3.2, when training our GPHLVM on large datasets, we resort to variational inference as originally proposed in (Titsias & Lawrence, 2010). Here we provide the mathematical details about the changes that are needed to train our model via variational inference.

### C.1   COMPUTING THE KL DIVERGENCE BETWEEN TWO HYPERBOLIC WRAPPED NORMAL DISTRIBUTIONS

As mentioned in § 3.2, we approximate the KL divergence between two hyperbolic wrapped distributions via Monte-Carlo sampling. Namely, given two hyperbolic wrapped distributions $q_\phi(\boldsymbol{x})$ and $p(\boldsymbol{x})$, we write

$$\mathrm{KL}\big(q_\phi(\boldsymbol{x})||p(\boldsymbol{x})\big) = \int q_\phi(\boldsymbol{x}) \log \frac{q_\phi(\boldsymbol{x})}{p(\boldsymbol{x})} d\boldsymbol{x} \approx \frac{1}{K} \sum_{k=1}^{K} \log \frac{q_\phi(\boldsymbol{x}_k)}{p(\boldsymbol{x}_k)}, \tag{26}$$

where we used $K$ independent Monte-Carlo samples drawn from $q_\phi(\boldsymbol{x})$ to approximate the KL divergence. These samples are obtained via the procedure described in § 2, i.e., by sampling an element on the tangent space of the origin $\boldsymbol{\mu}_0 = (1, 0, \ldots, 0)^\mathsf{T}$ of $\mathbb{H}^d$, via a Euclidean normal distribution, and then applying the parallel transport operation and the exponential map to project it onto $\mathbb{H}^d$.

### C.2   DETAILS OF THE VARIATIONAL PROCESS

As mentioned in the main text, the marginal likelihood $p(\boldsymbol{Y})$ is approximated via variational inference by approximating the posterior $p(\boldsymbol{X}|\boldsymbol{Y})$ with the hyperbolic variational distribution $q_\phi(\boldsymbol{X})$ as defined by equation 6. The lower bound equation 7 is then obtained, similarly as in (Titsias & Lawrence, 2010), as

$$\log p(\boldsymbol{Y}) = \log \int p(\boldsymbol{Y}|\boldsymbol{X})p(\boldsymbol{X})d\boldsymbol{X} \tag{27}$$

$$= \log \int p(\boldsymbol{Y}|\boldsymbol{X})p(\boldsymbol{X})\frac{q_\phi(\boldsymbol{X})}{q_\phi(\boldsymbol{X})}d\boldsymbol{X} = \log \mathbb{E}_{q_\phi(\boldsymbol{X})}\left[\frac{p(\boldsymbol{Y}|\boldsymbol{X})p(\boldsymbol{X})}{q_\phi(\boldsymbol{X})}\right] \tag{28}$$

$$\geq \mathbb{E}_{q_\phi(\boldsymbol{X})}\left[\log \frac{p(\boldsymbol{Y}|\boldsymbol{X})p(\boldsymbol{X})}{q_\phi(\boldsymbol{X})}\right] = \int q_\phi(\boldsymbol{X}) \log \frac{p(\boldsymbol{Y}|\boldsymbol{X})p(\boldsymbol{X})}{q_\phi(\boldsymbol{X})}d\boldsymbol{X} \tag{29}$$

$$= \int q_\phi(\boldsymbol{X}) \log p(\boldsymbol{Y}|\boldsymbol{X})d\boldsymbol{X} - \int q_\phi(\boldsymbol{X}) \log \frac{q_\phi(\boldsymbol{X})}{p(\boldsymbol{X})}d\boldsymbol{X} \tag{30}$$

$$= \mathbb{E}_{q_\phi(\boldsymbol{X})}[\log p(\boldsymbol{Y}|\boldsymbol{X})] - \mathrm{KL}\big(q_\phi(\boldsymbol{X})||p(\boldsymbol{X})\big), \tag{31}$$

following Jensen's inequality in equation 29. As mentioned in § 3.2, the expectation $\mathbb{E}_{q_\phi(\boldsymbol{X})}[\log p(\boldsymbol{Y}|\boldsymbol{X})]$ can be decomposed into individual terms for each observation dimension as $\sum_{d=1}^{D} \mathbb{E}_{q_\phi(\boldsymbol{X})}[\log p(\boldsymbol{y}_d|\boldsymbol{X})]$, where $\boldsymbol{y}_d$ is the $d$-th column of $\boldsymbol{Y}$. We then define the inducing inputs $\boldsymbol{Z}_d$ and inducing variables $\boldsymbol{u}_d$ the same way as the noiseless observations $\boldsymbol{f}_d$, so that the joint distribution of $\boldsymbol{f}_d$ and $\boldsymbol{u}_d$ can be written as

$$p(\boldsymbol{f}_d, \boldsymbol{u}_d) = \begin{pmatrix} \boldsymbol{f}_d \\ \boldsymbol{u}_d \end{pmatrix} = \mathcal{N}\left(\begin{pmatrix} \boldsymbol{m}_d(\boldsymbol{X}) \\ \boldsymbol{m}_d(\boldsymbol{Z}_d) \end{pmatrix}, \begin{pmatrix} k_d(\boldsymbol{X}, \boldsymbol{X}) & k_d(\boldsymbol{X}, \boldsymbol{Z}_d) \\ k_d(\boldsymbol{Z}_d, \boldsymbol{X}) & k_d(\boldsymbol{Z}_d, \boldsymbol{Z}_d) \end{pmatrix}\right). \tag{32}$$

The lower bound equation 8 is then obtained for each dimension, similarly as in (Hensman et al., 2015), as

$$\log p(\boldsymbol{y}_d|\boldsymbol{X}) = \int \log p(\boldsymbol{y}_d|\boldsymbol{X}, \boldsymbol{u}_d) p(\boldsymbol{u}_d) d\boldsymbol{u}_d \tag{33}$$

$$= \log \int p(\boldsymbol{y}_d|\boldsymbol{X}, \boldsymbol{u}_d) p(\boldsymbol{u}_d) \frac{q_\lambda(\boldsymbol{u}_d)}{q_\lambda(\boldsymbol{u}_d)} d\boldsymbol{u}_d = \log \mathbb{E}_{q_\lambda(\boldsymbol{u}_d)} \left[ \frac{p(\boldsymbol{y}_d|\boldsymbol{X}, \boldsymbol{u}_d) p(\boldsymbol{u}_d)}{q_\lambda(\boldsymbol{u}_d)} \right] \tag{34}$$

$$\geq \mathbb{E}_{q_\lambda(\boldsymbol{u}_d)} \left[ \log \frac{p(\boldsymbol{y}_d|\boldsymbol{X}, \boldsymbol{u}_d) p(\boldsymbol{u}_d)}{q_\lambda(\boldsymbol{u}_d)} \right] = \int q_\lambda(\boldsymbol{u}_d) \log \frac{p(\boldsymbol{y}_d|\boldsymbol{X}, \boldsymbol{u}_d) p(\boldsymbol{u}_d)}{q_\lambda(\boldsymbol{u}_d)} d\boldsymbol{u}_d \tag{35}$$

$$= \int q_\lambda(\boldsymbol{u}_d) \log p(\boldsymbol{y}_d|\boldsymbol{X}, \boldsymbol{u}_d) d\boldsymbol{u}_d - \int q_\lambda(\boldsymbol{u}_d) \log \frac{q_\lambda(\boldsymbol{u}_d)}{p(\boldsymbol{u}_d)} d\boldsymbol{u}_d \tag{36}$$

$$= \mathbb{E}_{q_\lambda(\boldsymbol{u}_d)} \left[ \log p(\boldsymbol{y}_d|\boldsymbol{X}, \boldsymbol{u}_d) \right] - \mathrm{KL}\big(q_\lambda(\boldsymbol{u}_d)||p(\boldsymbol{u}_d)\big) \tag{37}$$

$$\geq \mathbb{E}_{q_\lambda(\boldsymbol{u}_d)} \left[ \mathbb{E}_{p(\boldsymbol{f}_d|\boldsymbol{u}_d)} \left[ \log p(\boldsymbol{y}_d|\boldsymbol{f}_d(\boldsymbol{X})) \right] \right] - \mathrm{KL}\big(q_\lambda(\boldsymbol{u}_d)||p(\boldsymbol{u}_d)\big) \tag{38}$$

$$= \mathbb{E}_{q_\lambda(\boldsymbol{f}_d)} \left[ \log p(\boldsymbol{y}_d|\boldsymbol{f}_d(\boldsymbol{X})) \right] - \mathrm{KL}\big(q_\lambda(\boldsymbol{u}_d)||p(\boldsymbol{u}_d|\boldsymbol{Z}_d)\big) \tag{39}$$

$$= \mathbb{E}_{q_\lambda(\boldsymbol{f}_d)} \left[ \log \mathcal{N}(\boldsymbol{y}_d; \boldsymbol{f}_d(\boldsymbol{X}), \sigma_d^2) \right] - \mathrm{KL}\big(q_\lambda(\boldsymbol{u}_d)||p(\boldsymbol{u}_d|\boldsymbol{Z}_d)\big), \tag{40}$$

where we defined $q_\lambda(\boldsymbol{f}_d) = \int p(\boldsymbol{f}_d|\boldsymbol{u}_d) q_\lambda(\boldsymbol{u}_d) d\boldsymbol{u}_d$ with the Euclidean variational distribution $q_\lambda(\boldsymbol{u}_d) = \mathcal{N}(\boldsymbol{u}_d; \tilde{\boldsymbol{\mu}}_d, \tilde{\boldsymbol{\Sigma}}_d)$, and wrote $p(\boldsymbol{u}_d|\boldsymbol{Z}_d) = p(\boldsymbol{u}_d)$ for simplicity. The inequality equation 35 corresponds to Jensen's inequality, while equation 38 is shown in (Titsias, 2009).

Finally, substituting equation 40 in equation 31 results in the following bound on the marginal likelihood

$$\log p(\boldsymbol{Y}) \geq \sum_{n=1}^{N} \sum_{d=1}^{D} \mathbb{E}_{q_\phi(\boldsymbol{x}_n)} \left[ \mathbb{E}_{q_\lambda(f_{n,d})} \left[ \log \mathcal{N}(y_{n,d}; f_{n,d}(\boldsymbol{x}_n), \sigma_d^2) \right] \right]$$
$$- \sum_{d=1}^{D} \mathrm{KL}\big(q_\lambda(\boldsymbol{u}_d)||p(\boldsymbol{u}_d|\boldsymbol{Z}_d)\big) - \sum_{n=1}^{N} \mathrm{KL}\big(q_\phi(\boldsymbol{x}_n)||p(\boldsymbol{x}_n)\big). \tag{41}$$

## D   MATÉRN KERNELS ON TAXONOMY GRAPHS

As explained in § 4 of the main paper, we leverage the Matérn kernel on graphs proposed by Borovitskiy et al. (2021) to design a kernel for our back-constrained GPHLVM that accounts for the geometry of the taxonomy graph. Here we provide the main equations of such a kernel, and refer the reader to (Borovitskiy et al., 2021) for further details. Formally, let us define a graph $G = (V, E)$ with vertices $V$ and edges $E$ and the *graph Laplacian* as $\boldsymbol{\Delta} = \boldsymbol{D} - \boldsymbol{W}$, where $\boldsymbol{W}$ is the graph adjacency matrix and $\boldsymbol{D}$ its corresponding diagonal degree matrix, with $\boldsymbol{D}_{ii} = \sum_j \boldsymbol{W}_{ij}$. The eigendecomposition $\boldsymbol{U}\boldsymbol{\Lambda}\boldsymbol{U}^\mathsf{T}$ of the Laplacian $\boldsymbol{\Delta}$ is then used to formulate both the SE and Matérn kernels on graphs, as follows,

$$k_{\infty,\kappa}^{\mathbb{G}}(c_n, c_m) = \boldsymbol{U} \left( e^{-\frac{\kappa^2}{2}\boldsymbol{\Lambda}} \right) \boldsymbol{U}^\mathsf{T}, \quad \text{and} \quad k_{\nu,\kappa}^{\mathbb{G}}(c_n, c_m) = \boldsymbol{U} \left( \frac{2\nu}{\kappa^2} + \boldsymbol{\Lambda} \right)^{-\nu} \boldsymbol{U}^\mathsf{T}, \tag{42}$$

where $\kappa$ is the lengthscale (i.e., it controls how distances are measured) and $\nu$ is the smoothness parameter determining mean-squared differentiability of the associated Gaussian process (GP). Note that the graph kernel expressions in equation 42 are obtained by considering the connection between Matérn kernel GPs and stochastic partial differential equations, originally proposed by Whittle (1963) and later extended to Riemannian manifolds in (Borovitskiy et al., 2020). This connection establishes that SE and Matérn GPs satisfy

$$e^{-\frac{\kappa^2}{4}\boldsymbol{\Delta}} \boldsymbol{f} = \mathcal{W}, \quad \text{and} \quad \left( \frac{2\nu}{\kappa^2} + \boldsymbol{\Delta} \right)^{\frac{\nu}{2}} \boldsymbol{f} = \mathcal{W}, \tag{43}$$

where $\mathcal{W} \sim \mathcal{N}(\boldsymbol{0}, \boldsymbol{I})$ and $\boldsymbol{f} : V \to \mathbb{R}$, which lead to definition of graph GPs (Borovitskiy et al., 2021).

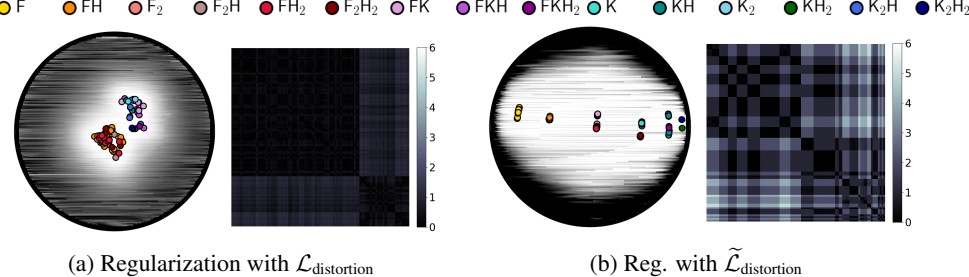

(a) Regularization with $\mathcal{L}_{\text{distortion}}$          (b) Reg. with $\widetilde{\mathcal{L}}_{\text{distortion}}$

Figure 5: Embeddings learned with distortion regularization. *(a)* and *(b)* display the latent embeddings along-side distance matrices after training our GPHLVM model with an added distortion loss $\mathcal{L}_{\text{distortion}}$ as it was originally defined, and with our modified distortion loss $\widetilde{\mathcal{L}}_{\text{distortion}}$, respectively. These embeddings indeed show that our regularizations failed to encode the distances in the graph.

## E    Distortion Loss

As explained in the paper, we focus on two ways of embedding the graph in the hyperbolic space: a global approach using a stress regularization which matches graph distances with geodesic distances, and a combination between this stress regularization and the use of back constraints (see § 4). However, the literature on graph embeddings also surveys a *distortion loss* (Cruceru et al., 2021) given by

$$\mathcal{L}_{\text{distortion}}(\boldsymbol{X}) = \sum_{i<j} \left| \frac{\text{dist}_{\mathbb{H}^Q}(\boldsymbol{x}_i, \boldsymbol{x}_j)^2}{\text{dist}_{\mathbb{G}}(c_i, c_j)^2} - 1 \right|^2, \tag{44}$$

which tries to match the graph and manifold distances by minimizing their ratio's distance to 1.

We found that our problem is more subtle than usual graph embeddings, given that several points in our dataset may correspond to the same graph node (e.g., two different poses in which the left foot is the only limb in contact). Indeed, notice that equation 45 is ill-defined for the case $i = j$, or equivalently, $\text{dist}_{\mathbb{G}}(c_i, c_j)^2 = 0$. This is because all nodes $\boldsymbol{x}_i$ are assumed to be different from each other. However, in our setup, several $\boldsymbol{x}_i$ may correspond to the exact same class in the taxonomy.

Our first attempt to remediate this was to add a simple regularizer $\varepsilon = 10^{-1}$ to the denominator. However, this caused the loss to give more weight to the points where $\text{dist}_{\mathbb{G}}(c_i, c_j)^2 = 0$ (see Fig. 5a for the outcome of training a GPHLVM with this type of regularization). We then considered an alternate definition of distortion in which the term inside the sum is given by

$$\widetilde{\mathcal{L}}_{\text{distortion}}(\boldsymbol{x}_i, \boldsymbol{x}_j) = \begin{cases} \lambda_1 \, \text{dist}_{\mathbb{H}^Q}(\boldsymbol{x}_i, \boldsymbol{x}_j) & \text{if } \boldsymbol{x}_i \text{ and } \boldsymbol{x}_j\text{'s classes are identical} \\ \lambda_2 \mathcal{L}_{\text{distortion}}(\boldsymbol{x}_i, \boldsymbol{x}_j) & \text{otherwise} \end{cases} \tag{45}$$

where $\lambda_1, \lambda_2 \in \mathbb{R}^+$ are hyperparameters. $\lambda_1$ governs how much we encourage latent codes of the same class to collapse into a single point, while $\lambda_2$ weights how much the geodesic distance should match the graph distance. After manual hyperparameter tuning, we obtained the latent space and distance matrix portrayed in Figs. 5a 5b. As can be seen in both accounts, the distortion loss produced lackluster results and failed to properly match the latent space distances with that of the graph. For these experiments, we used a loss scale of 50, $\lambda_1 = 0.01$ and $\lambda_2 = 10$, meaning that we strongly encouraged the distances between non-identical classes to match in ratio.

# F  Additional details on the experiments of § 5

## F.1  Data

For all experiments, we used humans recordings from the KIT Whole-Body Human Motion Database[1] (Mandery et al., 2016b). Additional details on the data of each experiments are described in the sequel.

### F.1.1  Support poses

Table 3 describes the data of the whole-body support pose taxonomy used in the experiments reported in § 5. Each pose is identified with a support pose category, i.e., a node of the graph in Fig. 1-*right*, and with a set of associated contacts. As shown in the table, some support poses include several sets of contacts. For example, the support pose F groups all types of support poses where only one foot is in contact with the environment. In our experiments, we consider an augmented version of the taxonomy that explicitly distinguishes between left and right contacts. Notice that some sets of contacts are not represented in the data and thus do not appear in Table 3.

| Support pose | Augmented support pose | Contacts | Number |
|:---:|:---:|:---:|:---:|
| F | $F^l$ | Left foot | 7 |
| | $F^r$ | Right foot | 6 |
| FH | $F^l H^l$ | Left foot, left hand | 5 |
| | $F^r H^r$ | Right foot, right hand | 6 |
| | $F^l H^r$ | Left foot, right hand | 5 |
| | $F^r H^l$ | Right foot, left hand | 6 |
| $F_2$ | $F_2$ | Left foot, right foot | 6 |
| $FH_2$ | $F^l H_2$ | Left foot, left hand, right hand | 6 |
| | $F^r H_2$ | Right foot, left hand, right hand | 6 |
| $F_2 H$ | $F_2 H^l$ | Left foot, right foot, left hand | 5 |
| | $F_2 H^r$ | Left foot, right foot, right hand | 7 |
| $F_2 H_2$ | $F_2 H_2$ | Left foot, right foot, left hand, right hand | 7 |
| K | $K^l$ | Left knee | 1 |
| | $K^r$ | Right knee | 1 |
| FK | $F^l K^r$ | Left foot, right knee | 2 |
| | $FK^l$ | Right foot, left knee | 3 |
| KH | $K^l H^l$ | Left knee, left hand | 4 |
| | $K^r H^r$ | Right knee, right hand | 1 |
| $K_2$ | $K_2$ | Left knee, right knee | 1 |
| FKH | $F^r K^l H^l$ | Right foot, left knee, left hand | 5 |
| | $F^l K^r H^r$ | Left foot, right knee, right hand | 2 |
| $KH_2$ | $K^l H_2$ | Left knee, left hand, right hand | 1 |
| $K_2 H$ | $K_2 H^l$ | Left knee, right knee, left hand | 2 |
| | $K_2 H^r$ | Left knee, right knee, right hand | 1 |
| $FKH_2$ | $F^r K^l H_2$ | Right foot, left knee, left hand, right hand | 2 |
| $K_2 H_2$ | $K_2 H_2$ | Left knee, right knee, left hand, right hand | 2 |

Table 3: Description of the support poses extracted from the whole-body support pose taxonomy (Borràs et al., 2017) used in our experiments.

### F.1.2  Hand grasps

Fig. 6 shows the hand grasps taxonomy (Stival et al., 2019) and Table 4 describes the data used in § 5. We use grasp data[2] from subjects 2122, 2123, 2125, 2177. The considered human recordings consist of a human grasping an object on a table, lifting it, and placing it back. We consider a single object per grasp type and extract the wrist and finger joint angles of the human when the object is at the highest position. Each grasp is identified with a leaf node of the taxonomy tree. Notice that no data was available for the three-fingers-sphere grasp type.

---

[1] https://motion-database.humanoids.kit.edu/

[2] https://motion-database.humanoids.kit.edu/list/motions/?datasets=3534

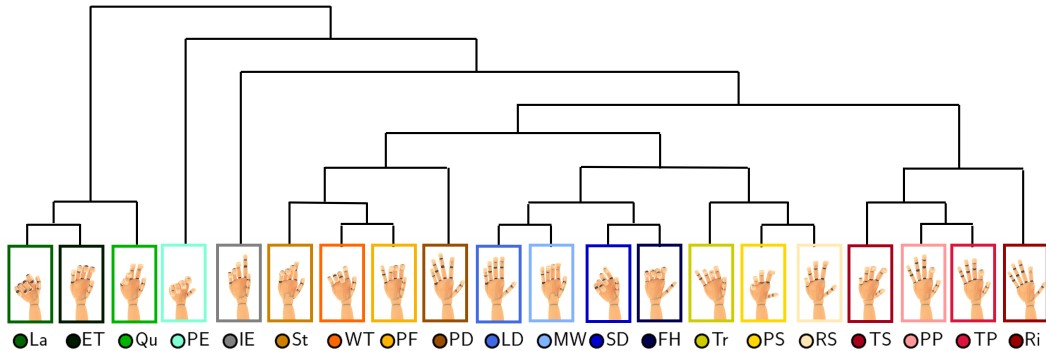

Figure 6: Hand grasps taxonomy (Stival et al., 2019) used in our experiment. Each leaf node of the tree is a hand grasp type.

| Category | Grasp type | Abbreviation | Grasped object | Number |
|---|---|---|---|---|
| Flat grasps | Lateral | La | Padlock | 5 |
| | Extension type | ET | Fruit bars | 5 |
| | Quadpod | Qu | Lever red | 5 |
| | Parallel extension | PE | Fruit bars | 5 |
| | Index finger extension | IE | Knife | 5 |
| Distal grasps | Stick | St | Fizzy tablets | 5 |
| | Writing tripod | WT | Syringe | 5 |
| | Prismatic four fingers | PF | Mixing bowl | 5 |
| | Power disk | PD | Dog | 5 |
| Cylindrical grasps | Large diameter | LD | Dwarf | 5 |
| | Medium wrap | MW | Power tool | 5 |
| | Small diameter | SD | Clamp | 5 |
| | Fixed hook | FH | Fizzy tablets | 5 |
| Spherical grasps | Tripod | Tr | Padlock | 5 |
| | Power sphere | PS | Dog | 5 |
| | Precision sphere | RS | Ball | 5 |
| Ring grasps | Three fingers sphere | TS | - | 0 |
| | Prismatic pinch | PP | Flower cup | 5 |
| | Tip pinch | TP | Chopsticks | 4 |
| | Ring | Ri | Cola bottle | 5 |

Table 4: Description of the grasps extracted from the quantitative grasp taxonomy (Stival et al., 2019) used in our experiments.

### F.1.3 BIMANUAL MANIPULATION

Table 5 describes the data of the bimanual manipulation taxonomy used in the experiments reported in § 5. We use data from subject 1723 executing five different bimanual household activities, namely cut and peel a cucumber, roll dough, stir, and wipe. The taxonomy categories are obtained using the annotations provided in (Krebs & Asfour, 2022). We obtain data for two bimanual categories that do not appear in the dataset (unimanual right and tightly-coupled asymmetric left dominant) by mirroring the motions.

### F.2 TRAINING PARAMETERS AND PRIORS

Table 6 describes the hyperparameters used for the experiments reported in § 5. We used the hyperbolic kernels defined in § 3.1 for the GPHLVMs, and the classical SE kernel for the Euclidean models. For the back-constraints mapping equation 11, we defined $k^{\mathbb{R}^D}(\boldsymbol{y}_n, \boldsymbol{y}_m)$ as the product of a Euclidean SE kernel with lengthscale $\kappa_{\mathbb{R}^D}$, and $k^{\mathbb{G}}(c_n, c_m)$ as a graph Matérn kernel with smoothness $\nu = 2.5$ and lengthscale $\kappa_{\mathbb{G}}$. We additionally scaled the product of kernels with a variance $\sigma_{\mathbb{R}^D, \mathbb{G}}$. For training the back-constrained GPHLVM and GPLVM, we used a Gamma prior

| Bimanual category | Abbreviation | Bimanual activity | Number |
|---|---|---|---|
| Unimanual left | $U_{left}$ | Peel | 5 |
| | | Wipe | 5 |
| Unimanual right | $U_{right}$ | Peel | 5 |
| | | Wipe | 5 |
| Uncoordinated bimanual | B | - | 0 |
| Loosely coupled | LC | Cut | 2 |
| | | Peel | 4 |
| | | Stir | 2 |
| | | Wipe | 2 |
| Tightly-coupled asymmetric left dominant | $TCA_{left}$ | Cut | 2 |
| | | Peel | 3 |
| | | Stir | 2 |
| | | Wipe | 3 |
| Tightly-coupled asymmetric right dominant | $TCA_{right}$ | Cut | 2 |
| | | Peel | 3 |
| | | Stir | 2 |
| | | Wipe | 3 |
| Tightly-coupled symmetric | TCS | Roll | 10 |

Table 5: Description of the bimanual manipulation patterns extracted from the bimanual manipulation taxonomy (Krebs & Asfour, 2022) used in our experiments.

Gamma$(\alpha, \beta)$ with shape $\alpha$ and rate $\beta$ on the lengthscale $\kappa$ of the kernels. The embeddings of all GPLVMs were initialized by minimizing the stress associated with their taxonomy nodes, so that,

$$\boldsymbol{X} = \min_{\boldsymbol{X}} \mathcal{L}_{\text{stress}}, \tag{46}$$

with $\mathcal{L}_{\text{stress}}$ as in equation 10, using the Euclidean and hyperbolic distance between two embeddings for the GPLVMs and GPHLVM, respectively. All models were trained by maximizing the loss $\mathcal{L} = \mathcal{L}_{\text{MAP}} - \gamma \mathcal{L}_{\text{stress}}$, where $\mathcal{L}_{\text{MAP}}$ denotes the log posterior of the model, $\mathcal{L}_{\text{stress}}$ is the stress-based regularization loss defined in equation 10, and $\gamma$ is a parameter balancing the two losses. The optimization was conducted using the Riemannian Adam optimizer (Bécigneul & Ganea, 2019) implemented in Geoopt (Kochurov et al., 2020) with a learning rate of $0.05$.

For the first part of the experiments on taxonomy expansion, we encoded unseen poses of each class for the back-constrained GPLVM and GPHLVM with a stress regularization using the models presented in Table 6. For the second part of the experiments, we left one or several classes out during training and we "embedded" them using the back-constraints mapping. The left-out classes are: FH = $\{F^lH^l, F^rH^l, F^lH^r, F^rH^r\}$, $\{Qu, St, MW, Ri\}$, and tightly-coupled asymmetric left dominant, for the support pose, hand grasp, and bimanual manipulation taxonomies, respectively. The newly-trained models also followed the same hyperparameters presented in Table 6.

### F.3 HYPERBOLIC EMBEDDINGS OF BIMANUAL MANIPULATION CATEGORIES

Fig. 7a- 7c show the learned embeddings of the bimanual manipulation taxonomy alongside error matrices depicting the difference between geodesic and taxonomy graph distances. As discussed in § 5, the models with stress prior result in embeddings that comply with the taxonomy graph structure, with additional intra-class organizations for the back-constrained models. It is worth noticing that, despite the fact that the bimanual taxonomy graph is smaller than the support pose and grasp taxonomy graphs, all Euclidean GPLVMs remain outperformed by the hyperbolic models, which most closely match the taxonomy structure (see also Table 1). As reported in § 5, the back-constrained GPHLVM and GPLVM allow us to properly place unseen poses or taxonomy classes into the latent space (see Figs. 7d- 7e).

### F.4 HYPERBOLIC EMBEDDINGS IN $\mathbb{H}^3$

In this section, we embed the taxonomy data of the three taxonomies used in § 5 into 3-dimensional hyperbolic and Euclidean spaces to analyze the performance of the proposed models in higher-

Table 6: Summary of experiments and list of hyperparameters.

| Taxonomy | Model | Regularization | Loss scale $\gamma$ | Prior on $\kappa_{\mathbb{H}/\mathbb{R}^Q}$ | $\kappa_{\mathbb{R}^D}$ | $\kappa_{\mathbb{G}}$ | $\sigma_{\mathbb{R}^D,\mathbb{G}}$ |
|---|---|---|---|---|---|---|---|
| Support poses | GPLVM on $\mathbb{R}^2$ | No regularizer | 0 | None | - | - | - |
| | | Stress | 7000 | None | - | - | - |
| | | BC + Stress | 5000 | Gamma(2, 2) | 2.0 | 0.8 | 2 |
| | GPHLVM on $\mathbb{H}^2$ | No regularizer | 0 | None | - | - | - |
| | | Stress | 7000 | None | - | - | - |
| | | BC + Stress | 5000 | Gamma(2, 2) | 2.0 | 0.8 | 2 |
| | GPLVM on $\mathbb{R}^3$ | No regularizer | 0 | None | - | - | - |
| | | Stress | 10000 | None | - | - | - |
| | | BC + Stress | 8000 | Gamma(2, 2) | 2.0 | 0.8 | 2 |
| | GPHLVM on $\mathbb{H}^3$ | No regularizer | 0 | None | - | - | - |
| | | Stress | 10000 | None | - | - | - |
| | | BC + Stress | 8000 | Gamma(2, 2) | 2.0 | 0.8 | 2 |
| Grasps | GPLVM on $\mathbb{R}^2$ | No regularizer | 0 | None | - | - | - |
| | | Stress | 5500 | None | - | - | - |
| | | BC + Stress | 2000 | Gamma(2, 2) | 1.8 | 1.5 | 2 |
| | GPHLVM on $\mathbb{H}^2$ | No regularizer | 0 | None | - | - | - |
| | | Stress | 5500 | None | - | - | - |
| | | BC + Stress | 2000 | Gamma(2, 2) | 1.8 | 1.5 | 2 |
| | GPLVM on $\mathbb{R}^3$ | No regularizer | 0 | None | - | - | - |
| | | Stress | 6000 | None | - | - | - |
| | | BC + Stress | 3000 | Gamma(2, 2) | 1.8 | 1.5 | 2 |
| | GPHLVM on $\mathbb{H}^3$ | No regularizer | 0 | None | - | - | - |
| | | Stress | 6000 | None | - | - | - |
| | | BC + Stress | 3000 | Gamma(2, 2) | 1.8 | 1.5 | 2 |
| Bimanual manipulation categories | GPLVM on $\mathbb{R}^2$ | No regularizer | 0 | None | - | - | - |
| | | Stress | 1500 | None | - | - | - |
| | | BC + Stress | 1000 | Gamma(2, 2) | 3.0 | 1.5 | 2 |
| | GPHLVM on $\mathbb{H}^2$ | No regularizer | 0 | None | - | - | - |
| | | Stress | 1500 | None | - | - | - |
| | | BC + Stress | 1000 | Gamma(2, 2) | 3.0 | 1.5 | 2 |
| | GPLVM on $\mathbb{R}^3$ | No regularizer | 0 | None | - | - | - |
| | | Stress | 6000 | None | - | - | - |
| | | BC + Stress | 1200 | Gamma(2, 2) | 3.0 | 1.5 | 2 |
| | GPHLVM on $\mathbb{H}^3$ | No regularizer | 0 | None | - | - | - |
| | | Stress | 6000 | None | - | - | - |
| | | BC + Stress | 1200 | Gamma(2, 2) | 3.0 | 1.5 | 2 |

dimensional latent spaces. We test the GPHLVM and GPLVM without regularization, with stress prior, and with back-constraints coupled with stress prior, similarly to the experiments on 2-dimensional latent spaces reported in § 5 and App. F.3. Figs. 8a-8c, Figs. 9a- 9c, and Figs. 10a- 10c show the learned embeddings alongside the corresponding error matrices for the whole-body support pose taxonomy, the hand grasps taxonomy, and the bimanual manipulation taxonomy, respectively. As expected, and similarly to the 2-dimensional embeddings, the models without regularization do not encode any meaningful distance structure in the latent spaces (see Figs. 8a- 9a, and 10a). In contrast, the models with stress prior result in embeddings that comply with the taxonomy graph structure, and the back constraints further organize the embeddings inside a class according to the similarity between their observations (see Figs. 8b-8c, Figs. 9b- 9c, and Figs. 10b- 10c).

In general, we observe a prominent stress reduction for the Euclidean and hyperbolic 3-dimensional latent spaces compared to the 2-dimensional ones (see Table 1). This is due to the increase of volume available to match the graph structure in 3-dimensional spaces relative to 2-dimensional ones. In the case of the whole-body support pose taxonomy, the Euclidean models with 3-dimensional latent space slightly outperform the 3-dimensional hyperbolic embeddings. We attribute this to the cyclic graph structure of the taxonomy. Such type of structure has been shown to be better embedded in spherical or Euclidean spaces (Gu et al., 2019). Interestingly, despite the cyclic graph structure of the support pose taxonomy, the Euclidean models are still outperformed by the hyperbolic embeddings

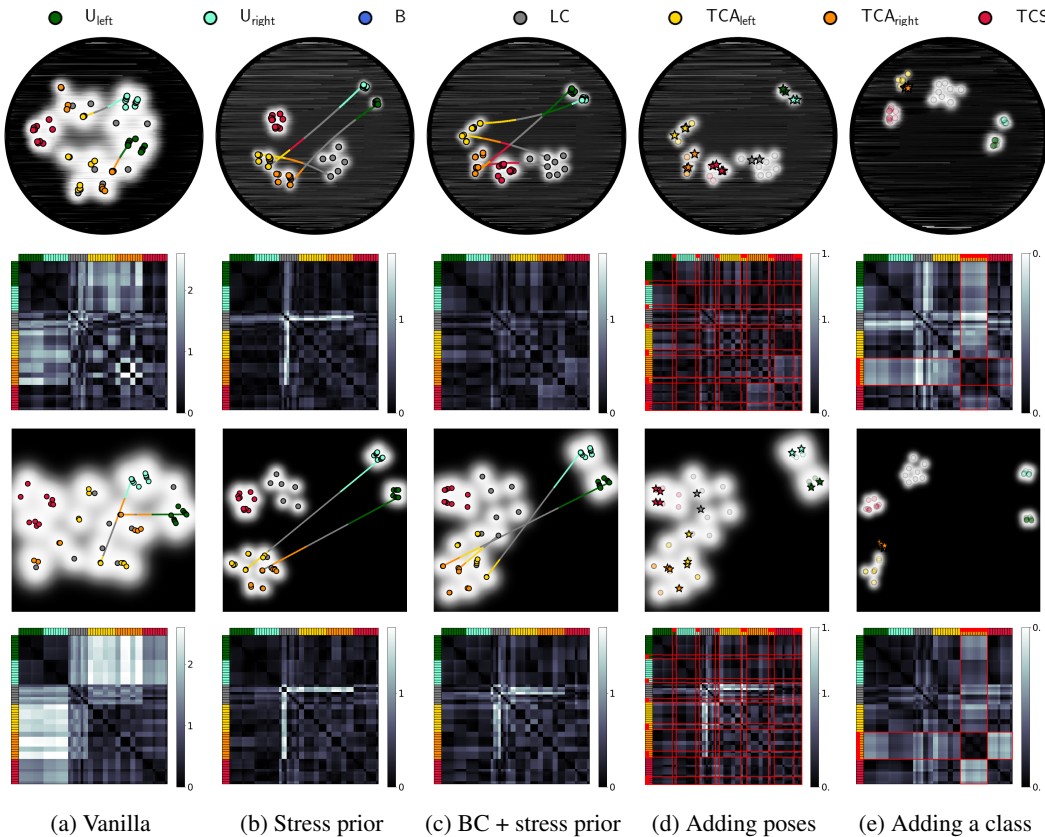

Figure 7: Bimanual manipulation categories: The first and last two rows show the latent embeddings and examples of interpolating geodesics in $\mathcal{P}^2$ and $\mathbb{R}^2$, followed by pairwise error matrices between geodesic and taxonomy graph distances. Background colors indicate the GPLVM uncertainty. Added poses *(d)* and classes *(e)* are marked with stars and highlighted with red in the error matrices.

in the 2-dimensional case (see § 5 and Table 1). This suggests that the increase of volume available to match the graph structure in hyperbolic spaces compared to Euclidean spaces leads to better low-dimensional representations of taxonomy data, including those with cyclic graph structure.

Concerning the hand grasps taxonomy and the bimanual manipulation taxonomy, all Euclidean models are still outperformed by the 3-dimensional hyperbolic embeddings (see Table 1). This is due to the fact that the volume of balls in hyperbolic space increases exponentially with respect to the radius of the ball rather than polynomially as in Euclidean space. This property makes hyperbolic spaces ideal to embed taxonomies with a tree-like structure. Moreover, similarly to the 2-dimensional cases, the back-constrained GPHLVM and GPLVM allow us to properly place unseen poses or taxonomy classes into the latent space (see Figs. 8d-8e, Figs. 9d- 9e, and Figs. 10d- 10e).

### F.5 MARGINAL LOG-LIKELIHOODS OF TRAINED MODELS

Table 7 shows the marginal log-likelihood (MLL) of the GPHLVM and GPLVM described in § 5. We observe that the hyperbolic and Euclidean models achieve the highest likelihood for the whole-body support pose taxonomy with 2 and 3-dimensional latent spaces, respectively. For the grasps taxonomy, all hyperbolic models achieve a higher likelihood than their Euclidean counterparts. Finally, the hyperbolic and Euclidean models achieve similar likelihood in the case of the bimanual manipulation taxonomy.

### F.6 ADDITIONAL MOTIONS OBTAINED VIA GEODESIC INTERPOLATION AND COMPARISONS

Figs. 11- 13 show additional examples of motions obtained via geodesic interpolation between two embeddings of the whole-body support pose taxonomy in the latent space of the GPHLVM. The generated motions look realistic, smoothly interpolate between the given initial and final body poses, and are consistent with the transitions between classes encoded in the taxonomy. In comparison, motions obtained via linear interpolation between two embeddings in the Euclidean latent space of

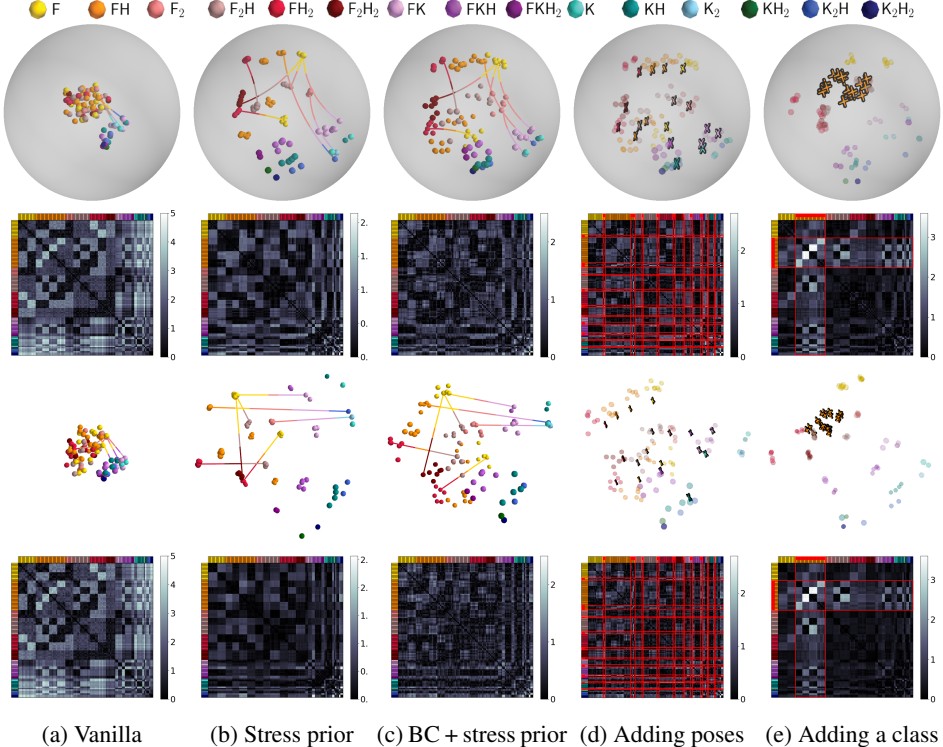

Figure 8: Support poses: The first and last two rows show the latent embeddings and examples of interpolating geodesics in $\mathcal{P}^3$ and $\mathbb{R}^3$, followed by pairwise error matrices between geodesic and graph distances. Embeddings colors match those of Fig. 1-*right*. Added poses *(d)* and classes *(e)* are marked with crosses and highlighted with red in the error matrices.

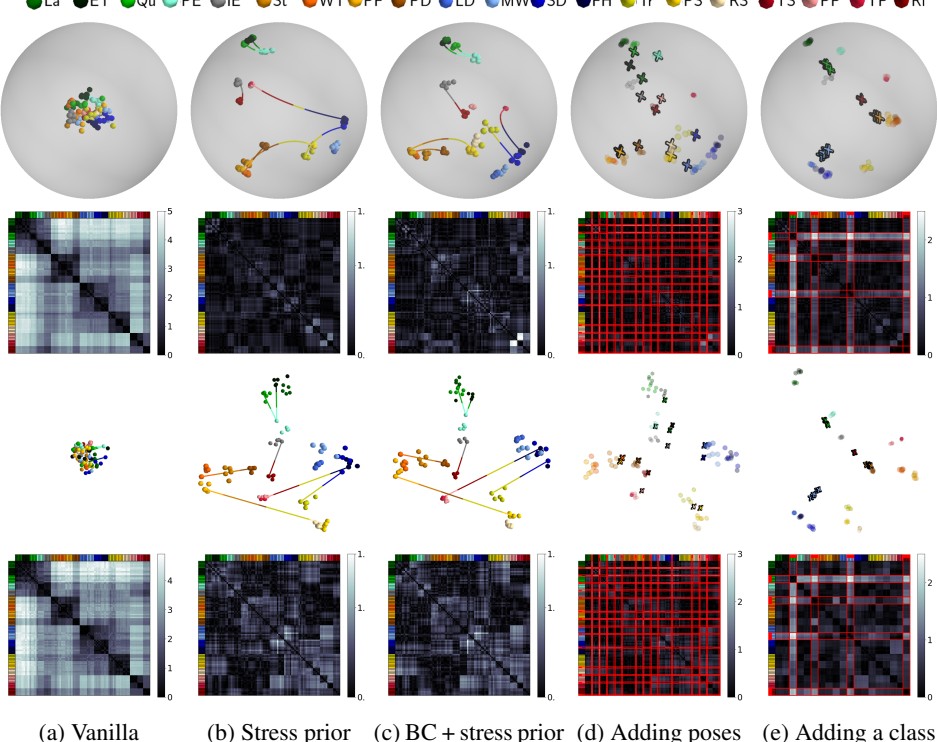

Figure 9: Grasps: The first and last two rows show the latent embeddings and examples of interpolating geodesics in $\mathcal{P}^3$ and $\mathbb{R}^3$, followed by pairwise error matrices between geodesic and taxonomy graph distances. Embeddings colors match those of Fig. 4. Added poses *(d)* and classes *(e)* are marked with crosses and highlighted with red in the error matrices.

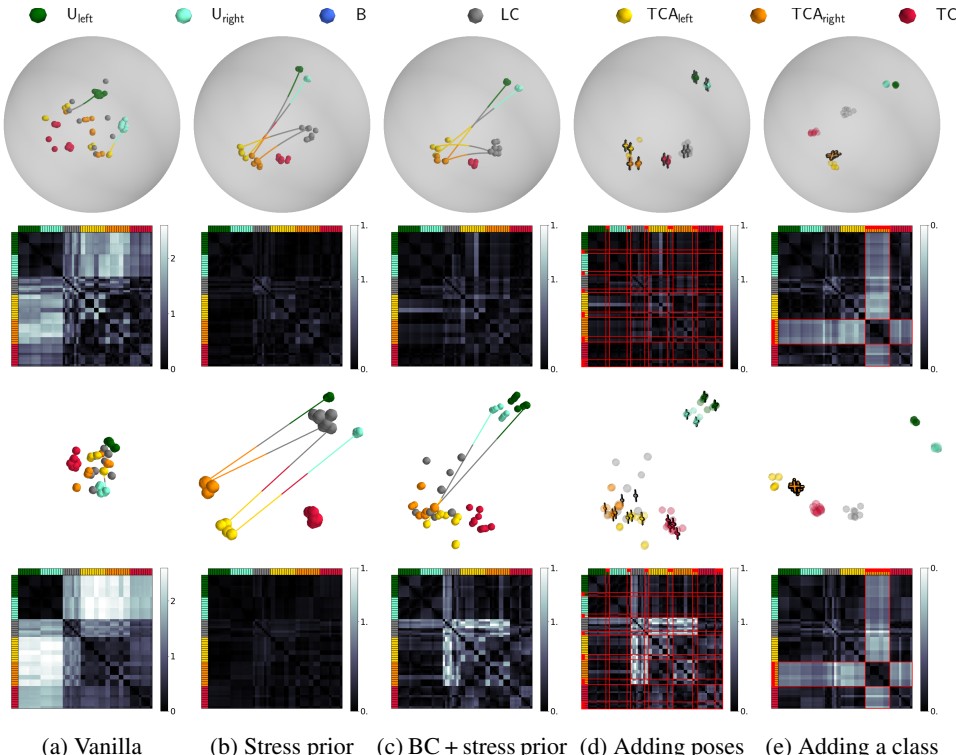

|  |  |  |  |  |
|---|---|---|---|---|
| (a) Vanilla | (b) Stress prior | (c) BC + stress prior | (d) Adding poses | (e) Adding a class |

Figure 10: Bimanual manipulation categories: The first and last two rows show the latent embeddings and examples of interpolating geodesics in $\mathcal{P}^3$ and $\mathbb{R}^3$, followed by pairwise error matrices between geodesic and taxonomy graph distances. Added poses *(d)* and classes *(e)* are marked with crosses and highlighted with red in the error matrices.

Table 7: Marginal log-likelihood per geometry and regularization.

| Taxonomy | Model | No reg. | Stress | BC + Stress |
|---|---|---|---|---|
| Support poses | GPLVM, $\mathbb{R}^2$ | **6.96** | $-53.71$ | $-37.36$ |
|  | GPHLVM, $\mathbb{H}^2$ | 5.52 | $\mathbf{-48.39}$ | $\mathbf{-33.96}$ |
|  | GPLVM, $\mathbb{R}^3$ | **10.63** | $\mathbf{-38.17}$ | $\mathbf{-27.91}$ |
|  | GPHLVM, $\mathbb{H}^3$ | 8.71 | $-45.64$ | $-32.01$ |
| Grasps | GPLVM, $\mathbb{R}^2$ | **9.97** | $-18.03$ | 0.92 |
|  | GPHLVM, $\mathbb{H}^2$ | 7.88 | $\mathbf{-4.03}$ | **1.94** |
|  | GPLVM, $\mathbb{R}^3$ | **12.15** | $-4.19$ | 4.84 |
|  | GPHLVM, $\mathbb{H}^3$ | 9.60 | **0.60** | **6.80** |
| Bimanual manipulation categories | GPLVM, $\mathbb{R}^2$ | **79.50** | **71.91** | 66.19 |
|  | GPHLVM, $\mathbb{H}^2$ | 78.76 | 70.64 | **72.11** |
|  | GPLVM, $\mathbb{R}^3$ | 83.13 | **68.89** | **80.02** |
|  | GPHLVM, $\mathbb{H}^3$ | **84.55** | 66.97 | 79.02 |

the GPLVM look less realistic. In particular, the resulting kneeling poses often look unnatural (see Figs. 11 and 13), and wavering wrist angles are often observed during the motions (see Figs. 11 and 12).

We also compare the trajectories generated via geodesic interpolation with the trajectories generated in the latent space of VPoser (Pavlakos et al., 2019, Sec. 3.3), a state-of-the art human pose latent space obtained from a VAE trained on MoCap data and used to generate human motions. VPoser was introduced by Pavlakos et al. (2019) as a body pose prior to address the problem of building a full 3D model of human gestures by learning a deep neural network that jointly models the human body, face and hands from RBG images. Pavlakos et al. (2019) released the weights of their model under a non-commercial licence.[3] Of the two models available, we downloaded version 2, and

---

[3]https://smpl-x.is.tue.mpg.de/

Table 8: Poses used when comparing with VPoser (Pavlakos et al., 2019). In our notation, the files inside the KIT subset of AMASS (Mahmood et al., 2019) are structured into subfolders of name `entry_id`; each `.npz` file contains an array of body poses, and the exact pose used in the comparison is specified by the index $t$.

| Trajectory | File for source (`entry_id`, $t$ =`index`) | File for target (`entry_id`, $t$ =`index`) |
|---|---|---|
| $F^lH^r$ to $K_2H^r$ (Fig. 11) | Walk w. handrail table beam, left, Nr. 01 (675, $t = 250$) | Kneel up w. right hand, Nr. 01 (3, $t = 185$) |
| $F^r$ to $F^rH_2$ (Fig. 12a) | Walk w. handrail table beam, left, Nr. 01 (675, $t = 100$) | Walk w. handrail table beam, left, Nr. 01 (675, $t = 300$) |
| $F^l$ to $F_2H_2$ (Fig. 12b) | Walk at medium speed Nr. 01 (450, $t = 320$) | Walk w. handrail table beam, left, Nr. 01 (675, $t = 250$) |
| $F^l$ to $F^lK^r$ (Fig. 13a) | Walk at medium speed Nr. 01 (450, $t = 320$) | Kneel up w. right hand Nr. 09 (3, $t = 150$) |
| $F_2$ to $K_2$ (Fig. 13b) | Walk at medium speed Nr. 01 (450, $t = 10$) | Kneel up w. left hand Nr. 01 (3, $t = 50$) |

followed the instructions on their repository for set-up.[4] Since our human poses used a different number of joints, we searched inside the KIT dataset part of the AMASS dataset (Mahmood et al., 2019) for similar poses with the same contacts configuration. Table 8 shows the exact poses used in the comparison. These poses were embedded into the latent space of VPoser. The motions obtained via linear interpolation in the space of VPoser are displayed in the bottom rows of Figs. 11- 13. We observe that the motions generated by our approach are as realistic as the ones obtained from VPoser. It is worth noticing that VPoser is trained on full human motion trajectories and a large dataset of 1M datapoints. Therefore, it is natural that it can retrieve realistic human motions. This is also the case for other models such as TEACH Athanasiou et al. (2022) and text-conditioned human motion diffusion models (Shafir et al., 2023), which are trained on full human motion trajectories and conditioned on textual prompts to generate sequences of human motions. In contrast, the GPHLVM is not trained on full trajectories, but only on 100 single human poses. Instead, GPLHVM leverages the robotic taxonomy and geodesic interpolation as a motion generation mechanism. Notice that the latent space of the GPHLVM is of low dimension compared to the 32-dimensional latent space of VPoser.

It is important to emphasize that augmenting the support pose taxonomy to explicitly distinguish between left and right contact is crucial for generating realistic motions with the GPHLVM. With the original taxonomy, poses with very different feet and hands positions may belong to the same class. For instance, *a right foot contact with a left hand contact on the handrail* or a *left foot contact with a right hand contact on the table* both belong to the same FH node in the original taxonomy. In contrast, differentiating between left and right contacts allows very different poses to be placed far apart in the latent space. For instance, the two aforementioned poses are identified with the nodes $F^lH^r$ and $F^rH^l$ in the augmented taxonomy.

Figs. 14 and 15 show additional examples of motions obtained via geodesic interpolation between two embeddings of the hand grasps taxonomy in the latent space of the GPHLVM. Similarly as for the support pose taxonomy, the generated motions look realistic and smoothly interpolate between the given initial and final grasps. In comparison, motions obtained via linear interpolation between two embeddings in the Euclidean latent space of the GPLVM are less realistic. They display less regular interpolation patterns (see Fig. 14c) and are often noisy, featuring wavering wrist or finger motions (see Figs. 14a, 14b, and 15b). Moreover, the generated grasps reflect less accurately the taxonomy categories (see, e.g., the parallel extension grasp in Fig. 14b or the tip pinch grasp of Fig. 15a). Interestingly, the geodesic interpolation between two grasps in the latent space of the GPHLVM allows us to generate unobserved transitions between hand grasps. As such, it offers us a mechanism to generate data that are generally difficult to collect via human motion recordings.

## F.7 RUNTIME

In order to show the computational cost of our approach, we ran a set of experiments to measure the average runtime for the training and decoding phases, using 2 and 3-dimensional latent spaces. As a reference, we added the runtime measurements of Euclidean counterpart, that is, the vanilla GPLVM. Table 9 shows the runtime measurements. Note that the main computational burden arises in our GPLHVM with a 2-dimensional latent space, which is in sharp contrast with the experiments using a 3-dimensional latent space. As discussed in the main paper, this increase in computational cost is mainly attributed to the 2-dimensional hyperbolic kernel.

---

[4] https://github.com/nghorbani/human_body_prior (vposer.ipynb).

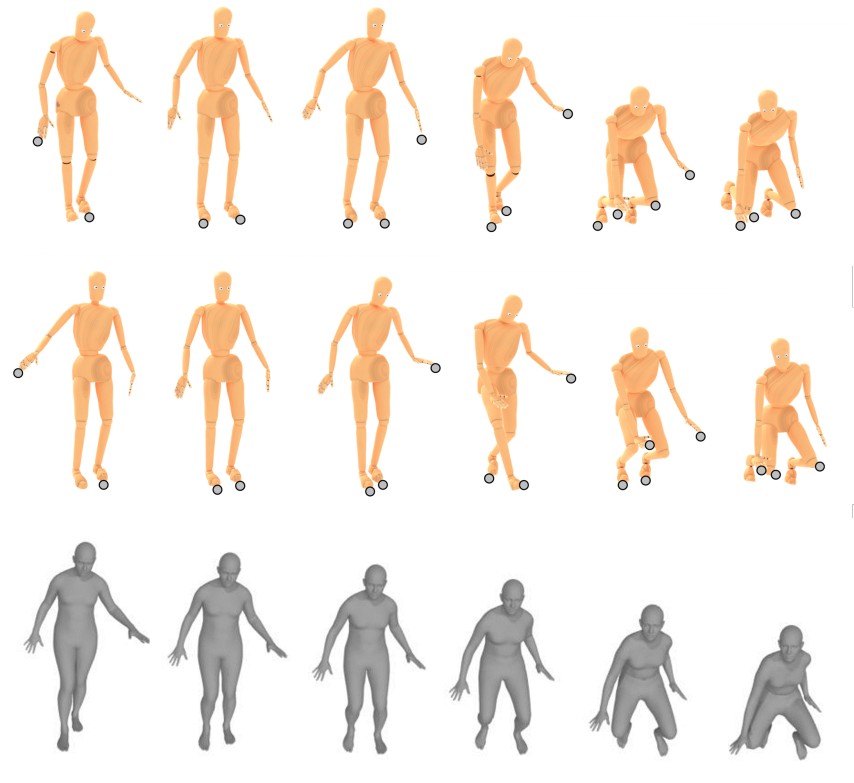

(a) $F^lH^r$ to $K_2H^r$

Figure 11: Generated motions for support poses. *Top:* Motion obtained via geodesic interpolation in the latent space of the back-constrained GPHLVM trained on the support pose taxonomy (Fig. 2c). *Middle:* Motion obtained via linear interpolation in the latent space of the corresponding back-constrained GPLVM. *Bottom:* Motion obtained via linear interpolation in the latent space of VPoser. Contacts are depicted by gray circles in the two first rows.

Table 9: Average runtime for training and decoding phases of our GPHLVM and vanilla GPLVM over 10 experiments of the hand grasps taxonomy, using 2 and 3-dimensional latent spaces for both models. Training time was measured over 500 iterations for both models. The implementations are fully developed on Python, and the runtime measurements were taken using a standard laptop with 32 GB RAM, Intel Xeon CPU E3-1505M v6 processor, and Ubuntu 20.04 LTS.

|  | **GPHLVM** $Q = 2$ | **GPHLVM** $Q = 3$ | **GPLVM** $Q = 2$ | **GPLVM** $Q = 3$ |
|---|---|---|---|---|
| Training | $414.67s \pm 30.87$ | $6.887s \pm 0.307$ | $2.978s \pm 0.082$ | $3.148s \pm 0.171$ |
| Decoding | $2.74s \pm 0.487$ | $10.34ms \pm 1.05$ | $6.256ms \pm 0.314$ | $6.774ms \pm 0.545$ |

# G ADDITIONAL COMPARISONS

## G.1 COMPARISON AGAINST VARIATIONAL AUTOENCODERS

In this section, we compare the trained GPHLVMs of Figs. 2, 3, and 7 with two additional baselines: a vanilla variational autoencoder (VAE) and a hyperbolic variant of this VAE in which the latent space is the Lorentz model of hyperbolic geometry (akin to Mathieu et al. (2019)). Both VAEs are designed with 12 input nodes, 6 hidden nodes, a 2-dimensional latent space, and a symmetric decoder. Their encoder specifies the mean and standard deviation of a normal distribution (resp. wrapped normal for the hyperbolic VAE), and their decoder specifies the mean and standard deviation of the normal distribution that governs the reconstructions. Both models are trained by maximizing an Evidence Lower Bound (ELBO) under similar regimes as the GPHLVMs, i.e., 1000 epochs with a learning rate of 0.05. The KL divergence for the hyperbolic VAE is computed using Monte Carlo estimates.

Table 10: Average stress and reconstruction error per model, geometry, and regularization.

| | | Stress | | | Reconstruction error | | |
|---|---|---|---|---|---|---|---|
| | | No reg. | Stress | BC + Stress | No reg. | Stress | BC + Stress |
| Support poses | GPLVM, $\mathbb{R}^2$ | $3.93 \pm 3.97$ | $0.58 \pm 0.94$ | $0.63 \pm 0.94$ | $\mathbf{0.05 \pm 0.05}$ | $0.17 \pm 0.18$ | $\mathbf{0.11 \pm 0.12}$ |
| | VAE, $\mathbb{R}^2$ | $1.75 \pm 2.29$ | $0.54 \pm 0.80$ | | $0.15 \pm 0.18$ | $0.18 \pm 0.20$ | |
| | VAE, $\mathbb{H}^2$ | $4.81 \pm 4.29$ | $0.57 \pm 0.85$ | | $0.18 \pm 0.21$ | $0.18 \pm 0.20$ | |
| | GPHLVM, $\mathbb{H}^2$ | $\mathbf{2.05 \pm 2.50}$ | $\mathbf{0.51 \pm 0.82}$ | $\mathbf{0.53 \pm 0.83}$ | $0.07 \pm 0.07$ | $\mathbf{0.16 \pm 0.17}$ | $0.15 \pm 0.16$ |
| | GPLVM, $\mathbb{R}^3$ | $\mathbf{3.76 \pm 3.74}$ | $\mathbf{0.24 \pm 0.40}$ | $\mathbf{0.29 \pm 0.39}$ | $\mathbf{0.03 \pm 0.03}$ | $0.17 \pm 0.18$ | $\mathbf{0.08 \pm 0.09}$ |
| | VAE, $\mathbb{R}^3$ | $2.10 \pm 2.64$ | $0.31 \pm 0.40$ | | $0.38 \pm 0.47$ | $0.16 \pm 0.19$ | |
| | VAE, $\mathbb{H}^3$ | $4.53 \pm 4.23$ | $0.38 \pm 0.55$ | | $0.17 \pm 0.21$ | $0.17 \pm 0.20$ | |
| | GPHLVM, $\mathbb{H}^3$ | $3.78 \pm 3.71$ | $0.30 \pm 0.38$ | $0.35 \pm 0.45$ | $\mathbf{0.03 \pm 0.03}$ | $\mathbf{0.16 \pm 0.17}$ | $\mathbf{0.08 \pm 0.09}$ |
| Grasps | GPLVM, $\mathbb{R}^2$ | $7.25 \pm 5.40$ | $0.39 \pm 0.41$ | $0.40 \pm 0.44$ | $\mathbf{0.04 \pm 0.04}$ | $\mathbf{0.03 \pm 0.03}$ | $\mathbf{0.03 \pm 0.03}$ |
| | VAE, $\mathbb{R}^2$ | $3.52 \pm 4.31$ | $0.48 \pm 0.55$ | | $0.11 \pm 0.12$ | $0.13 \pm 0.15$ | |
| | VAE, $\mathbb{H}^2$ | $8.99 \pm 6.20$ | $0.70 \pm 1.28$ | | $0.13 \pm 0.16$ | $0.14 \pm 0.15$ | |
| | GPHLVM, $\mathbb{H}^2$ | $\mathbf{5.47 \pm 4.07}$ | $\mathbf{0.14 \pm 0.16}$ | $\mathbf{0.18 \pm 0.29}$ | $0.05 \pm 0.05$ | $0.08 \pm 0.07$ | $0.09 \pm 0.09$ |
| | GPLVM, $\mathbb{R}^3$ | $\mathbf{8.15 \pm 5.85}$ | $0.14 \pm 0.18$ | $0.15 \pm 0.19$ | $0.03 \pm 0.03$ | $0.14 \pm 0.18$ | $0.15 \pm 0.19$ |
| | VAE, $\mathbb{R}^3$ | $2.71 \pm 3.47$ | $0.25 \pm 0.32$ | | $0.10 \pm 0.13$ | $0.14 \pm 0.16$ | |
| | VAE, $\mathbb{H}^3$ | $8.28 \pm 5.94$ | $0.33 \pm 0.59$ | | $0.11 \pm 0.14$ | $0.12 \pm 0.14$ | |
| | GPHLVM, $\mathbb{H}^3$ | $8.37 \pm 5.71$ | $\mathbf{0.04 \pm 0.08}$ | $\mathbf{0.07 \pm 0.18}$ | $\mathbf{0.03 \pm 0.02}$ | $\mathbf{0.01 \pm 0.01}$ | $\mathbf{0.02 \pm 0.02}$ |
| Bimanual manipulation categories | GPLVM, $\mathbb{R}^2$ | $2.03 \pm 2.15$ | $0.13 \pm 0.33$ | $0.15 \pm 0.31$ | $0.01 \pm 0.02$ | $0.01 \pm 0.01$ | $0.02 \pm 0.02$ |
| | VAE, $\mathbb{R}^2$ | $1.70 \pm 1.97$ | $0.12 \pm 0.20$ | | $0.11 \pm 0.18$ | $0.12 \pm 0.17$ | |
| | VAE, $\mathbb{H}^2$ | $1.89 \pm 1.85$ | $0.10 \pm 0.15$ | | $0.12 \pm 0.18$ | $0.12 \pm 0.17$ | |
| | GPHLVM, $\mathbb{H}^2$ | $\mathbf{0.98 \pm 1.26}$ | $\mathbf{0.11 \pm 0.33}$ | $\mathbf{0.09 \pm 0.12}$ | $0.04 \pm 0.04$ | $0.03 \pm 0.04$ | $0.04 \pm 0.04$ |
| | GPLVM, $\mathbb{R}^3$ | $2.39 \pm 2.36$ | $\mathbf{0.01 \pm 0.01}$ | $0.20 \pm 0.38$ | $0.01 \pm 0.01$ | $\mathbf{0.01 \pm 0.01}$ | $0.01 \pm 0.01$ |
| | VAE, $\mathbb{R}^3$ | $2.58 \pm 2.76$ | $0.05 \pm 0.09$ | | $0.08 \pm 0.15$ | $0.12 \pm 0.18$ | |
| | VAE, $\mathbb{H}^3$ | $1.76 \pm 1.84$ | $0.11 \pm 0.17$ | | $0.03 \pm 0.04$ | $0.12 \pm 0.18$ | |
| | GPHLVM, $\mathbb{H}^3$ | $\mathbf{1.18 \pm 1.35}$ | $\mathbf{0.01 \pm 0.03}$ | $\mathbf{0.04 \pm 0.08}$ | $\mathbf{0.00 \pm 0.01}$ | $\mathbf{0.01 \pm 0.01}$ | $\mathbf{0.00 \pm 0.01}$ |

Figs. 16, 17, and 18 show the learned embeddings of the Euclidean and hyperbolic VAE with 2 and 3-dimensional latent spaces alongside the corresponding error matrices between geodesic and taxonomy graph distances for the whole-body support pose taxonomy, the hand grasps taxonomy, and the bimanual manipulation taxonomy. Concerning the support pose taxonomy, the vanilla VAE models only seem to capture a global structure that separates standing from kneeling poses. Although adding a stress regularization as for the GPHLVM helps preserve the graph distance structure, the embeddings of different classes are not as well separated as in our GPHLVM models (see Fig. 16 vs Fig. 2). Moreover, when compared to our proposed GPHLVM, all VAE models provide a subpar uncertainty modeling in their latent spaces. Similar insights can be drawn for the embeddings learned for the hand grasps taxonomy, shown in Fig. 17, and for the bimanual manipulation taxonomy, displayed in Fig. 18.

Table 10 shows that the VAE baselines result in higher average stress than the GPLVMs. In other words, our proposed GPHLVM consistently outperforms all VAEs to encode meaningful taxonomy information in the latent space. Moreover, the GPLVMs consistently achieve a lower reconstruction error than the VAE baseline. We argue that VAEs are not the right tool for our target applications. When training VAEs, the Kullback-Leibler term in the ELBO tries to regularize the latent space to match a unit Gaussian. This regularization is in stark contrast with our goal of separating the embeddings to preserve the taxonomy graph distances.

## G.2 COMPARISON AGAINST LEARNED MANIFOLDS

We compare the proposed GPHLVM to a GPLVM that learns a Riemannian manifold from data (Tosi et al., 2014). Fig. 19 shows the learned latent space including the embeddings and the volume of the Riemannian metric of the learned manifold, alongside distance matrices for the three considered robotics taxonomies. Overall, the model is unable to capture the local and global taxonomy structure. This is due to the fact that the learned Riemannian metric is designed to be high in regions with high uncertainty, thus leading to shortest paths, i.e., geodesics, avoiding these regions. As such, this model was not designed for hierarchical discrete data and does not embed any knowledge about the taxonomy. This is further reflected by the resulting high stress values (see Fig. 19d).

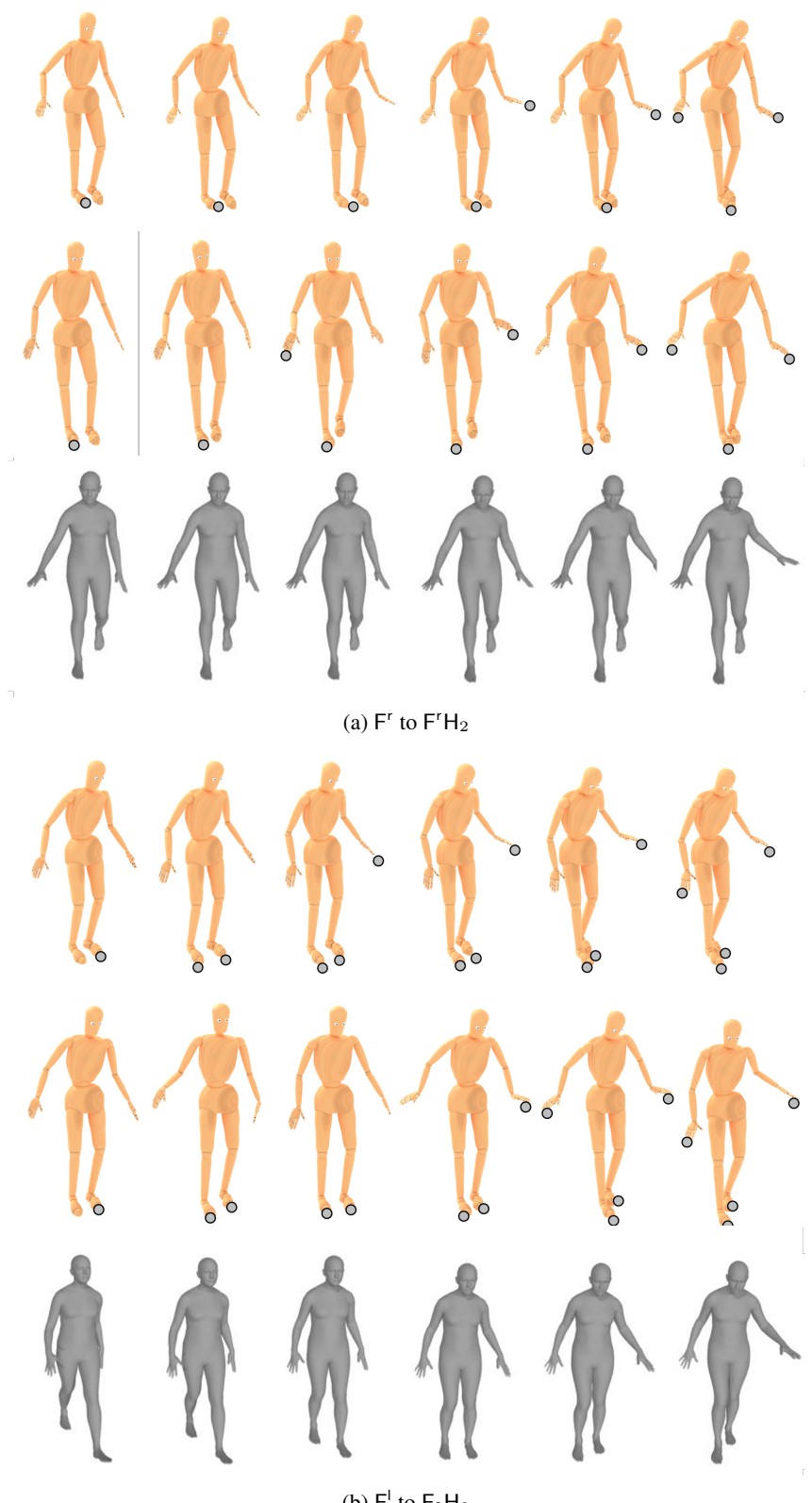

(a) $F^r$ to $F^r H_2$

(b) $F^l$ to $F_2 H_2$

Figure 12: Generated motions for support poses. *Top:* Motions obtained via geodesic interpolation in the latent space of the back-constrained GPHLVM trained on the support pose taxonomy (Fig. 2c). *Middle:* Motions obtained via linear interpolation in the latent space of the corresponding back-constrained GPLVM. *Bottom:* Motions obtained via linear interpolation in the latent space of VPoser. Contacts are depicted by gray circles.

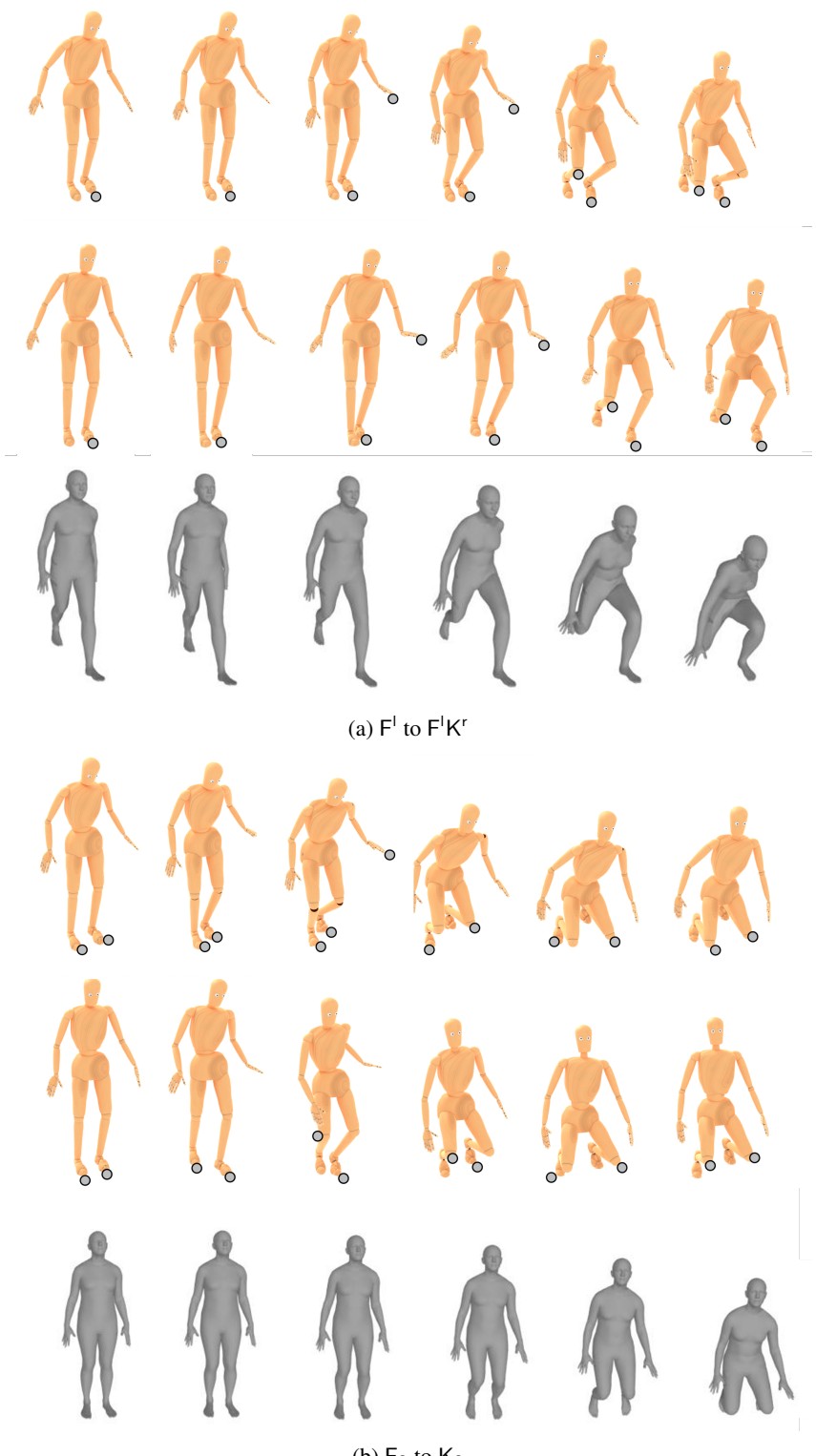

Figure 13: Generated motions for support poses. *Top:* Motions obtained via geodesic interpolation in the latent space of the back-constrained GPHLVM trained on the support pose taxonomy (Fig. 2c). *Middle:* Motions obtained via linear interpolation in the latent space of the corresponding back-constrained GPLVM. *Bottom:* Motions obtained via linear interpolation in the latent space of VPoser. Contacts are depicted by gray circles.

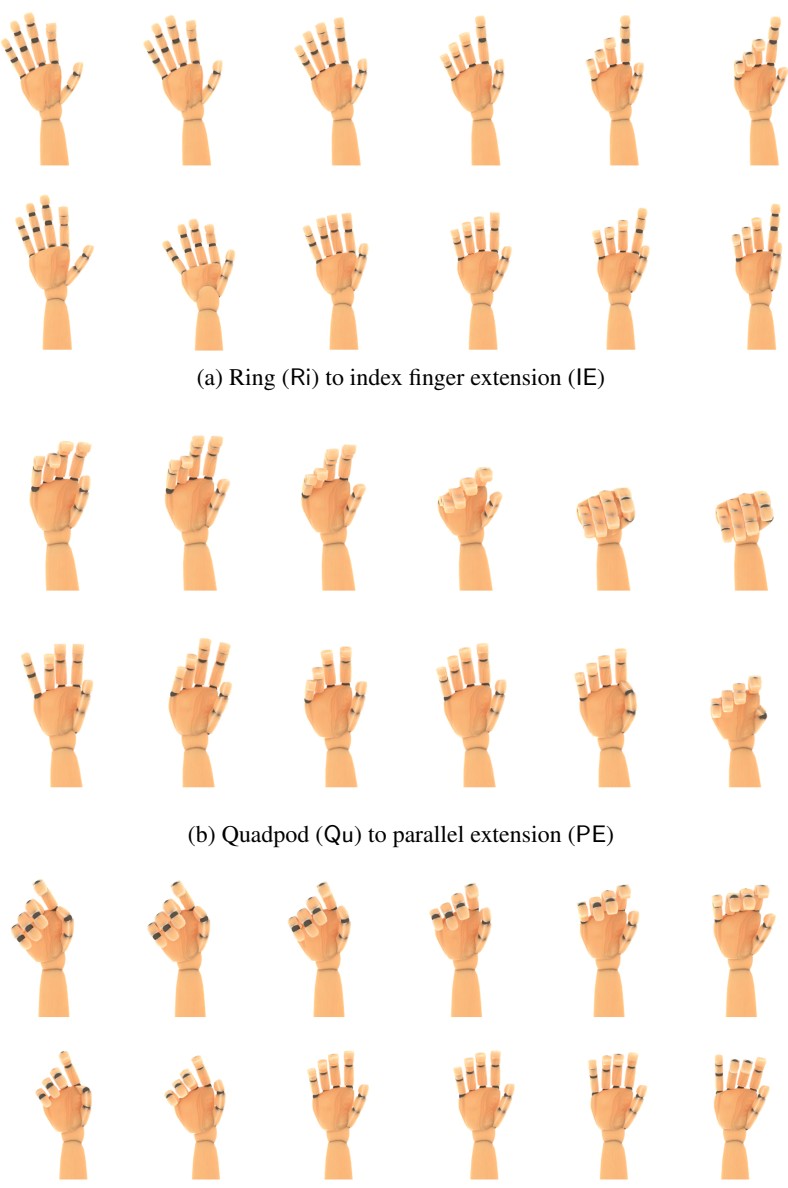

(a) Ring (Ri) to index finger extension (IE)

(b) Quadpod (Qu) to parallel extension (PE)

(c) Small diameter (SD) to tripod (Tr)

Figure 14: Generated motions for grasps. *Top:* Motions obtained via geodesic interpolation in the latent space of the back-constrained GPHLVM trained on the the hand grasp taxonomy (Fig. 3c). *Bottom:* Motions obtained via linear interpolation in the latent space of the corresponding back-constrained GPLVM.

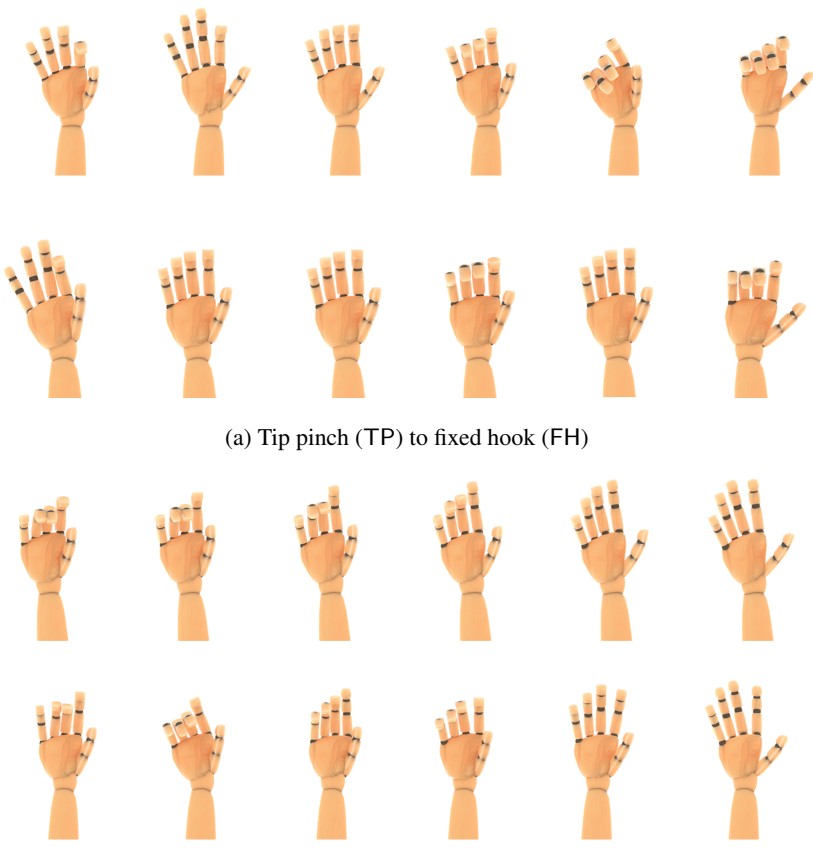

(a) Tip pinch (TP) to fixed hook (FH)

(b) Writing tripod (WT) to power disk (PD)

Figure 15: Generated motions for grasps. *Top:* Motions obtained via geodesic interpolation in the latent space of the back-constrained GPHLVM trained on the the hand grasp taxonomy (Fig. 3c). *Bottom:* Motions obtained via linear interpolation in the latent space of the corresponding back-constrained GPLVM.

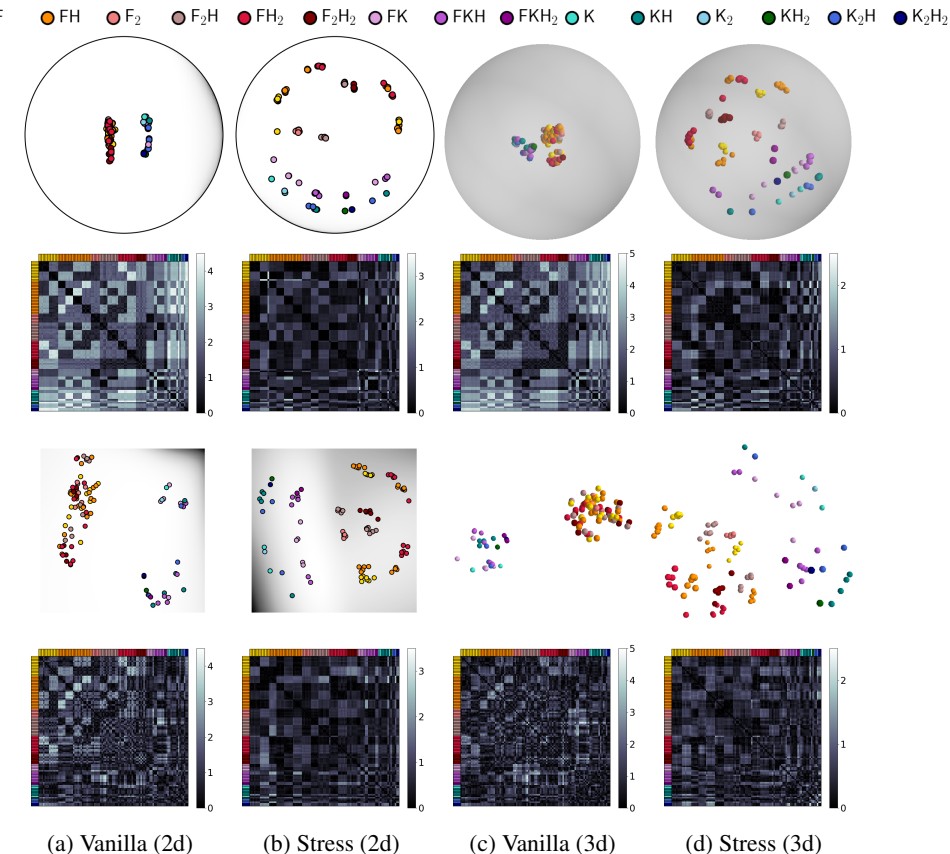

Figure 16: Embeddings of support poses with VAEs: The first and last two rows respectively show the latent embeddings of the hyperbolic and Euclidean VAE in $\mathcal{P}^Q$ and $\mathbb{R}^Q$, followed by pairwise error matrices.

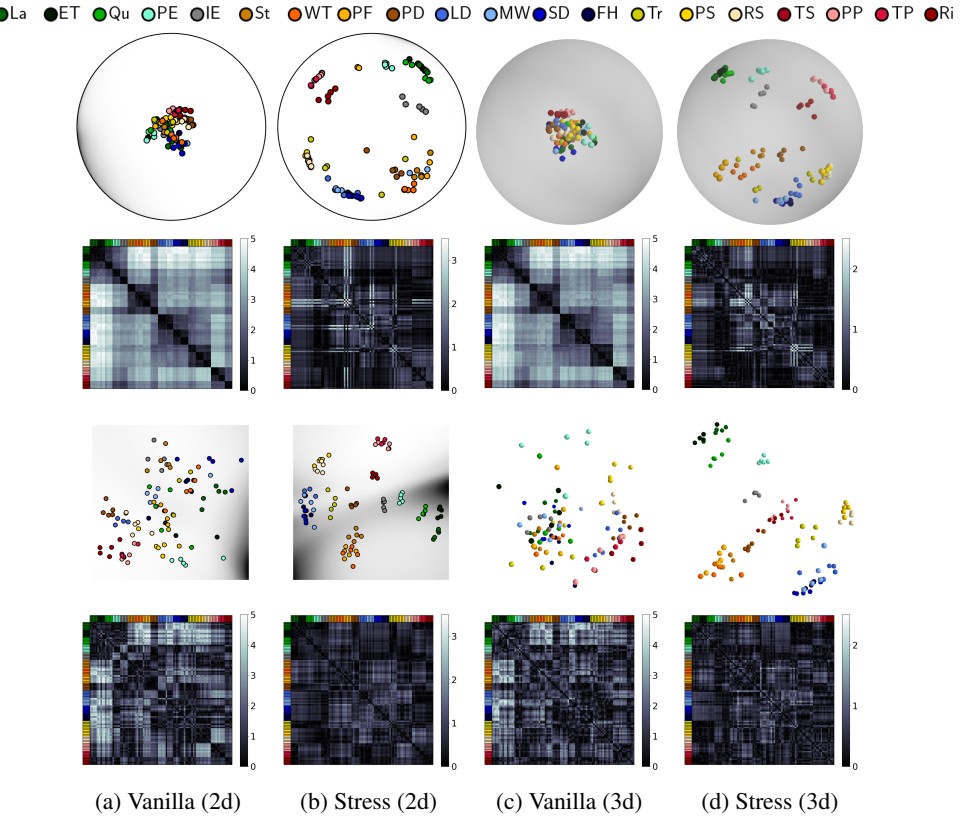

Figure 17: Embeddings of grasps with VAEs: The first and last two rows show the latent embeddings of the hyperbolic and Euclidean VAE in $\mathcal{P}^Q$ and $\mathbb{R}^Q$, followed by pairwise error matrices.

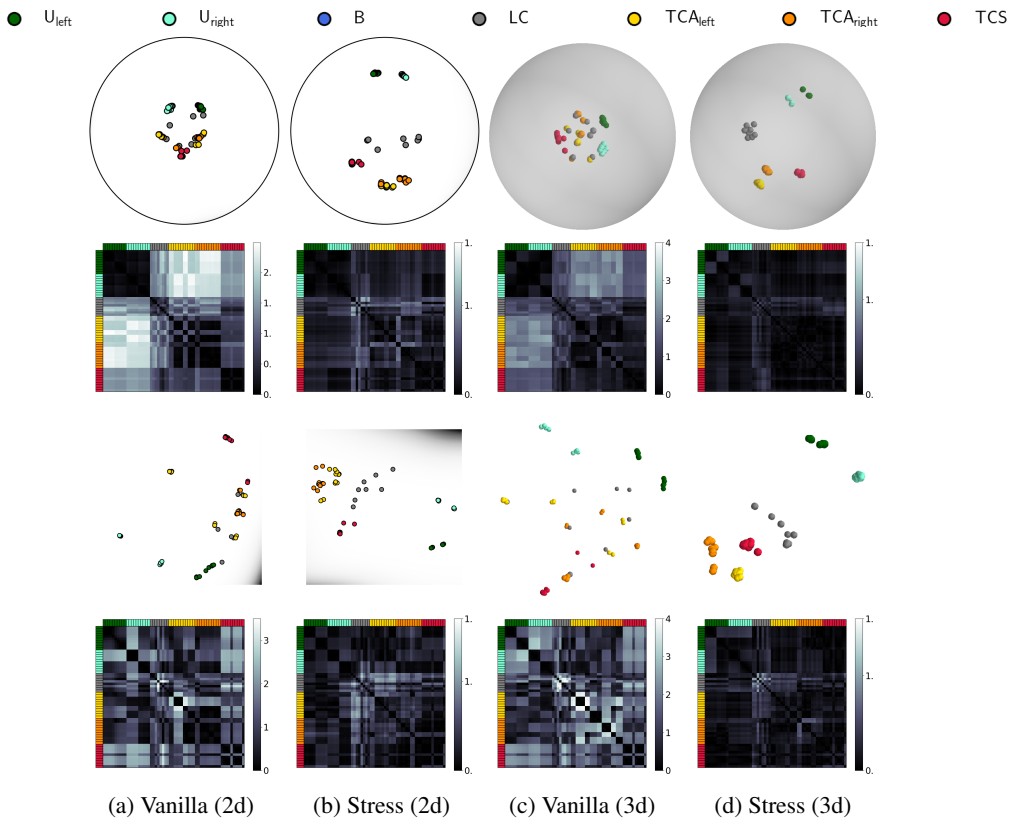

(a) Vanilla (2d)  (b) Stress (2d)  (c) Vanilla (3d)  (d) Stress (3d)

Figure 18: Embeddings of bimanual manipulation categories with VAEs: The first and last two rows show the latent embeddings of the hyperbolic and Euclidean VAE in $\mathcal{P}^Q$ and $\mathbb{R}^Q$, followed by pairwise error matrices between geodesic and taxonomy graph distances.

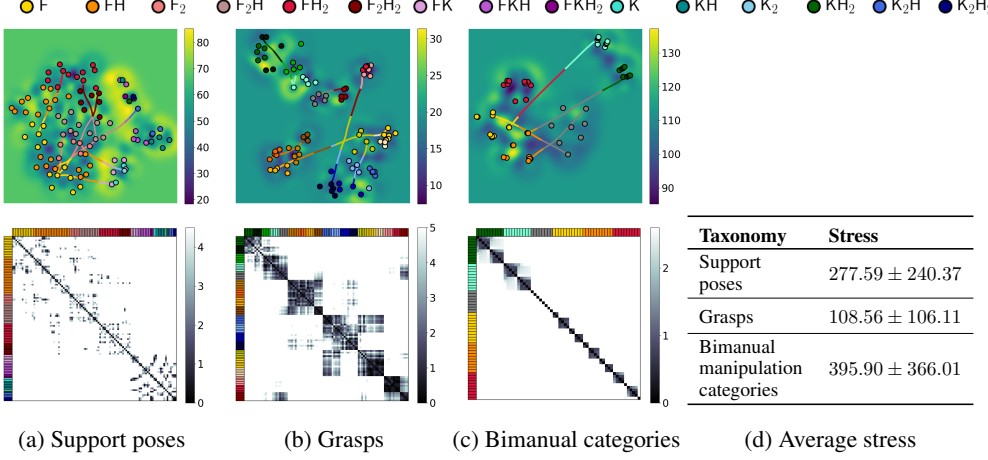

(a) Support poses  (b) Grasps  (c) Bimanual categories  (d) Average stress

Figure 19: Embeddings of taxonomy data on learned manifolds: The first row shows the latent spaces of the GPLVM. The background color is proportional to volume of the learned Riemannian metric. The second row displays the error matrix between the geodesic and taxonomy graph distances.

