# OpenReview forum: "Bringing robotics taxonomies to continuous domains via GPLVM on hyperbolic manifolds"
_ICLR.cc/2024/Conference — Submitted to ICLR 2024_

### Official Review · Reviewer_vvht · 2023-11-01

**Soundness:** 2 fair
**Presentation:** 2 fair
**Contribution:** 2 fair
**Rating:** 5
**Confidence:** 3

**Summary:**

The paper proposes to use hyperbolic embeddings to represent and utilize hierarchical taxonomies for whole humanoid body poses, 5 fingered hand grasps and bimanual manipulation. It introduces the Gaussian Process Hyperbolic Latent Variable Model (GPHLVM), extending Gaussian Process Latent Variable Models to hyperbolic spaces. By incorporating graph-based priors and distance-preserving back constraints, the model preserves the hierarchical structure of taxonomies. Experimental results on a whole-body support pose taxonomy demonstrate that hyperbolic embeddings outperform their Euclidean counterparts, enabling improved encoding of unseen poses and facilitating motion interpolation within the latent space.

**Strengths:**

1. The paper is well-detailed as the authors include extensive technical details in the main paper and appendix. The authors also  provide background information on GPLVMs, Riemannian geometry, the hyperbolic manifold, variational inference.
1. The experiments are shown for euclidean vs proposed hyperbolic spaces, as well as VAE and hyperbolic VAE. The figures show the embeddings of support poses for the whole body and hand grasp poses.  GPHLVM show the ability to encode the unseen class instance and unseen taxonomy nodes with lower stress value as compared to GPLVM. The model is shown to be suitable for trajectory generation by geodesic interpolation.
1. The motion generation results look promising and indicate that the learned representations work well.

**Weaknesses:**

1. The paper is not well situated with other works in human motion generation, for example, TEACH (https://arxiv.org/pdf/2209.04066.pdf), Human Motion Diffusion as a Generative Prior (https://arxiv.org/abs/2303.01418),
1. “this paper is the first to leverage the hyperbolic manifold for robotic applications” seems like over-claim in robotics for two reasons: (1) the proposed solution does not check collisions due to interpolation, and (2) GPHLVM has high training and decoding times in 3-d space (appendix reports >100 seconds for decoding).
1. The applicability and impact of the proposed solution is unclear. The readers can not judge the advantage of the proposed GPHLVM, as other methods may generate similar or better performances.

**Questions:**

1. The proposed taxonomies are graphs and there are existing approaches such as Hyperbolic Graph Convolutional Neural Networks (https://arxiv.org/pdf/1910.12933.pdf) discusses how hyperbolic spaces are well suited for representing tree-like structures instead of euclidean spaces. Is the approach using shallow embeddings? If yes, how are the listed shortcoming justified?
1. How does the work compare to neural network based geodesic interpolation approaches? For example, Geodesic Graph Neural Network for Efficient Graph Representation Learning, https://openreview.net/forum?id=6pC5OtP7eBx
1. While the limitations of the work are well-noted in the conclusion, can the learned robot taxonomies be applied in a simulator to test its collision aware interpolation?

---

> ### Author Response · Authors · 2023-11-19
> **Answer to Review (1)**
>
> Thank you very much for your time reviewing our work! We are glad that the reviewer appreciated our experiments and the fact that we included “extensive technical details” in our paper!
> Below, we address the concerns raised as part of the review.
>
> **Weaknesses:**
>
> 1. **Relationship to other works on human motion generation:**
>
>     Paper [1] tackles the problem of text-conditioned motion generation, where a sequence of human motions is generated given a sequence of textual prompts. This work uses a VAE trained on textual instructions and human motion frames. Shafir’s et al [2] addresses a similar problem as [1], where the main goal is human motion (sequential and parallel) composition to generate long-horizon human motion sequences using a text-conditioned human motion diffusion model (MDM). Our work significantly differs from [1,2] in that we do not train on full human motion trajectories and do not condition human motion generation on textual prompts. Instead, our model focuses on leveraging the taxonomies to better structure the learned embeddings and uses geodesic as a human motion generation mechanism between single human/grasp poses (i.e., no motion/grasping trajectories were observed/used). We believe that in our future work we may combine our hyperbolic taxonomy-based approach and textual prompts for driving the human motion generation.
>
>     We updated our paper in Section 6 and App. F.6 by including the above references and better position our paper w.r.t these text-conditioned human motion generation approaches. As discussed in the answer to Reviewer zvAH, we added comparisons  in App. F.6 and in the supplementary video of the trajectories obtained via geodesic interpolation in the latent space of the back-constrained GPHLVM trained on the support pose taxonomy with trajectories generated via linear interpolation in the latent space of VPoser [A] (see Figures 11-13. We observe that the motions generated by our approach are as realistic as the ones obtained from VPoser, despite the fact that VPoser is trained on full human motion trajectories and 1M datapoints. In contrast, our approach leverages the prior knowledge of the taxonomy.
>
>     **Reference:**
>
>     [A] Georgios Pavlakos, Vasileios Choutas, Nima Ghorbani, Timo Bolkart, Ahmed A. Osman, Dimitrios Tzionas, and Michael J. Black. Expressive body capture: 3d hands, face, and body from a single image. In Conf. on Computer Vision and Pattern Recognition (CVPR), pp. 10967–10977, 2019
>
> 2. **Over-claim:**
>
>     We agree with the reviewer that our current results are not yet showcased in a real robotic system. However, our approach may readily be used to train a taxonomy-aware GPHLVM with robotics data similarly as achieved with human motion data, and then to expand the taxonomy, encode unseen poses, and generate trajectories. Concerning the limitations raised by the reviewer:
>
>     (1) Check for collisions: It is true that the proposed solution does not check for collisions. To guarantee collision avoidance, we would consider the trajectory generated via geodesics as a target trajectory to be fed to a robotics controller which would robustly handle aspects such as self-collision avoidance, joint limits, etc. We would like to emphasize that the trajectories that we generated via geodesics in our experiments mostly did not display visually-apparent collisions, and were always realistic enough to be tracked accurately while avoiding self-collisions by such a controller. In this sense, the application of the generated trajectories is straightforward.
>
>     (2) High training and decoding times in 3-d space: The decoding time reported in App. F.7 is $10.34\pm1.05$ **milliseconds** for the GPHLVM with 3-dimensional latent space. We believe that this decoding time is reasonable for robotics applications. We acknowledge that the training time is longer, namely $>5$ seconds and $>100$ seconds for the GPHLVM with 3 and 2-d latent spaces, respectively. However, as the model requires to be trained once before being used, the training time also remain reasonable.
>
>     We modified our claim in the main paper as “*this paper is the first to leverage the hyperbolic manifold for robotic domains*” to account for the fact that we did not showcase our approach on a real robot.

---

> ### Author Response · Authors · 2023-11-19
> **Answer to Review (2)**
>
> 3. **Applicability and impact of the proposed solution:**
>
>     Concerning comparisons with other approaches, we added a discussion in Section 6 to better position our paper with respect to the state of the art in human motion generation. Namely:
>
>     - Other latent variable models (LVMs), such as VPoser [A], GAN-S [B], or TEACH [1] which are trained with thousands of datapoints, our method is trained on single human or hand poses.
>     - As the aforementioned LVMs are trained on full human motion trajectories, it is natural that they can retrieve human motions. In contrast, our model is not trained on motion data but leverages instead the robotic taxonomy and geodesic interpolation as a human/hand motion mechanism.
>     - The latent space of VPoser is of relatively high dimension (32). Our approach is able to generate motions from a 2-dimensional latent space.
>
>     Moreover, our paper included a comparison of our model against a vanilla VAE and a hyperbolic VAE in App. G.1, which in a way resemble what the VAE in VPoser may implement if trained only on single human/hand poses. As reported in Figs 16-18 and Table 9, our model outperforms its VAE counterparts.
>
>     Concerning the applicability of our approach, the incorporation of the taxonomy structure enables the use of the learned hyperbolic embeddings in a variety of downstream robotics applications, for instance:
>
>     - The proposed trajectory generation mechanism may be leveraged for human and robot motion generation. It is important to notice that realistic trajectories can be generated thanks to the incorporated taxonomy structure despite the fact that our training data only consist of discrete poses, i.e., our approach allows the generation of trajectories interpolating between two different nodes in the taxonomy, although no trajectories were used to train the model. In this sense, our approach leverages the taxonomy structure as inductive bias for a plausible data generation mechanism. This is of particular interest for:
>         1. Grasping and in-hand manipulation of objects, as grasping data are usually limited to transitions from an open hand to a specific grasp and do not contain transitions between different grasps.
>         2. Whole-body motion generation of humanoid robots, where the taxonomy may be leveraged to define a sequence of actions. For instance, a robot climbing stairs with a handrail will move according to a cycle in the whole-body support pose taxonomy (2 feet — 2 feet 1 hand — 1 foot 1 hand — 2 feet 1 hand). This knowledge may be combined with our GPHLVM to generate the corresponding joint trajectories in a principled manner.
>     - In Section 4, we demonstrate that our model allows the embedding of unseen taxonomy nodes and unseen poses in the GPHLVM latent space. This may also be leveraged, for instance, in grasping pipelines as new data, e.g., demonstrations involving grasping a previously-unseen object, may readily be incorporated into the latent space and classified according to the taxonomy.
>     - The taxonomy structure imposed in the latent space may be leveraged in (human) motion prediction tasks, e.g., in applications involving human-robot interactions. By starting a motion at a specific pose and moving along a certain direction, our model can predict that we may move towards a subset of poses according to a taxonomy.
>
>     We expanded Section 6 to discuss these downstream tasks.
>
>
>
>     **References:**
>
>     [A] Georgios Pavlakos, Vasileios Choutas, Nima Ghorbani, Timo Bolkart, Ahmed A. Osman, Dimitrios Tzionas, and Michael J. Black. Expressive body capture: 3d hands, face, and body from a single image. In Conf. on Computer Vision and Pattern Recognition (CVPR), pp. 10967–10977, 2019
>
>     [B] Andrey Davydov, Anastasia Remizova, Victor Constantin, Sina Honari, Mathieu Salzmann, and Pascal Fua. “Adversarial parametric pose prior”. In Conf. on Computer Vision and Pattern Recognition (CVPR), pp. 10987–10995, 2022

---

> ### Author Response · Authors · 2023-11-19
> **Answer to Review (3)**
>
> **Questions:**
>
> 1. >The proposed taxonomies are graphs and there are existing approaches such as Hyperbolic Graph Convolutional Neural Networks discusses how hyperbolic spaces are well suited for representing tree-like structures instead of euclidean spaces. Is the approach using shallow embeddings? If yes, how are the listed shortcoming justified?
>
>     If with "shallow embeddings" the reviewer refers to the dimensionality of the latent space, we would like to point out that we may use higher-dimensionality embeddings if necessary. However, such a model choice is not justified because:
>
>     1. We are not working with large dataset of human motion trajectories but instead we rely on a small dataset of single human/grasp poses (100 datapoints or less).
>     2. We believe that as our model is able to capture the data structure into a 2D hyperbolic space, there is no a practical reason to increase the embeddings dimensionality. Moreover, 2D and 3D embeddings can be easily visualzed, making any qualitative analysis easier.
>
>     Regarding hyperbolic GNNs, it is worth highlighting that vanilla GNNs are designed to work with graphs, that is, they are usually trained over large datasets of graphs, and their training implements several NN operations (e.g., pooling) tailored to graph data. In our setting, our data comes from a single type of graph. In other words, we are leveraging the taxonomy graph as structural inductive bias for the latent space embeddings (via distance-preserving regularization), and we exploit the hyperbolic geometry to properly accommodate such structure. Regarding the latter, the hyperbolic version of GNNs builds on the same reasoning as our model: learning low-distortion latent embeddings from observations characterized by hierarchical structures .
>
>     If with "listed shortcoming" the reviewer refers to the shortcomings of shallow embeddings listed in [1, Sec 1], we would like the emphasize that:
>
>     1. Our model can scale to larger dataset if required by using inducing points as summarized in Sec 3.2 and explained  in details in App. C.
>     2. We did not face any optimization problems when employing Riemannian optimization methods.
>     3. Our approach does consider the features of the taxonomy graphs for training. Indeed, the training data is composed of observed features (e.g., joint positions of the human hand) to train our GPHLVM.
>     4. We are unsure what "lack of inductive capabilities" refers to in the hyperbolic GNN paper.
>
>     In case we misunderstood this reviewer's comment, we would like to kindly ask the reviewer to elaborate it.
>
> 2. >How does the work compare to neural network based geodesic interpolation approaches?
>
> The geodesics used in the GDGNN of [3] are shortest paths on graphs. In other words, such geodesic are shortest paths from a given node to another in the graph. The method in [3] leverages these geodesics to gather additional information about shortest paths in graphs, which is then used to improve the performance of a GNN. In our approach, we instead use geodesics, i.e., shortest paths, in the latent hyperbolic space, which we decode via the GPHLVM mapping to obtain trajectories in joint space.
> Despite the fact that the geodesics on graphs as in [3] are conceptually similar to geodesics in hyperbolic spaces (both are shortest paths), the approaches have different goals. Similarly to GNNs, the GDGNN are designed to work with graphs, while our approach leverages a single taxonomy graph as inductive bias for learning taxonomy-aware latent space embeddings with low distortion.
>
> [3] Lecheng Kong, Yixin Chen, Muhan Zhang, “Geodesic Graph Neural Network for Efficient Graph Representation Learning”, NeurIPS 2022.
>
> 3. >Can the learned robot taxonomies be applied in a simulator to test its collision aware interpolation?
>
>     Yes, the trajectories generated via geodesics could be tested in a simulator to check potential collisions. As mentioned in our answer above, the trajectories generated via interpolations in the latent space of the GPHLVM usually did not display visually-apparent collisions. Moreover, in practice, such trajectories would be leveraged as target trajectories for a controller (e.g., using MPC or SQP-based control) which would handle joint limits and self-collision avoidance.

---

> > ### Comment · Reviewer_vvht · 2023-11-22
> > **Thanking Authors for detailed rebuttal**
> >
> > I would like to thank the authors for detailed answers to all the reviewers. While the proposed solution does not have visually-apparent collisions, it is unclear how to qualitatively compare it existing approaches.
> > I believe the paper's contribution will be clearer with a more systematic evaluation and I would maintain my borderline rating for now.

---

> ### Author Response · Authors · 2023-11-23
> **Answer to Reviewer vvht**
>
> Thank you very much for your feedback!
>
> Concerning your comment about the qualitative comparisons to existing approaches, we would like to emphasize that **we added qualitative comparisons with VPoser [A] in App. F.6 (Figures 11-13) and in the video available as supplementary material**. We observe that the motions generated by our approach look as realistic as the ones obtained from VPoser, thanks to the prior knowledge about the taxonomy which is leveraged as inductive bias in our approach.
>
> Concerning collisions, we acknowledged in Section 6 that our motion generation mechanism does not use explicit knowledge on how physically feasible the generated trajectories are and we will investigate this point as future work. However, the fact that the trajectories generated by our model mostly did not display visually-apparent collisions suggests potential for combining our model with explicit collision-avoidance mechanisms to generate physically-feasible motions. This may be achieved either (i) by considering the trajectory generated via geodesics as a target trajectory to be fed to a robotics controller which would robustly avoid self-collisions, or (ii) by adding an additional collision penalizer term to the training loss, similarly to SMPL-X [A] or HuMoR [B].
>
> Finally, we would like to emphasize that **our proposed motion generation mechanism stands as only one of our contributions** and would like to highlight the significance of the other contributions of our paper, namely:
>
> - We propose a Gaussian process hyperbolic latent variable model, whose latent space is well suited to embed continuous data associated to trees and graphs;
> - We propose to introduce inductive bias in the form graph-based priors and graph-distance-preserving back constraints to enforce taxonomy graph structure in the GPHDM latent space. Embedding the taxonomy structure in latent spaces is beneficial for a variety of downstream tasks (including motion generation), as shown in our experiments and discussed in Section 6 of our paper and in our answers to the Reviewers.
>
> References:
>
> [A] Georgios Pavlakos, Vasileios Choutas, Nima Ghorbani, Timo Bolkart, Ahmed A. Osman, Dimitrios Tzionas, and Michael J. Black. Expressive body capture: 3d hands, face, and body from a single image. In Conf. on Computer Vision and Pattern Recognition (CVPR), pp. 10967–10977, 2019
>
> [B] Davis Rempe, Tolga Birdal, Aaron Hertzmann, Jimei Yang, Srinath Sridhar, and Leonidas J. Guibas. HuMoR: 3D Human Motion Model for Robust Pose Estimation. in ICCV 2021.

---

### Official Review · Reviewer_zvAH · 2023-11-02

**Soundness:** 3 good
**Presentation:** 2 fair
**Contribution:** 2 fair
**Rating:** 6
**Confidence:** 3

**Summary:**

This work proposes to use GPHLVM (Gaussian Process Hyperbolic Latent Variable Model) to model robot taxonomy data. The learned GPHLVM, enhanced by the proposed graph-distance priors and back constraints, can learn geometric structures similar to the hierarchical taxonomy using very low-dimensional latent space. Experiments on three different robotics taxonomy data (full body pose with support, hand grasping, and bimanual manipulation) show that the proposed latent space can learn semantically rich latent space that can be applied to quantify new class/pose and be used for interpolation.

**Strengths:**

- The proposed formulation seems suitable for the taxonomy learning task and has proposed a well-principled solution for this task. Using the hyperbolic latent space’s inherent geometric structure to embed hierarchical concepts.
- While I am not an expert in this space, the proposed GPHLVM, the formulation, and the proposed priors are sound. Embedding high-dimensional pose data in a 2D latent space is quite impressive.
- Results shown on three different taxonomy embedding show that the proposed approach learns better structure than Euclidian counterparts, and embeds the taxonomy data well. It can also handle unseen pose and class, and embed them into semantically meaningful places.

**Weaknesses:**

- I feel like this paper lacks a clear motivation for the proposed method. “Robotic application” is alluded to, but no concrete robotic application is demonstrated. How does embedding the taxonomy in this latent space help downstream tasks? Some intuition or concrete examples would be really helpful. The introduction mentioned taxonomy hasn’t been widely adopted, but how does the proposed method help in that matter?
- Missing comparison with some of the popular human pose latent space like VPoser [1] or adversarial poser [2]. While they are from slightly different domains and only learned from data, they can show very similar interpolation capabilities. I can imagine an analysis of these could really help show the importance of the proposed geometric priors.
- Figures 2 and 4 are very hard to parse. The “comply with taxonomy graph structure” isn’t very obvious from the plot, as well “geodesic distance match graph nodes”. Instead of dumping everything into one big graph, it would be very beneficial if a selected few sample cases (like e $K_2H_2$ and $FH$ could be picked out as examples to better elaborate on the case. Also, put Figures 2 and 4 together 3 to indicate GT.
    - Also from Figures 2 and 4, it appears that performance on $P^2$ and $R^2$ are similar visually, and adding more prior during optimization seems to be the key to achieving good results. Then how important is the hyperbolic latent space for downstream tasks? While hyperbolic space achieves better stress, all models’s stress improves with more regularization. So how important, from a downstream task’s point of view, is the hyperbolic space?

[1] Pavlakos, Georgios, Vasileios Choutas, Nima Ghorbani, Timo Bolkart, Ahmed A. A. Osman, Dimitrios Tzionas, and Michael J. Black. 2019. “Expressive Body Capture: 3D Hands, Face, and Body from a Single Image.” *Proceedings of the IEEE Computer Society Conference on Computer Vision and Pattern Recognition* 2019-June: 10967–77.

[2] Davydov, Andrey, Anastasia Remizova, Victor Constantin, Sina Honari, Mathieu Salzmann, and Pascal Fua. 2021. “Adversarial Parametric Pose Prior.” *arXiv [Cs.CV]*, December. https://doi.org/10.48550/ARXIV.2112.04203.

**Questions:**

- What is the computation cost for optimizing thee GPHLVM model? Is the sampling strategy expensive?
- How well can the proposed approach scale to a larger dataset? I can imagine a 2-D latent space would not be able to capture the broad range of human motion.

---

> ### Author Response · Authors · 2023-11-19
> **Answer to Review (1)**
>
> Thank you very much for your time reviewing our work! We are glad that the reviewer found that our paper proposes “a well-principled solution” for embedding taxonomy and that “embedding high-dimensional pose data in a 2D latent space [was] quite impressive”!
>
> **Motivation and adoption of robotics taxonomies:**
>
> Some examples of downstream tasks which would benefit from our embedding the taxonomy in latent spaces are described below.
>
> 1. Our approach may be leveraged in a grasping pipeline, where the proposed trajectory generation mechanism may be used to reproduce trajectories on a robotic hand to grasp different types of objects and thus requiring different grasp types. Notice that such realistic trajectories can be generated due to the incorporated taxonomy structure despite the fact that our training data only consist of discrete poses (no trajectories were used as training data). In other words, our approach allows the generation of trajectories interpolating between two different nodes in the taxonomy, although no trajectories were used to train the model. In this sense, our approach leverage the taxonomy structure as inductive bias for a plausible data generation mechanism. This is of particular interest for:
>     1. Grasping and in-hand manipulation of objects, as grasping data are usually limited to transitions from an open hand to a specific grasp and do not contain transitions between different grasps.
>     2. Whole-body motion generation of humanoid robots, where the taxonomy may be leveraged to define a sequence of actions. For instance, a robot climbing stairs with a handrail will move according to a cycle in the whole-body support pose taxonomy (2 feet — 2 feet 1 hand — 1 foot 1 hand — 2 feet 1 hand). This knowledge may be combined with our GPHLVM to generate the corresponding joint trajectories in a principled manner.
> 2. In Section 4, we demonstrate that our model allows the embedding of unseen taxonomy nodes and unseen poses in the GPHLVM latent space. This may also be leveraged, for instance, in grasping pipelines as new data, e.g., demonstrations involving grasping a previously-unseen object, may readily be incorporated into the latent space and classified according to the taxonomy. The new grasping pose can then be reproduced and used in the generation of new interpolated trajectories.
> 3. In a similar line of thought, our approach may be leveraged for applications related to hand motions when interpreting sign languages. Previously-unseen signs may be embedded in the GPHLVM latent space, and our trajectory generation mechanism may be used to interpolate between signs.
> 4. Finally, the taxonomy structure imposed in the latent space may be leveraged in (human) motion prediction tasks, e.g., in applications involving human-robot interactions. By starting a motion at a specific pose and moving along a certain direction, our model may predict that we may move towards a subset of poses  by leveraging the taxonomy knowledge.
>
> Finally, we would like to emphasize an important experimental insight: leveraging the taxonomy along with the hyperbolic manifold allowed our model to learn embeddings that closely follow the associated taxonomy graph. Without it, as observed in Fig. 3, tasks such as pose classification or motion interpolation often failed.
>
> We expanded Section 6 to discuss the aforementioned downstream applications as follows:
>
> “*The learned hyperbolic embeddings may be used in a variety of downstream robotics applications, such as generation of hard-to-collect data (e.g., transitions between grasps types), taxonomy-aware whole-body motion generation, motion prediction, or pose classification*.”

---

> ### Author Response · Authors · 2023-11-19
> **Answer to Review (2)**
>
> **Comparisons with popular human pose latent space:**
>
> Paper [1] addressed the problem of building a full 3D model of human gestures by learning a deep NN that jointly models the human body, face and hands from RBG images. In other words, their learning problem is a mapping from human poses, shape and facial expression parameters (all extracted from RBG images) to vertices that parametrize the 3D human body model. We believe this work tackles a radically different learning problem than our approach. Nevertheless, as the reviewer pointed out, a small part of the pipeline of that paper includes a body pose prior trained using a VAE on MoCap data [1, Sec 3.3], called VPoser. In this regard, we would like to point out that:
>
> - VPoser is trained on full human motion trajectories and 1M data (AMASS dataset including 40 hours of motion data), while our method is trained on single human/hand poses.
> - As VPoser is trained on full human motion trajectories, it is natural that it can retrieve human motions. In contrast, our model **is not** trained on motion data but leverages instead the robotic taxonomy and geodesic interpolation as a human/hand motion mechanism.
> - The latent space of VPoser is of relatively high dimension (32). Our approach is able to generate motions from a 2-dimensional latent space.
> - Our paper included a comparison of our model against a vanilla VAE and a hyperbolic VAE in App. G.1, which in a way resemble what the VAE in VPoser may implement if trained only on single human/hand poses. As reported in Figs. 16-18 and Table 9, our model outperforms its VAE counterparts.
>
> Paper [2] tackles the same problem as in [1] but the pose prior is learned via GANs instead of VAEs as in VPoser. Similarly as in [1], the prior model is trained over full human motion trajectories. Therefore, we believe this work is also very different from what we propose in this paper.
>
> Finally, besides the VAE experiments reported in App. G.1, we would like to emphasize that we carried out experiments that included geometry-unaware models such those based on the vanilla GPLVM in order to show the importance of having embeddings whose underlying geometry matches better the geometry of the observations. Our experiments provided evidence that such geometric priors are indeed relevant.
>
> We updated our paper in Section 6 by including the above references and better position our paper w.r.t the state of the art. Moreover, we updated App. F.6 and the supplementary video to include comparisons of the trajectories obtained via geodesic interpolation in the latent space of the back-constrained GPHLVM trained on the support pose taxonomy with trajectories generated via linear interpolation in the latent space of VPoser (see Figures 11-13). We observe that the motions generated by our approach are as realistic as the ones obtained from VPoser, despite the fact that VPoser is trained on full human motion trajectories and 1M datapoints. In contrast, our approach leverages the prior knowledge of the taxonomy.

---

> ### Author Response · Authors · 2023-11-19
> **Answer to Review (3)**
>
> **Figures 2 and 4 hard to parse:**
>
> We modified Figures 2 and 4 (now Figure 3) to display the error between the latent space geodesic distances and the taxonomy graph distances (instead of the latent space geodesic distances as previously), so that (1) the parsing of the figures is facilitated and (2) the difference of performance between the hyperbolic and Euclidean models is more visible. The modified figures clearly show that the hyperbolic model better comply with taxonomy graph structure, i.e., that geodesic distances match the graph distances: The hyperbolic models generally display lower errors (depicted in dark tones) than the Euclidean models, which feature large regions with high errors (light tones).
>
> We removed Figure 3, as it became unnecessary to interpret Figures 2 and 4, and updated Figures in App. E and F.
>
> **Difference of performance between** $\mathcal{P}^2$ and $\mathbb{R}^2$ **in Figures 2 and 4:**
>
> As hinted by the reviewer and as explained in Section 4, the graph-based priors (in the form of the back-constraints and of the stress loss) are key to enforce the taxonomy graph structure in the learned embeddings. Without such priors, both GPLVM and GPHLVM organize the embeddings only as a function of their relationship in the original data space, as explained in Section 5. Although both GPLVM and GPHLVM models improve with regularization, the hyperbolic models comply better with the taxonomy graph structure as shown by the average stress values displayed in Table 1. This is due to the fact that the geometry of the hyperbolic manifold provides an increased volume to match the graph structure: As theoretically shown by Sarkar, 2011 [A], trees can be embedded with arbitrarily low distortion — in other words, arbitrarily low mistmatch between the graph and hyperbolic distances or arbitrarily low stress — into the 2D hyperbolic space, while this cannot be achieved in Euclidean space. As the grasps and bimanual manipulation taxonomies are trees, they can be embedded with arbitrarily low distortion in hyperbolic spaces, but not in Euclidean spaces. It is interesting to observe that, despite its non-tree structure, the support pose taxonomy is also better embedded in a 2-D hyperbolic space than in a 2-d Euclidean space. Moreover, it is important to emphasize that the stress values displayed in Table 1 are close to the minimum stress values that can be achieved when embedding the different taxonomy graphs in hyperbolic and Euclidean low-dimensional spaces. In other words, increasing the stress loss in both models will not lead to lower stress values without obtaining degenerated models, i.e., models that would not be able to reconstruct the observations in the original data spaces.
>
> Finally, we refer the readers to the above answer on “**Motivation and adoption of robotics taxonomies**” for the importance of latent spaces complying with the taxonomy structure, i.e., for the importance of the hyperbolic latent space, in downstream tasks.
>
> Reference:
>
> [A] R Sarkar, “Low Distortion Delaunay Embedding of Trees in Hyperbolic Plane”, in International Symposium on Graph Drawing, 2011.
>
> **Computational cost:**
>
> The computational cost of our approach is analyzed in App. F.7, Table 9, where we display the average runtime for the training and decoding phases of the GPHLVM and GPLVM using 2 and 3-dimensional latent spaces. The main computational burden arise in the GPHLVM with a 2-dimensional latent space and sharply contrasts with the GPHLVM with a 3-dimensional latent space. As hinted by the reviewer, the main cause of the increase of computational cost is due to 2-dimensional hyperbolic kernel, which requires sampling for its computation.

---

> ### Author Response · Authors · 2023-11-19
> **Answer to Review (4)**
>
> **Scaling to larger datasets:**
>
> In the case of larger datasets, the main limitations of the GPHLVM are the same as for the classical GPLVM, that is, that the time complexity of Gaussian process (GP) models scales as $O(n^3 )$ and the storage as $O(n^2 )$ with n is the number of training examples. This is resolved by training the proposed GPHLVM via variational inference by maximizing a lower bound of the marginal likelihood of the data, akin to the Euclidean Bayesian GPLVM introduced by Titsias & Lawrence, 2010. In this case, the training parameters of the model comprise a set of $m$ inducing inputs, where $m<<n$, instead of the entire set of $n$ latent variables. In this sense, scaling the GPHLVM to larger datasets is no different from scaling GPLVM to such datasets, except that it must take into account the geometry of the hyperbolic latent space, as detailed in Section 3.2.
>
> We would also like to highlight that the data used in our experiments already cover a wide range of human motion. In particular, the data from the whole-body support pose taxonomy feature a large range of standing and kneeling poses with different arm and leg postures. Moreover, remember that distances grow exponentially in the hyperbolic manifold when moving away from the origin, leading to a high volume available away from the origin. This volume is increased compared to the volume available in 2-d Euclidean spaces. By taking this property into account, one can see that a large volume remains available away from the origin in the 2-d hyperbolic space of Figures 2 and 3 to embedd additional poses.

---

> ### Comment · Reviewer_zvAH · 2023-11-22
> **Response to Author Answer**
>
> Thanks a ton for the detailed response.
>
> In this particular part (response 2), I would like to point out that both VPoser and adversarial poser are **single-frame** pose models that do not involve sequences of poses (motion). There is a variant VPoser-t that deals with sequence of poses (motion). In that sense, both VPoser and adversarial poser are tackling the same problem as the proposed method: compressing a single frame of motion into latent space but scaling up to more data (which the proposed method might have trouble in).
>
> Overall, I think my questions are answered though some concerns remain (applicability and scalability). I remain positive about this work.

---

> > ### Author Response · Authors · 2023-11-22
> > **Answer to Reviewer zvAH**
> >
> > Thank you very much for your positive feedback!
> >
> > Regarding your comment about VPoser, we agree that VPoser does not consider explicitly that training data are full human trajectories during training and we apologize if our reply was misleading in this regard. However, when interpolating between two embeddings in the latent space of VPoser, it is very likely that single motion frames, similar to those produced along the interpolation path transitions, were observed in the training dataset due to the high amount of dataipoint VPoser is trained on and the fact that this dataset is composed of full motion trajectories (approx. 1M datapoints). Therefore, it is interesting that our geodesic interpolation between two poses in the latent space of the GPHLVM allows us to generate such transitions despite that they were entirely absent in our training dataset. In other words, the domain knowledge encapsulated in the taxonomies is leveraged by GPHLVM to learn taxonomy-aware latent embeddings that bias our geodesic motion generation, alleviating the need to use large human motion datasets (approx. 100 datapoints for GPHLVM).
> >
> > Concerning the scalability of our model, we have trained Bayesian GPHLVMs with ≥ 1000 datapoints. Such examples will be released along with the code source accompanying our paper upon acceptance. We would also like to point out that Euclidean Bayesian GPLVM and its extension based on stochastic variational inference [1] have been shown to scale to *thousands and millions of data points*, respectively.  Therefore, we expect our model to also scale to several thousands of datapoints.
> >
> > [1] Lalchand, V., Ravuri, A., Dann, E., Kumasaka, N., Sumanaweera, D., Lindeboom, R.G.H., Madad, S., Teichmann, S. & Lawrence, N.D.. (2022). Modelling Technical and Biological Effects in scRNA-seq data with Scalable GPLVMs. Proceedings of the 17th Machine Learning in Computational Biology meeting 200:46-60

---

> > > ### Comment · Reviewer_zvAH · 2023-11-23
> > > **Thanks for the detailed response**
> > >
> > > The reviewer thanks the authors for the additional response.
> > >
> > > I think my issues with scalability is now cleared with the additional experiments. I would support accepting this work.

---

> > > > ### Author Response · Authors · 2023-11-23
> > > > **Response to Reviewer zvAH**
> > > >
> > > > Thank you very much for your positive feedback! We are glad that the issue with scalability is now cleared.
> > > > Thank you also for supporting the acceptance of our paper! We would be grateful if the review and assessment of our paper may be updated accordingly.

---

### Official Review · Reviewer_gni3 · 2023-11-06

**Soundness:** 3 good
**Presentation:** 3 good
**Contribution:** 3 good
**Rating:** 6
**Confidence:** 4

**Summary:**

This paper proposes to embed robot taxonomy into a hyperbolic latent space, and as such, distances on taxonomies along with kernels can be defined. This allows one to construct a Gaussian Process latent variable model (GPLVM) over the taxonomy. The paper additionally proposes to construct a graph distance-based regulariser, encouraging the distances of the embeddings to match the graph distance. The paper then provides empirical results on how well the model is able to capture the structure of the taxonomy, and then on taxonomy expansion and trajectory generation (as geodesics of the latent space).

The paper is in general well-written, the method appears sound. My concerns are as above:
1. More clarity is needed in how the embeddings are constructed. A lot of detail has gone into how to construct the kernel itself -- but it's unclear to me how you obtained the hyperbolic latent variables. More clarity here would be appreciated.

2. How is the uncertainty of the GP used? A GP in the latent space is defined using the hyperbolic kernel, but there seems to be no use of the uncertainty from the GP. Some elaboration on this would be appreciated.

**Strengths:**

See above

**Weaknesses:**

See above

**Questions:**

See above

---

> ### Author Response · Authors · 2023-11-19
> **Answer to Review**
>
> 1. **Construction of the embeddings:**
>
>     In the GPLVM framework, the embeddings, or latent variables $\mathcal{X}=$ {$\mathbb{x}_n$}, $n=1..N$ are viewed as hyperparameters of the model. Therefore, the embeddings are obtained along the other hyperparameters $\mathbb{\Theta}$ of the model (mean and kernel parameters, as well as likelihood noise) during training. Training is typically achieved by maximizing the log-posterior of the model $\mathcal{L}_M$$_A$$_P$ or, in the Bayesian setting, by maximizing the marginal likelihood of the data data $\mathcal{L}_M$$_a$$_L$, which we approximate via variational inference. In both cases, the resulting optimization problem is of the form:
>
>     $$ argmax_\mathcal{X, \Theta}  \mathcal{L}(\mathcal{X}, \mathbb{\Theta} ) $$
>
>     In the case of the GPHLVM, special care must be taken during optimization to account for that the embeddings belong to the hyperbolic manifold. To do so, we leverage Riemannian optimization to solve the optimization problem described above such that $\mathbb{x}_n\in\mathbb{H}^Q$  $\forall  n$, as explained in the last paragraph of Section 3.2. Moreover, in the Bayesian case, we adapted the approximation via variation inference to hyperbolic latent spaces, as detailed in Section 3.2 and in App. C.
>
>     We clarified these points and added the optimization (1) at the beginning of Section 3.2. In case more clarity is required, we would like to kindly ask the reviewer to point out which aspects of the construction of the embeddings remain unclear, and we would gladly elaborate on them.
>
>
> 2. **Uncertainty of the GP:**
>
>     As of now, we used the uncertainty of the GP as a mean to analyse the certainty, and thus the quality, of the latent embeddings. The GPHLVM and GPLVM uncertainty are depicted in the first and third rows of Figures 2 and 3 (previously 4) in the main text, and Figures 9-12 in App. F.
>     For instance, the uncertainty depicted in Figure 3c-top for the GPHLVM indicates that the model is certain about the mapping to the observation space of the latent embeddings and in the regions around them. Therefore, we can trust that the geodesic interpolations that pass through rather certain regions (e.g., Ri to IE or ET to PE) will result in meaningful trajectories in the observation space. Interpolations that pass through an uncertain region (e.g., in the middle TP to FH) may instead revert to the GP mean in the observation space in this region. Interestingly, the Euclidean model of Figure 3c-bottom present much more uncertain regions and the model is precise only very locally around the embeddings. This indicates that most of the interpolation trajectories will simply follow the GP mean.
>
>     Related to the above, we think that an interesting usage of the uncertainty may be to explicitly leverage it to design interpolations that generally stay in certain regions. We believe that this is an interesting extension for uncertainty-aware interpolation methods and we may consider it as future work.

---

### Official Review · Reviewer_bere · 2023-11-07

**Soundness:** 4 excellent
**Presentation:** 3 good
**Contribution:** 3 good
**Rating:** 8
**Confidence:** 4

**Summary:**

The paper introduces a method that exploits the properties of hyperbolic space to suitably embed taxonomy data for robotic applications into a latent space. Specifically, taxonomies are modeled via a Gaussian process hyperbolic latent variable model (GPHLVM), an adaption of GPLVM to a latent space with hyperbolic geometry. The modeling of the various components of GPHLVM is presented in detail, drawing comparisons to the basic GPLVM model, while optimization of the proposed model is also covered. A combination of back-constraints and embedding priors is used to maintain the graph-distance information while embedding taxonomies into the hyperbolic latent space.

### Comments after rebuttal
I thank the authors for their detailed answers that have largely addressed my concerns. I greatly appreciate the addition of the trajectories generated by the competing methods. I think that the regularized GPLVM manages to produce sufficiently good results, although the proposed GPHLVM certainly shows an advantage. In any case, I think that the overall contribution is important and that the revised version of the manuscript has become stronger thanks to the reviewers' comments, hence I propose the acceptance of this work.

**Strengths:**

The paper, besides its heavy theoretical content related to the adaptation of GPLVM to a hyperbolic latent space, is well written and the presentation quality is high. It even manages to do a  good job at making the content available to an audience not very familiar with GPLVMs and Riemannian geometry, in spite of the tight page limitation. As stated in the text, in comparison to widely used solutions based on deep neural network architectures, the use of constrained statistical models like GPLVMs and GPHLVMs offers much higher data efficient while providing uncertainty quantification at the same time. The developed GPHLVM model is evaluated across three distinct types of human motion taxonomies, namely contact-aware whole-body pose sequences, hand-grasps and bimanual manipulation motions. The evaluation shows that the proposed model is able to better embed these taxonomies into a common latent space, while preserving the corresponding graph distances.

Overall, the paper proposes a strong theoretical contribution with the introduction of the GPHLVM model, that can be applied for embedding hierarchical and graph-like structures.

**Weaknesses:**

There are many choices involved regarding the design of the final model, as for example the different types of kernels that can be used both for the basic model and the back-constraints. These provide liberty regarding the application of the model, but also introduce a high number of hyperparameters.

Another weakness of the paper regards the presentation, as a large part of important material is provided outside of the main text, e.g. the comparison to VAE. This is understandable due to the tight page limitation. In fact, it seems that the paper would benefit from a having more pages available, I would better see it as a journal paper.

An important issue is the difference between the performance of the proposed model with respect to the base GPLVM model in the case of the constrained (non-vanilla) versions. It is difficult to judge if the improvement is significant or not. It seems that the introduction of distance preserving constraints greatly reduces the theoretical advantage of the hyperbolic structure of the latent space. To show if the differences in the stress is significant it should be made clear to what difference do they correspond in the observation space, either quantitatively or qualitatively. In fact, a comparison of the trajectories generated from GPHLVM and GPLVM should be included (no GPLVM based generation is provided, if I am not mistaken).

Regarding the comparison with the baseline model, for fairness, the comparison should account for the difference in efficiency and computational complexity, which is quite significant as reported in Section F7. Regarding this comparison, please note that it would be preferable to move it to the main text, if possible.

**Questions:**

- Are there other, even simple, applications involving hierarchical and graph-based structures that the proposed framework could be easily applied to? It would be interesting to include some toy-like problem with synthetic data to better demonstrate the power and the possible limitation of the proposed model in representing such data.

---

> ### Author Response · Authors · 2023-11-19
> **Answer to Review (1)**
>
> Thank you for taking the time to review our work! We are delighted to read that our paper proposes a "strong theoretical contribution [...] that can be applied for embedding hierarchical and graph-like structures"!
> Below, we address the key concerns raised as part of the review.
>
> **Many design choices and high number of hyperparameters:**
>
> We agree with the reviewers that our GPHLVM offers various design choices. We would like to emphasize that this is a characteristic of all (back-constrained) GPLVMs, where different types of kernels can be used for the basic model and the back-constraints. Moreover, as for other GPVLMs, these design choices can generally be driven, and thus facilitated, by the prior knowledge of the problem at hand. In the case of the GPHLVM, the choices of kernel for the basic model and graph kernel in the back-constraints mapping are additionally limited by the availability of hyperbolic kernels and graph kernels in the literature (only SE and Matérn kernels in both cases). The observation kernel in the back-constraints can then be chosen according to prior knowledge of the observation data.
>
> In practice, only the hyperparameters of the back-constraints mappings and the loss scale must be chosen by the user, as other hyperparameters are optimized during the GPHLVM training. In our experiments, this amounts to only 4 hyperparameters (the observation kernel lengthscale, the graph kernel lengthscale, the product-of-kernel variance, and the loss scale), which remains reasonable in our opinion.
>
> We would like to emphasize that the fact that these design choices can be driven by prior knowledge is an advantage of statistical models such as GPLVMs and GPHLVMs compared to deep neural-network-based models, where the design of the underlying networks (number of layers, neurons, and activation functions) can hardly be driven by such prior knowledge while still influencing their performance.
>
> **Presentation and material in Appendices:**
>
> We agree with the reviewer that our paper would benefit from having more pages available. We believe that this is a common problem for many papers submitted and published in machine learning venues such as NeurIPS, ICML, ICLR, among others.
> We did our best to provide the core of our method and experiments in the main text, while providing additional derivations, details, and experiments in appendices. We are glad that the reviewer still found the presentation of our paper of high quality and that our paper did "a good job at making the content available to an audience not very familiar with GPLVMs and Riemannian geometry in spite of the tight page limitation".
>
> Due to the novelty of our idea of embedding taxonomy structures in hyperbolic latent spaces of GPLVM, we believe that it remains well suited to be published as a conference paper. We are planning a journal extension of this work, where the GPHLVM additionally include physics constraints and explicit contact data, and where we plan to showcase various downstream applications of our model.

---

> ### Author Response · Authors · 2023-11-19
> **Answer to Review (2)**
>
> **Difference with respect to the base GPLVM model:**
>
> To better illustrate the significance of the improvement of the GPHLVM over the GPLVM, we updated our paper as follows:
>
> 1. We modified the second and fourth rows of Figures 2 and 3 (previously Figures 2 and 4), as well as the Figures in App. E and F to display the error between the geodesic distances and the graph distances (instead of the geodesic distances as previously), thus making the difference of performance between the GPHLVM and GPLVM more visible. The updated figures clearly show that the hyperbolic models generally display lower errors (depicted in dark tones) than the Euclidean models, which feature large regions with high errors (light tones).
> 2. As suggested by the reviewer, we added a comparison of the trajectories generated by the back-constrained GPHLVM and GPLVM in App. F.6. For the support poses taxonomy, motions obtained via linear interpolation between two embeddings in the Euclidean latent space of the GPLVM look less realistic than motions obtained via geodesic interpolation in the latent space of the GPHLVM. In particular, the kneeling poses obtained from the GPLVM often look unnatural (see Figs. 11 and 13), and wavering wrist angles are often observed during these motions (see Figs. 11 and 12).
> In the case of the grasping taxonomy, motions obtained via linear interpolation between two embeddings in the Euclidean latent space of the GPLVM are also less realistic than those obtained for the GPHLVM. They display less regular interpolation patterns (see Fig. 14c and are often noisy, featuring wavering wrist or finger motions (see Figs. 14a, 14b, and 15b). Moreover, the generated grasps reflect less accurately the taxonomy categories (see, e.g., the parallel extension grasp in Fig. 14b or the tip pinch grasp of Fig. 15a).
>
> This shows that the improvement of the GPHLVM over the GPLVM, i.e, the differences in the stress values, are significant for the generated trajectories.
>
> We would like to emphasize that this improvement is also significant in the downstream tasks of taxonomy expansion and unseen pose encoding. In addition to the lower stress values (see two last columns of Table 1), we can observe some instances of inconsistent embeddings in the Euclidean models. For instance, the unseen class Qu (green) is embedded on top of the class PE (light green) in the GPLVM latent space of Figure 3e. There is no such overlap in the hyperbolic latent space of the GPHLVM, as this geometry allows us the accommodate the graph-like structure of the data.
>
> Finally, we would like to point out that the introduction of distance-preserving priors and back-constraints is necessary for the embeddings to follow the graph structure of the taxonomy. Neither the vanilla GPHLVM nor the vanilla GPLVM respect the taxonomy graph structure.
>
> **Comparison accounting for computational complexity:**
>
> As mentioned by the reviewer, comparisons with respect to the computational complexity of our approach are presented in App. F.7. Although we unfortunately lack the space to display this comparison in the main text, we modified the introduction of Section 5 to explicitly refer to the computational complexity comparisons of App. F.7.
>
> **Other applications of the proposed framework:**
>
> Other applications of the proposed framework include:
>
> - Embeddings of continuous data associated to symbolic data such text and graphs. For instance, our model could be used to embed data associated with noun hierarchies such as WordNet [1], e.g., continuous data related to the characteristics of the animal in WordNet mammals subtree. The obtained embeddings would be globally organized according to the WordNet hierarchy and locally organized according to the animal characteristics. Our model could also be used to embed continuous data associated with cell trajectories, which present a hierarchical structure and are well embedded in hyperbolic spaces (see [2]).
> - Motion planning in robotics. In motion planning, environments are usually discretized and represented as graphs. These graphs may be used as a graph priors in the latent space of our GPHLVM. Planning may then be achieved in the latent space by using geodesics.
>
> If the reviewer was referring to possible downstream tasks, we would like to kindly point to our first answer to the comments of Reviewer zvAH.
>
> References:
>
> [1] G. Miller and C.Fellbaum. “Wordnet: An electronic lexical database”. 1998
>
> [2] A. Klimovskaia, D. Lopez-Paz, L. Bottou, and M. Nickel. “Poincaré maps for analyzing complex hierarchies in single-cell data”. Nature Communication 11, 2966 (2020).

---

> > ### Comment · Reviewer_bere · 2023-11-22
> > **Response to authors**
> >
> > I thank the authors for their detailed answers that have largely addressed my concerns. I greatly appreciate the addition of the trajectories generated by the competing methods. I think that the regularized GPLVM manages to produce sufficiently good results, although the proposed GPHLVM certainly shows an advantage. In any case, I think that the overall contribution is important and that the revised version of the manuscript has become stronger thanks to the reviewers' comments, hence I continue to support the acceptance of this work.

---

> > > ### Author Response · Authors · 2023-11-22
> > > **Response to Reviewer bere**
> > >
> > > Thank you very much for your positive feedback! We are grateful for the reviewers' comments and agree that they helped us making our paper stronger.
> > > Thank you also for supporting the acceptance of our paper! We would be grateful if the review and assessment of our paper may be updated accordingly.

---

### Author Response · Authors · 2023-11-19
**Summary of changes**

We would like to thank all reviewers for taking the time to review our work and for their constructive comments and questions.

**Summary of the main changes in the paper (highlighted in blue)**:

- Added discussion concerning possible downstream tasks in Sec. 6;
- Added comparisons of the motions generated as trajectories in the latent space of our GPHLVM with (1) motions generated from the latent space of the GPLVM, and (2) motions generated from the latent space of VPoser in App. F.6 and in the supplementary video;
- Added discussions to better position our paper with respect to other human motion generation approaches in Section 6 and App. F. 6;
- Updated second and fourth rows of Figures 2-4 and Figures in the App. F and G to display error matrices between geodesic and graph distances for improved clarity;
- Clarification the construction of the embeddings in Section 3.2;
- Explicit mention of our computational cost analysis in Section 5.

We further address the comments and questions in individual replies to the reviewers. We would be glad to address any remaining concerns of the reviewers.

---

### Author Response · Authors · 2023-11-22
**Feedback on our responses**

Dear Reviewers,

Thank you again for taking the time to review our work and for your constructive comments and questions!

As the discussion period will end shortly, we wanted to check if our responses and the changes that we made in our paper answered your comments? We would also be glad to answer any remaining questions.

Thank you in advance for your replies!

---

### Meta-Review · Area_Chair_zF62 · 2023-12-09

**Metareview:**

This paper designs hyperbolic embeddings for robotic grasps (single- and dual-arm) and whole-body poses. The proposed model, GPHLVM, extends the well-known Gaussian Process Latent Variable Model (GPLVM) to account for the hyperbolic geometry of the latent space. To do so, the paper provides a hyperbolic kernel, and performs optimization on a Riemannian manifold. The paper also advocates for preserving distances in the pose graph into distances in the latent space. The paper is well-executed and well-written, however, reviewer zvAH mentions that the motivation behind the paper is unclear and I tend to agree with this. The authors replied with a list of possible downstream applications, which was good, but I think this would have been a much stronger paper if it actually showed results from 1-2 of those downstream uses of these embeddings. As such, even though the paper is well-executed and the results support the claims, I am inclined to recommend that the paper be rejected until this is addressed in a future iteration of the work.

**Justification For Why Not Higher Score:**

See reason above.

**Justification For Why Not Lower Score:**

N/A

---

### Decision · Program_Chairs · 2024-01-16

Reject